# Generating with Confidence: Uncertainty Quantification for Black-box Large Language Models

## Abstract

Large language models (LLMs) specializing in natural language generation (NLG) have recently started exhibiting promising capabilities across a variety of domains. However, gauging the trustworthiness of responses generated by LLMs remains an open challenge, with limited research on uncertainty quantification (UQ) for NLG. Furthermore, existing literature typically assumes white-box access to language models, which is becoming unrealistic either due to the closed-source nature of the latest LLMs or computational constraints. In this work, we investigate UQ in NLG for *black-box* LLMs. We first differentiate *uncertainty* vs *confidence*: the former refers to the "dispersion" of the potential predictions for a fixed input, and the latter refers to the confidence on a particular prediction/generation. We then propose and compare several confidence/uncertainty measures, applying them to *selective NLG* where unreliable results could either be ignored or yielded for further assessment. Experiments were carried out with several popular LLMs on question-answering datasets (for evaluation purposes). Results reveal that a simple measure for the semantic dispersion can be a reliable predictor of the quality of LLM responses, providing valuable insights for practitioners on uncertainty management when adopting LLMs.

## 1 Introduction

Large language models (LLMs) have recently gained significant attention in natural language generation (NLG) (Touvron et al., 2023a; Katz et al., 2023; OpenAI, 2023; Chowdhery et al., 2022). Trained on vast amounts of data, they exhibit impressive abilities in generating human-like responses. As such advances invariably lead to wider adoption of LLM for language generation tasks, such as question-answering (QA), it is crucial to quantify their uncertainty.

Uncertainty quantification (UQ) is well-studied within machine learning. A reliable measure of uncertainty is important to decide when to trust a model. When a model shows high uncertainty or returns low-confidence predictions[1], the input should either be rejected or yielded to further evaluation (Gal & Ghahramani, 2016). This is an area of active research typically called selective classification (Chow, 1970; Lin et al., 2022b; Geifman & El-Yaniv, 2017). Similarly, in NLG, one might refuse the generation $\mathbf{s}$ by the LLM to the input $x$ with high uncertainty/low confidence. Selective generation basing on uncertainty estimates could potentially improve the decision making process, especially for high-stakes applications such as medical or legal question-answering.

While UQ has been an important topic for classical machine learning tasks like classification or regression (Lakshminarayanan et al., 2017; Gal & Ghahramani, 2016; Hernández-Lobato & Adams, 2015; Abdar et al., 2021a), it bears specific challenges for NLG and has attracted limited attention until recently (Lin et al., 2022a; Kuhn et al., 2023; Malinin & Gales, 2021). Numerous challenges hinder the direct application of UQ methods from classification or regression. First, the output space has forbiddingly high dimensionality, rendering most measures, such as the exact entropy of predicted probabilities, unfeasible. Furthermore, distinct token sequences may convey identical meanings, necessitating an effective uncertainty measure to operate in the semantic meaning

---

[1]See Section 3.2 for a discussion of uncertainty vs. confidence.

space (Kuhn et al., 2023). Lastly, many existing LLMs are black-boxes served via APIs, implying that end-users typically lack white-box access. Even when such access is available, many LLMs are too large to run for most users. Note that such considerations are orthogonal to the problem of overconfidence: We need a ranking-wise informative confidence/uncertainty measure before the problem of overconfidence even appears (see discussion in Section 2).

In this study, we explore the problem of uncertainty quantification in NLG with black-box LLMs. Unlike some prior research (Mielke et al., 2020; Lin et al., 2022a; Kadavath et al., 2022; Kuhn et al., 2023), we focus on the more realistic scenario where we only possess access to the generated text, rather than the numerical outputs such as token-level logits. We first introduce a set of measures designed to assess the uncertainty of the input and the confidence of each generation - The main idea entails estimating the uncertainty/confidence from multiple generations from the LLM. Then, we demonstrate the application of these uncertainty estimates in the context of free-form QA. QA datasets are used for the simplicity of evaluation, as completely open-ended NLG tasks requires expensive human evaluation. The main contributions of this paper are summarized as follows:

- We investigate uncertainty quantification for black-box LLMs, a previously under-explored topic, and assess its value in the downstream task of selective natural language generation.
- We put forward several simple yet effective techniques for estimating uncertainty associated with input data and determining the confidence level of each individual generated response.
- Through extensive experiments on several popular LLMs and question-answering datasets, we observe that proposed measuress demonstrate a high level of effectiveness in pinpointing challenging (uncertain) questions and predicting the quality of their corresponding answers.

## 2 RELATED WORKS

The quantification of uncertainty has emerged as a significant area of research across various machine learning domains, including natural language processing (NLP). However, previous studies in NLP have predominantly addressed the associated uncertainty quantification challenges in the same vein as classification or regression methodologies (Desai & Durrett, 2020; Jiang et al., 2021; Kamath et al., 2020; Wang et al., 2022; Xiong et al., 2023). For instance, Kamath et al. (2020) examines the selective question-answering task as a multiple-choice problem, reducing it to a de facto classification task rather than directly engaging with free-form generation. As was recently argued in Kuhn et al. (2023), such approaches enable the application of uncertainty quantification measures akin to those employed in more extensively researched classification or regression contexts, but overlook the generative aspects and distinct challenges inherent to NLG.

Recently, some research has started to study uncertainty quantification for NLG. One line of research involves asking the LLM itself for its confidence, with or without additional fine-tuning (Kadavath et al., 2022; Lin et al., 2022a; Mielke et al., 2020; Chen & Mueller, 2023). Apart from being expensive, such approaches can be hard to generalize due to opaque training details or differences between LLMs (Kuhn et al., 2023). The work most relevant to ours is Kuhn et al. (2023), which proposes to compute the "semantic entropy" by considering the equivalence relationships amongst generated answers, and requires no training. Nonetheless, it still requires access to the token-level numerical output of the LLM, which is not always available.

As discussed, one of the most pertinent applications of uncertainty quantification in NLG involves the development of methods for selective NLG (or, NLG with rejection). This emerging field has limited research to date, but shares close ties with *classification with rejection*. Both tasks can be viewed as determining when to trust a model, whether it is a classifier or an LLM. Numerous classification with rejection methods emphasize the identification of a reliable confidence score (some of which are jointly trained with the classifier) (Corbière et al., 2019; Fumera et al., 2000; Geifman & El-Yaniv, 2017; Jiang et al., 2018), which is often not only dependent on the input but also on the prediction. As existing uncertainty quantification research for NLG primarily focuses on input uncertainty (Kuhn et al., 2023; Malinin & Gales, 2021), it overlooks the crucial aspect of confidence, which is essential in deciding when to trust an LLM's response (see Section 3.2 for more discussion). Recent works have explored selective classification in NLP tasks (Varshney et al., 2022a;b). However, the distinct generative nature of NLG precludes the direct adaptation of confidence measures from the classification with rejection literature. This paper serves as a step to bridge this gap and enhance the effectiveness of uncertainty quantification in NLG.

It is worth noting that the issue of LLMs being overconfident (as discussed in (Mielke et al., 2022; Si et al., 2022; Xiong et al., 2023)) is orthogonal to our work, as we evaluate measures basing how they *rank* different samples - such measures may then be calibrated by distribution-free uncertainty quantification methods like in Schuster et al. (2022). Giulianelli et al. (2023) also provides an interesting exploration of the inherent uncertainty in human responses for many NLG tasks, in a black-box manner. Finally, many methods have been proposed to improve the quality of the generated responses in general, typically by better prompting (Zhou et al., 2023; Si et al., 2023; Wei et al., 2022). Orthogonal to UQ but related to selective NLG, Varshney & Baral (2023) focuses on reattempting rejected samples, with the help of an auxiliary model trained on an additional dataset that predicts the correctness of the generation. This paper focuses on providing uncertainty/confidence measures quantitatively, although such measures could also be used to identify high-quality generations.

## 3 BACKGROUND

In this section, we describe the specific type of uncertainty under examination in the context of Natural Language Generation (NLG), while introducing terminologies used in the rest of the paper.

### 3.1 PREDICTIVE UNCERTAINTY IN NLG

Although we do not adopt a specific Bayesian approach, predictive uncertainty is a prevalent subject within Bayesian statistical analysis (Malinin & Gales, 2018; Gal & Ghahramani, 2016). Consequently, we will utilize their terminology and language in our presentation. Recall that the predictive uncertainty quantifies the degree of dispersion present in the posterior distribution of $Y$, conditioned on the input $X = x$. As it is generally a characteristic of the posterior distribution, we denote it as $U(x)$. For example, when $Y|X = x$ adheres to a Gaussian distribution, the variance serves as an indicator of the uncertainty. Note that NLG can be viewed as a classification problem characterized by an exceedingly high dimension.

In classification, the uncertainty is frequently measured by the entropy of the prediction (e.g. Abdar et al. (2021b); Kuhn et al. (2023); Sun et al. (2019); Wellmann & Regenauer-Lieb (2012)). The predictive entropy is formally defined as $H(Y|x) = -\int p(y|x) \log (p(y|x)dy$. Drawing a parallel, we could define the **uncertainty score** in NLG as:

$$U(x) = H(\mathbf{S}|x) = -\sum_{\mathbf{s}} p(\mathbf{s}|x) \log (p(\mathbf{s}|x)). \tag{1}$$

Here, $x$ represents the input, $\mathbf{S}$ represents the random sequence of generated tokens, and the summation is taken over all potential sequences (responses).

Predictive uncertainty is occasionally characterized as total uncertainty, encompassing both epistemic and aleatoric uncertainty. Epistemic uncertainty (model uncertainty) can potentially be reduced with additional information, such as the use of a better model and/or additional training data (Hüllermeier & Waegeman, 2021; Lahlou et al., 2023). For example, an enhanced LLM trained on more math problems could potentially generate better proofs with lower epistemic uncertainty. In contrast, aleatoric uncertainty (data uncertainty) pertains to the irreducible component of uncertainty inherently associated with the data generation process (Senge et al., 2014). In a sense, this is related to the "open-endedness" in NLG. For instance, for the question "when did the Philadelphia Eagles win their latest Super Bowl" (as of 2023), the answer could be either 2017 or 2018, as the game took place in February 2018 but belongs to the 2017 season. Some questions ($x$) intrinsically allow for markedly different answers ($\mathbf{s}$). Although differentiating aleatoric vs epistemic uncertainty is interesting, such decomposition is often complex and typically not required for learning algorithms in real-world predictive applications (Hüllermeier & Waegeman, 2021).

Like most existing literature, we focus on quantifying the total uncertainty[2]. We would like to emphasize again that like Kuhn et al. (2023), we adopt QA datasets for the simplicity of evaluation. Methods proposed in this paper could still potentially be applied to more open-ended tasks with no reference answers, but the question then becomes: First, can we evaluate the quality of the UQ in a

---

[2]In fact, when the level of "open-endedness" of an input distribution is similar, the ranking of total uncertainty should still be indicative of that of the epistemic uncertainty.

scalable fashion? Given that extensive human evaluation might be necessitated, and second, do we still care about the level of uncertainty when these questions are intrinsically open-ended?

## 3.2 UNCERTAINTY VS. CONFIDENCE

Uncertainty and confidence are sometimes deemed antonyms. However, confidence scores typically bear a slightly different meaning, especially outside the Bayesian literature. Specifically, while $U$ only depends on $x$ and is a property of the posterior perceived by the model, the confidence scores are generally associated with both the input and the prediction and can be expressed as $C(x, y)$ (Chow, 1970; Corbière et al., 2019; Jiang et al., 2018; Lin et al., 2022b). As a concrete example in the Bayesian perspective, for a posterior $P(Y|x) = \mathcal{N}(\mu, \sigma^2)$, the variance of the posterior $\sigma^2$ is an uncertainty measure. For a particular prediction $Y = y$, the negative z-score $-\frac{|y-\mu|}{\sigma}$ could be a confidence measure. Notice the use of a lower-case $y$ in the notation, instead of a upper-case $Y$ that represents a random variable. In the context of classification, one of the simplest and most used confidence measures is just the predicted probability $\hat{p}(Y = y|x)$ (Geifman & El-Yaniv, 2017; Hendrycks & Gimpel, 2017). The corresponding **confidence score** in NLG is the joint probability:

$$C(x, \mathbf{s}) = \hat{p}(\mathbf{s}|x) = \prod_i \hat{p}(s_i|s_{<i}, x). \tag{2}$$

Obviously, Eq. (2) requires access to the original LLM[3]. In Section 4, we will elaborate some alternatives that do not require such white-box access.

Existing literature sometimes uses uncertainty estimate $U(x)$ to predict the correctness of a particular response $\mathbf{s}$ (Kuhn et al., 2023; Malinin & Gales, 2021), ignoring the distinction between uncertainty and confidence. Section 5 shows that this is problematic, and confidence serves as a more reliable indicator of the correctness of a given response.

## 4 QUANTIFYING THE UNCERTAINTY FOR NLG

In this section, we discuss several uncertainty quantification methods that can be applied to blackbox LLMs. Some of these methods are sourced from the existing literature, while the majority are simple proposals of our own. The discussed methods can be structured as taking the following steps:

1. For a given input $x$, generate $m$ response samples $\mathbf{s}_1, \ldots, \mathbf{s}_m$.
2. Calculate the pairwise similarity scores $a(\mathbf{s}_{j_1}, \mathbf{s}_{j_2})$ for these $m$ responses.
3. Compute an uncertainty estimate $U(x)$ or a confidence score $C(x, \mathbf{s}_j)$ using the similarity values.

### 4.1 MEASURING RESPONSE SIMILARITIES

We mainly focus on two ways to compare the similarity between a pair of responses.

**Jaccard Similarity**: The Jaccard similarity is a widely employed metric for determining the similarity between two sets. It is calculated by dividing the cardinality of the intersection of the two sets by the cardinality of their union. A rule-based metric that is easy to implement, the Jaccard index has been extensively utilized in Natural Language Processing (NLP) tasks (Cronin et al., 2017; Pilehvar et al., 2013; Qurashi et al., 2020), where sentences or documents are treated as sets of words. Specifically, the Jaccard similarity between two responses $\mathbf{s}_{j_1}$ and $\mathbf{s}_{j_2}$ (considered as sets of words) where $j_1, j_2 \in [m]$ is computed as:

$$a_{Jaccard}(\mathbf{s}_{j_1}, \mathbf{s}_{j_2}) = |\mathbf{s}_{j_1} \cap \mathbf{s}_{j_2}| / |\mathbf{s}_{j_1} \cup \mathbf{s}_{j_2}| \in [0, 1]. \tag{3}$$

Despite the computation efficiency, Jaccard similarity has certain limitations, including the lack of consideration for word order and the inability to capture crucial expressions such as negation.

**Natural Language Inference (NLI)**: As noted above, rule-based similarity may not effectively capture the nuances present in generated responses. A potential alternative approach involves utilizing a Natural Language Inference (NLI) classifier for this task. Numerous NLI datasets are available for

---

[3]Here, we assume a auto-regressive LLM. For other models, computing such a quantify might require a different approach, if possible.

training such classifiers (Williams et al., 2018; Bowman et al., 2015; Poliak, 2020). In Section 5, we will adopt the methodology outlined by Kuhn et al. (2023) and employ an off-the-shelf DeBERTa-large model (He et al., 2021) as the classifier. A NLI classifier typically predicts scores (logits) for three distinct classes: entailment, neutral, and contradiction. We can use the predicted probabilities as the similarity, denoted as $a_{NLI}(\mathbf{s}_{j_1}, \mathbf{s}_{j_2})$. To obtain a continuous value ranging from 0 to 1, we apply the softmax function to the predicted logits, resulting in $\hat{p}_{contra}(\mathbf{s}_{j_1}, \mathbf{s}_{j_2})$ and $\hat{p}_{entail}(\mathbf{s}_{j_1}, \mathbf{s}_{j_2})$ (both depend on $x$). We then define the following:

$$a_{NLI,entail}(\mathbf{s}_{j_1}, \mathbf{s}_{j_2}) = \hat{p}_{entail}(\mathbf{s}_{j_1}, \mathbf{s}_{j_2}) \quad a_{NLI,contra}(\mathbf{s}_{j_1}, \mathbf{s}_{j_2}) = 1 - \hat{p}_{contra}(\mathbf{s}_{j_1}, \mathbf{s}_{j_2}). \quad (4)$$

It should be emphasized that obtaining $\hat{p}_{entail}$ and $\hat{p}_{contra}$ is not in conflict with our primary objective of quantifying the uncertainty of a black-box LLM, for two key reasons. Firstly, the NLI model can be (and is) substantially smaller than the LLM, because NLI is a considerably simpler task, and the NLI model is not required to have the same "knowledge" as the LLM. Secondly, the LLM's function in NLG is to generate responses (sequences of tokens); thus, any additional information, such as token-level logits or embeddings, is not part of the standard output and may not be accessible to users. In contrast, the NLI model's output *is* the probabilities we utilize.

## 4.2 Estimating Uncertainty and Confidence from Similarities

In this section, we aim to convert similarities from Section 4.1 into uncertainty/confidence measures.

**Number of Semantic Sets** was first proposed in Kuhn et al. (2023). The original paper proposed to use a NLI classifier to group responses into several "semantic equivalence" subsets (which form a partition of all responses). They use such "semantic equivalence" classes as well as the numerical output of the base LLM to compute the "semantic entropy"[4]. While such a method cannot be applied to a black-box LLM, in their experiments they also used the *number* of "semantic sets" (equivalence classes), which is an uncertainty measure applicable to black-box LLMs[5]. We denote this uncertainty measure as $U_{\texttt{NumSet}}$[6]. For example, for the question "What city was Zeus the patron god of?", the three responses "Olympia", "Zeus was the patron god of Olympia, Greece", and "Corinth" form two semantic sets (with the first two responses in one set). Intuitively, if in the $m$ responses, the LLM generates more semantically different answers, then the total uncertainty is high.

**Sum of Eigenvalues of the Graph Laplacian** In reality, whether two responses share the same meaning is not black-and-white. In the example of Zeus above, potential responses "Olympia" and "Greece" are neither exactly the same nor completely different. Moreover, there is no guarantee that the semantic equivalence judged by the NLI model (or any other measure) is transitive. As a result, a more nuanced and "continuous" way to measure the number of meanings is preferable.

Since we only know the pairwise similarities $a_{j_1,j_2}$ between response $\mathbf{s}_{j_1}$ and $\mathbf{s}_{j_2}$, but not the embeddings of the generated responses, a natural choice for the clustering responses is spectral clustering. Fixing an input $x$, we first treat each generated response as one node and define the symmetric weighted adjacency matrix as $W = (w_{j_1,j_2})_{j_1,j_2=1,\ldots,m}$ where $w_{j_1,j_2} = (a_{j_1,j_2} + a_{j_2,j_1})/2$. The symmetric normalized graph Laplacian is then given by

$$L := I - D^{-\frac{1}{2}} W D^{-\frac{1}{2}} \qquad \text{where } D_{j_1,j_2} = \begin{cases} \sum_{j' \in [m]} w_{j_1,j'} & (j_1 = j_2) \\ 0 & (j_1 \neq j_2) \end{cases} \quad (5)$$

A continuous version of $U_{\texttt{NumSet}}$ could be defined with $\lambda_1 < \cdots < \lambda_m$, the eigenvalues of $L$:

$$U_{\texttt{EigV}} = \sum_{k=1}^{m} \max(0, 1 - \lambda_k). \quad (6)$$

To see the connection, between $U_{\texttt{EigV}}$ and $U_{\texttt{NumSet}}$, we recall the classical theorem:

---

[4]This is an improved version of Eq. (1), where $\sum_{\mathbf{s}}$ is replaced with $\sum_c$ where $c$ denotes a semantic concept instead of a response. $p(\mathbf{s}|x)$ is replaced with $p(c|x) = \sum_{\mathbf{s} \in c} p(c|x)$ correspondingly.

[5]https://github.com/lorenzkuhn/semantic_uncertainty

[6]We follow the bi-directional entailment algorithm in Kuhn et al. (2023) to construct such semantic sets using $a_{NLI,entail}$ and $a_{NLI,contra}$ and assuming transitivity of semantic equivalence. Specifically, we iterate over $j_1 < j_2$, and merge $\mathbf{s}_{j_2}$ into the same semantic set of $\mathbf{s}_{j_1}$ if $\hat{p}_{entail}(\mathbf{s}_{j_1}, \mathbf{s}_{j_2}) > \hat{p}_{contra}(\mathbf{s}_{j_1}, \mathbf{s}_{j_2})$ and $\hat{p}_{entail}(\mathbf{s}_{j_2}, \mathbf{s}_{j_1}) > \hat{p}_{contra}(\mathbf{s}_{j_2}, \mathbf{s}_{j_1})$.

**Theorem 1.** *(Von Luxburg, 2007) The multiplicity of the eigenvalue* $0$ *of* $L$ *is equal to the number of connected components in the graph represented by* $W$.

In other words, with a binary $W$ (two responses are either connected or not at all), the multiplicity of the zero eigenvalue coincides with the number of semantic sets ($U_{\texttt{NumSet}}$). With a weighted $W$ whose entries are continuous, there is typically only one connected component. However, in spectral clustering, the distribution of the eigenvalues is typically used to determine the number of clusters (Von Luxburg, 2007)[7]. An illustration is provided in Fig. 1, with $U_{\texttt{EigV}}$ roughly corresponding to the "number of semantic meanings". In Eq. (6) we ignore eigenvalues larger than 1 as only the smallest few eigenvalues carry important information about the clusters (Von Luxburg, 2007).

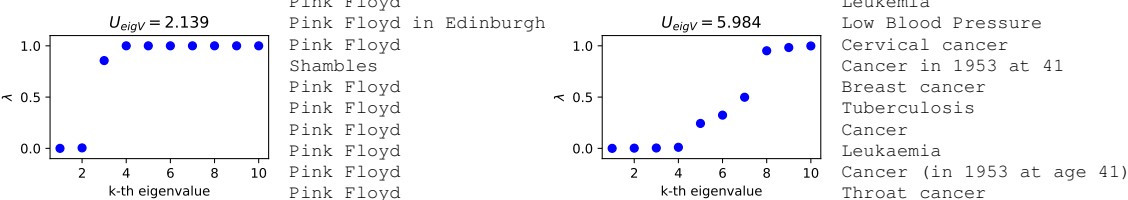

Figure 1: The distribution of the eigenvalues of the graph Laplacian generated by $a_{NLI,entail}$, for two questions from `trivia`, as well as the 10 generated responses. On the left, the question and reference answer are "Q: Dave Gilmore and Roger Waters were in which rock group? A: Pink Floyd". We have somewhere between two and three meanings, and $U_{\texttt{EigV}}$ is slightly above 2. The question on the right is "Q: What claimed the life of singer Kathleen Ferrier? A: Cancer", and we observe more diverse responses. As a result, we see a higher $U_{\texttt{EigV}}$ (almost 6).

**The Degree Matrix** The previous two methods helped us define the uncertainty $U(x)$ but cannot assign a confidence score to each generated response. We utilize the degree matrix $D$ in Eq. (5) to define both metrics, as $D$ already contains relevant information: A node with high degree is well-connected to other nodes, suggesting that it lies in a confident region of the LLM. We thus define uncertainty estimate $U_{\texttt{Deg}}(x)$ and confidence score $C_{\texttt{Deg}}(x, \mathbf{s}_j)$ as

$$U_{\texttt{Deg}}(x) = trace(m - D)/m^2 \qquad\qquad C_{\texttt{Deg}}(x, \mathbf{s}_j) = D_{j,j}/m. \qquad (7)$$

Here, we assume $W_{j_1,j_2} \in [0,1]$. $U_{\texttt{Deg}}$ can also be interpreted as the average pairwise distance.

**Eccentricity** Recall that one challenge from earlier is that we only have the similarity (or distance) between different responses, but do not know their actual embedding space. The graph Laplacian, however, can provide us with coordinates for the responses. Denote $\mathbf{u}_1, \ldots, \mathbf{u}_k \in \mathbb{R}^m$ as the smallest $k$ eigenvectors of $L$, then an informative embedding of $\mathbf{s}_j$ is simply $\mathbf{v}_j = [u_{1,j}, \ldots, u_{k,j}]$ (Ng et al., 2001; Von Luxburg, 2007). As a result, we could use the average distance from center as the uncertainty measure, and each response's distance from the center as the (negative) confidence. Formally, the "eccentricity" estimates are:

$$U_{\texttt{Ecc}}(x) = \|[\mathbf{v}_1^{'\top}, \ldots, \mathbf{v}_m^{'\top}]\|_2 \qquad\qquad C_{\texttt{Ecc}}(x, \mathbf{s}_j) = -\|\mathbf{v}_j'\|_2 \qquad (8)$$

where $\mathbf{v}_j' = \mathbf{v}_j - \frac{1}{m}\sum_{j'=1}^m \mathbf{v}_{j'}$ represents the offset from the average embedding. Ren et al. (2023) uses a similar idea for OOD detection for LLMs, which however requires white-box access to the original language model. Fig. 2 illustrates a sample embedding (from `coqa`).

## 5 EXPERIMENTS

In this section, we evaluate the quality of uncertainty and confidence measures proposed in Section 4.

### 5.1 SETUP FOR EXPERIMENTS

---

[7]In particular, a "gap" between a small eigenvalue $\lambda_k$ to a large one $\lambda_{k+1}$ indicates that there are $k$ clusters.

**Datasets** Following Kuhn et al. (2023), we use the open-book conversational question answering dataset, CoQA (`coqa`) (Reddy et al., 2019), and the closed-book QA dataset, TriviaQA (`trivia`) (Joshi et al., 2017). In addition, we also use the more challenging closed-book QA dataset, Natural Questions (`nq`) (Kwiatkowski et al., 2019). We use the development split of `coqa` with 7,983 questions, the validation split of `nq` with 3,610 questions, and the validation split of the `rc.nocontext` subset of `trivia` with 9,960 (de-duplicated) questions. We repeat all experiments 10 times, each time with a random subset of 1,000 questions as the calibration set for hyper-parameters of $U$ and $C$ measures, and test the performance on the remaining data. We report the mean and standard deviation of all evaluation metrics (see Section 5.2).

Figure 2: An example of the 2D UMAP (McInnes et al., 2018) projection of the embeddings used in `Ecc` for 20 responses. The question and answer are "Q: What is the bakery's name? A:the Dominique Ansel Bakery". Similar answers tend to live closely together, justifying the use of distance-based uncertainty and confidence measures ($U_{\mathrm{Ecc}}$ and $C_{\mathrm{Ecc}}$).

**LLMs** We followed Kuhn et al. (2023) and include OPT (Zhang et al., 2022) in our experiments. We also test two more recent models that have demonstrated superior performance: LLaMA (Touvron et al., 2023a), LLaMA2 (Touvron et al., 2023b) and the black-box `gpt-3.5-turbo` served by OpenAI via an API[8]. For both LLaMA and OPT, we use the 13B versions. We use the default generation configs for all models.

**Baselines** We compare all uncertainty and confidence measures listed in Section 4, including `NumSet`, `Deg`, `Ecc` and `EigV`. `Deg`, `Ecc` and `EigV` are constructed with three versions using $a_{Jaccard}$, $a_{NLI,entail}$, $a_{NLI,contra}$ (with suffix J/E/C, respectively). We also include "lexical similarity" ($U_{\mathrm{LexiSim}}$) from Kuhn et al. (2023) which measures the average rougeL between responses. Note that only $U_{\mathrm{NumSet}}$ and $U_{\mathrm{LexiSim}}$ were proposed before (in Kuhn et al. (2023)). To benchmark these methods with the existing literature, we include two *white-box* baselines for non-GPT LLMs:

- Semantic Entropy (`SE`) (Kuhn et al., 2023): This uncertainty estimate ($U_{\mathrm{SE}}$) groups answers like `NumSet`, and then computes the entropy over the aggregated semantic sets. This requires access to the token-level logits from the base LLM.
- `P(true)` (Kadavath et al., 2022): This confidence measure $C_{\mathrm{P(true)}}$ estimates the probability that a model's generation is correct by asking the model itself[9]. We follow the prompts provided in Kuhn et al. (2023); Kadavath et al. (2022), and convert $C_{\mathrm{P(true)}}$ to an uncertainty estimate $U_{\mathrm{P(true)}}$ by taking the average over all responses.

## 5.2 EVALUATION

**Evaluation Metrics**: Effective uncertainty measures must reflect the reliability of LLM responses, with higher uncertainty and lower confidence more likely leading to incorrect generations. Following prior works (Kuhn et al., 2023; Band et al., 2022), we evaluate the quality of the proposed UQ measures by using them to predict whether a generation is correct or not, and compute the Area Under Receiver Operating Characteristic (**AUROC**) for such prediction. Specifically, if we denote $acc_{i,j} = \mathbb{1}\{\mathbf{s}_{i,j} \text{ correctly answers } x_i\}$, we compute average AUROC using $C(x_., \mathbf{s}_{.,j})$ to predict $acc_{.,j}$ for $j \in [m]$. Unless otherwise noted, $m = 20$.

AUROC however bears two limitations: it can only be applied on binary labels, and its value is hard to interpret. Thus, we use Area Under Accuracy-Rejection Curve (**AUARC**) (Nadeem et al., 2009), as an alternative evaluation metric[10]. Illustrated in Fig. 3, Accuracy-Rejection Curve (ARC) is computed as the average *target* (i.e. accuracy) when we reject a subset of the samples basing on *predictor* (i.e. $U$ or $C$). As we reject more high-uncertainty samples, the remaining samples should

---

[8] All `gpt-3.5-turbo` used in this paper are the `0301` version.

[9] Note that `P(true)` may not require white-box access if one is willing to sample a lot of responses from the LLM, which is however computationally expensive.

[10] While originally AUARC was defined with a binary accuracy label, one could generalize it to any continuous label, such as the expected accuracy (using a Monte Carlo estimate).

have higher accuracy. The "Oracle" (max) AUARC is achieved by directly using the *target* as the *predictor*, while the AUARC of a random *predictor* equals to the base accuracy without rejection.

**Correctness of Generations**: We assess the correctness of the generated responses automatically using `gpt-3.5-turbo` from the OpenAI API. This model is provided with the question, reference answer, and LLM-generated response, and it assigns a correctness score between 0 and 1. Responses with scores above 0.7 are deemed correct. Like Kuhn et al. (2023) (which used rougeL score as a heuristic to evaluate the correctness of generated responses), we also perform human verification on the correctness of the auto-generated judgment by `gpt-3.5-turbo` and found that the accuracy is about 0.95 (see the Appendix for more details).

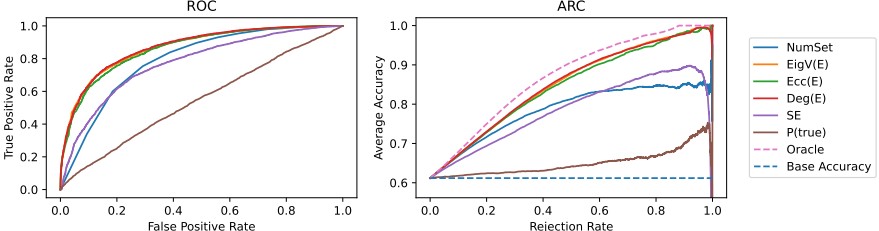

Figure 3: The ROC (left) computed using the accuracy of the 1st generated response and the corresponding confidence $C$ (when available, otherwise $U$), for LLaMA on `trivia`. ARC (right) is computed using $U$ and expected accuracy. (For ARC, average accuracy is noisy at high rejection rate due to small sample size.) The suffix (E) denotes for $a_{NLI,entail}$. Different ways to construct $U$ or $C$ from $a_{NLI,entail}$ make a relatively small difference, but are all noticeably better than the baselines (including the white-box baselines). The ARC suggests that if we select only the top 50% samples using, for example, `Deg` (E), we could improve the accuracy from 62% to around 90%.

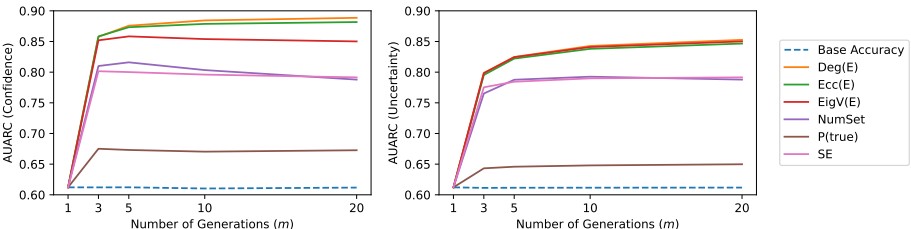

Figure 4: AUARC for confidence (left) and uncertainty (right), with different number of sampled responses. A small $m$ like 3 already greatly improves the accuracy in selective generation, compared with no rejection or rejection based on random noise (Base Accuracy). In general, confidence-based measures (`Deg` and `Ecc`) achieve better performance in the C+IA setting, confirming the intuition that response-dependent confidence measures predict the quality of individual responses better.

### 5.3 RESULTS

**Uncertainty Measures**: We first evaluate the uncertainty measures by using them as the predictor to predict *expected accuracy* (shorthanded as U+EA), estimated with $\hat{\mathbb{E}}[acc_i] = \frac{1}{m} \sum_{j=1}^{m} acc_{i,j}$ ($m=20$). The AUARCs are shown in Table 1 (AUROC is skipped as expected accuracy is not binary). For readability, we include only $a_{NLI,entail}$ which empirically outperforms the best, with full experiment results in the Appendix. We hypothesize that $a_{NLI,entail}$ outperforms $a_{NLI,contra}$ because even if generations do not contradict each other, they could be still mostly meaningless, whereas generations that actually align could indicate low uncertainty. `Deg`, `Ecc`, and `EigV` perform similarly as an uncertainty measure. In some cases, such as `trivia`(`llama`), the AUARC (for `Deg`) is close to the upper-bound (Oracle) (which can be seen from Fig. 3 as well). In general, we found that the choice of similarity (E-entailment, C-contra, J-Jaccard) matters more than the choice of `Deg`, `Ecc`, `EigV`.

**Confidence Measures**: To evaluate confidence measures, we compute AUROC and AUARC, and take the average across all $m$ generations. We refer to this as the "C+IA" (Confidence + Individual

Accuracy) setting. The results are reported in Table 2 (with AUROCs deferred to the Appendix). Note that `Ecc` and `Deg` are actual confidence measures and `EigV` is an uncertainty measure (not response-specific). As a result, able to further distinguish the quality of each response, `Deg` and `Ecc` typically observe higher performance in Table 2 compared with Table 1, while the uncertainty-only measures like `EigV` stay the same.

**Varying the Number of Generations** In Fig. 4, we report results when we vary the number of generations $m$. It is observed that even with $m = 3$, the confidence or uncertainty measures already achieve good performance. As $m$ increases, the quality of $C$ and $U$ typically increases. Note that for AUARC(C+IA), only the confidence measure ($C_{\text{Deg}}$ and $C_{\text{Ecc}}$) improves, and the other measures, being response-agnostic uncertainty measures, stay the same. (`NumSet` seemingly deteriorates as $m$ increases in this plot, because it happens to predict $acc_{.,1}$ better than $acc_{.,j}$ for $j > 1$.)

Table 1: AUARC when using $U(x)$ to predict expected accuracy. The best black-box methods are in **bold** and the best overall is underscored. In general, our proposed uncertainty measures perform significantly better than the baselines, sometimes outperforming the white-box methods.

| | | trivia(llama) | trivia(llama2) | trivia(opt) | trivia(gpt) | coqa(llama) | coqa(llama2) | coqa(opt) | coqa(gpt) | nq(llama) | nq(llama2) | nq(opt) | nq(gpt) |
|---|---|---|---|---|---|---|---|---|---|---|---|---|---|
| | Random | 61.18±0.07 | 76.24±0.11 | 25.75±0.12 | 87.42±0.08 | 62.46±0.11 | 78.71±0.13 | 53.81±0.18 | 79.76±0.14 | 23.63±0.36 | 44.13±0.68 | 8.60±0.18 | 62.72±0.39 |
| | Oracle | 87.03±0.05 | 96.50±0.03 | 54.72±0.19 | 99.09±0.01 | 86.29±0.06 | 96.92±0.03 | 79.41±0.14 | 97.45±0.03 | 47.67±0.55 | 77.62±0.63 | 23.28±0.43 | 90.65±0.21 |
| Baselines | NumSet | 78.78±0.17 | 91.37±0.12 | 39.46±0.29 | 93.18±0.11 | 67.58±0.17 | 83.66±0.17 | 60.41±0.27 | 80.69±0.22 | 28.18±0.55 | 57.55±0.98 | 10.36±0.27 | 68.92±0.74 |
| | LexiSim | 80.32±0.06 | 91.73±0.13 | 45.68±0.24 | 94.69±0.15 | 78.17±0.15 | 89.29±0.16 | 71.46±0.21 | 86.60±0.13 | 40.15±0.70 | 61.88±1.14 | 15.92±0.55 | 73.40±0.74 |
| Ours | EigV (E) | 85.01±0.08 | **93.07±0.17** | 51.54±0.21 | **95.16±0.21** | 81.27±0.12 | **90.21±0.24** | 73.46±0.20 | **88.80±0.10** | **40.42±0.67** | **63.58±0.96** | 18.20±0.44 | **75.17±0.80** |
| | Ecc (E) | 84.66±0.06 | **93.12±0.12** | 51.42±0.22 | **95.21±0.22** | 80.55±0.14 | **90.02±0.22** | 72.73±0.20 | 88.67±0.13 | 40.38±0.68 | **63.41±0.92** | **18.82±0.46** | **75.40±0.49** |
| | Deg (E) | **85.27±0.06** | 93.05±0.14 | **52.06±0.21** | **95.00±0.23** | **81.50±0.15** | **90.18±0.22** | **73.91±0.19** | 88.63±0.23 | **41.07±0.69** | **63.41±1.03** | **18.43±0.48** | 74.35±0.78 |
| White-box | SE | 79.15±0.08 | 93.99±0.06 | 51.11±0.20 | – | 78.83±0.16 | 89.29±0.17 | 70.75±0.21 | – | 36.03±0.54 | 62.40±0.98 | 18.40±0.44 | – |
| | P(true) | 64.98±0.10 | 82.53±0.09 | 20.25±0.11 | – | 64.04±0.18 | 79.92±0.17 | 50.23±0.23 | – | 24.72±0.42 | 44.25±0.65 | 7.63±0.22 | – |

Table 2: AUARC when using $C(x, \mathbf{s})$ to predict individual accuracy. The best black-box methods are in **bold** and the best overall is underscored. Compared with Table 1, the AUARC for confidence measures (`Deg` and `Ecc`) generally improve, as they could discriminate the quality of each response.

| | | trivia(llama) | trivia(llama2) | trivia(opt) | trivia(gpt) | coqa(llama) | coqa(llama2) | coqa(opt) | coqa(gpt) | nq(llama) | nq(llama2) | nq(opt) | nq(gpt) |
|---|---|---|---|---|---|---|---|---|---|---|---|---|---|
| | Random | 61.18±0.07 | 76.24±0.11 | 25.75±0.12 | 87.42±0.08 | 62.46±0.11 | 78.71±0.13 | 53.81±0.18 | 79.76±0.14 | 23.63±0.36 | 44.13±0.68 | 8.60±0.18 | 62.72±0.39 |
| | Oracle | 91.24±0.03 | 96.92±0.03 | 60.68±0.16 | 99.17±0.01 | 91.85±0.05 | 97.55±0.03 | 87.15±0.11 | 97.80±0.03 | 57.69±0.53 | 80.22±0.56 | 29.65±0.45 | 91.97±0.18 |
| Baselines | NumSet | 78.78±0.17 | 91.37±0.12 | 39.46±0.29 | 93.18±0.11 | 67.58±0.17 | 83.66±0.17 | 60.41±0.27 | 80.69±0.22 | 28.18±0.55 | 57.55±0.98 | 10.36±0.27 | 68.92±0.74 |
| | LexiSim | 80.32±0.06 | 91.73±0.13 | 45.68±0.24 | 94.69±0.15 | 78.17±0.15 | 89.29±0.16 | 71.46±0.21 | 86.60±0.13 | 40.15±0.70 | 61.88±1.14 | 15.92±0.55 | 73.40±0.74 |
| Ours | EigV (E) | 85.01±0.08 | 93.07±0.17 | 51.54±0.21 | **95.16±0.21** | 81.27±0.12 | **90.21±0.24** | 73.46±0.20 | **88.80±0.10** | 40.42±0.67 | **63.58±0.96** | 18.20±0.44 | **75.17±0.80** |
| | Ecc (E) | 88.17±0.11 | **93.21±0.07** | 55.82±0.21 | **95.11±0.18** | **84.62±0.13** | **90.21±0.25** | **78.14±0.20** | 88.42±0.19 | 45.78±0.69 | **63.89±1.08** | **21.00±0.49** | **75.43±0.67** |
| | Deg (E) | **88.86±0.05** | **93.19±0.11** | **56.11±0.19** | 94.94±0.16 | **84.60±0.15** | 89.75±0.19 | 77.83±0.20 | 88.02±0.31 | **46.54±0.69** | **63.55±1.01** | **21.30±0.50** | 74.67±0.64 |
| White-box | SE | 79.15±0.08 | 93.99±0.06 | 51.11±0.20 | – | 78.83±0.16 | 89.29±0.17 | 70.75±0.21 | – | 36.03±0.54 | 62.40±0.98 | 18.40±0.44 | – |
| | P(true) | 67.27±0.11 | 82.18±0.08 | 20.89±0.12 | – | 66.23±0.19 | 79.55±0.16 | 49.84±0.23 | – | 27.62±0.46 | 44.34±0.64 | 8.07±0.21 | – |

## 6 CONCLUSION AND DISCUSSION

In this paper, we studied the problem of uncertainty quantification in black-box LLMs, with an emphasis on assessing the quality of generated responses to a diverse set of questions. We developed and tested a range of easily implementable uncertainty and confidence measures. Our results demonstrated that using similarity as determined by an NLI model, along with simple measures that measure dispersion based on these similarities, can effectively identify difficult questions and confident answers, often outperforming existing white-box benchmarks. The objective of this paper is to provide practitioners with simple and effective methods to manage uncertainty, reduce incorrect answers (possibly by excluding them), and apply LLMs with confidence.

To conclude, we also note some limitations of our work and challenges remaining to be addressed. Currently, the evaluation of uncertainty/confidence measures is restricted to question-answering tasks, because it is generally difficult to acquire labels on whether a response is "reliable" or not for open-ended conversations. Even for the datasets used in our paper, while our use of GPT as the judge improves upon previous heuristics (of using rougeL), the evaluation can be improved if human labels are used. Moreover, we currently evaluate uncertainty and confidence separately. Methods that consider both have the potential to predict the quality of generations better. Finally, the uncertainty and confidence measures in this work (and those mentioned in Section 2) reflect those in the "posterior" represented by the LLM. This has two implications: When the sampling temperature is 0 (i.e. greedy decoding), all methods will give degenerate results, and one might resort to white-box methods or external information to quantify the uncertainty. Also, our methods may not identify factual errors propagated from the training corpus or identify when the LLM is being overconfident (which is related to *calibration*, an orthogonal research topic). We hope this work could serve as a foundation for future research in these directions.

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

## A    PROOF FOR THEOREM 1

Theorem 1 is the same as Proposition 4 in Von Luxburg (2007) (Note the $L$ defined in Theorem 1 is equivalent to $L_{sym}$ in Von Luxburg (2007)). We briefly recover the proof below, beginning with a proposition:

**Proposition 1.** *Von Luxburg (2007) For every $f \in \mathbb{R}^m$*

$$f^\top L f = \frac{1}{2} \sum_{i,j=1}^{m} w_{i,j} \left( \frac{f_i}{\sqrt{d_i}} - \frac{f_j}{\sqrt{d_j}} \right)^2 \tag{9}$$

Proposition 1 can be verified by simple algebra. Now, suppose the graph has $k$ connected components. We first find $k$ orthonormal eigenvectors. This can be done by letting $v_1, \ldots, v_k$ be $k$ vectors such that

$$v_l(j) = \begin{cases} \dfrac{\sqrt{d_j}}{\sqrt{\sum_{j' \in S_l} d_{j'}}} & j \in S_l \\ 0 & j \notin S_l \end{cases} \tag{10}$$

It is easy to verify that $\forall l = 1, \ldots, k, \|v_l\| = 1$. For $l_1 \neq l_2$, $\langle v_{l_1}, v_{l_2} \rangle = 0$ because $S_{l_2}$ and $S_{l_1}$ are disjoint. Finally, the $i$-th entry of $\sqrt{\sum_{j' \in S_l} d_{j'}} L v_l$ is 0 if $i \notin S_l$, and

$$\sqrt{d_i} - \frac{1}{\sqrt{d_i}} \sum_j w_{ij} \frac{1}{\sqrt{d_i}} \sqrt{d_i} = 0 \tag{11}$$

if $i \in S_l$ (because $L = I - D^{-\frac{1}{2}} W D^{-\frac{1}{2}}$).

Now, we argue that we cannot find another vector $v$ that is a zero eigenvector. By Proposition 1 $v$ must be be a scaled version of $v_l$ on component $S_l$. w.l.o.g., assume $v(j) \neq 0$, with $j \in S_l$. Then, by Proposition 1, $\exists c$ such that $\forall j' \in S_l, v(j') = c v_l(j')$. This means $\langle v, v_l \rangle \neq 0$. Thus, we cannot find another zero eigenvector. ☐

## B    ADDITIONAL EXPERIMENT DETAILS AND ABLATIONS

### B.1    PROMPTS FOR RESPONSE GENERATION

**TriviaQA**: We use the exact prompt in Touvron et al. (2023a) for TriviaQA, which is reproduced below:

```
Answer these questions:
Q: In Scotland a bothy/bothie is a?
A: House
Q: [Provided question]
A:
```

**Natural Questions** is a much harder dataset than TriviaQA, so we use the same 5-shot prompt version of the prompt in Touvron et al. (2023a) (with 5 questions randomly picked from the training set).

**CoQA**: For CoQA, we use the code provided by Kuhn et al. (2023) to prepare the data[11]. We provide the prompts below for convenience:

```
[The provided context paragraph]
[additional question-answer pairs]
Q: [Provided question]
A:
```

where additional question-answer pairs are preceding turns of the conversation about the paragraph consisting of questions and reference answers.

---

[11]https://github.com/lorenzkuhn/semantic_uncertainty/blob/main/code/parse_coqa.py

## B.2 PROMPTS FOR NLI SIMILARITY

Similar to Kuhn et al. (2023), we feed both the question and answer to DeBERTa-large[12] with the following prompt:

```
[question] [answer_1] [SEP] [question] [answer_2]
```

for any pair of answers to the same question.

## B.3 AUTOMATIC ACCURACY EVALUATION

We use `gpt-3.5-turbo` to evaluate the similarity between each response and the reference answer. The prompts for `coqa`, `trivia` and `nq` are shown below. The few-shot examples are chosen from the training split.

**CoQA**:

```
Rate the level of consistency between the answer to the question and the reference answer, from 0 to 100.
Question: When was the Vat formally opened?
Reference: It was formally established in 1475
Answer: In 1475
Rating: 100.

Question: what is the library for?
Reference: research
Answer: tourism
Rating: 0.

Question: [question]
Reference: [reference answer]
Answer: [generated response]
Rating:
```

**TriviaQA**:

```
Rate the level of consistency between the answer to the question and the reference answer, from 0 to 100.
Question: In Scotland a bothy/bothie is a?
Reference: House
Answer: House
Rating: 100.

Question: Where in England was Dame Judi Dench born?
Reference: York
Answer: London
Rating: 0.

Question: [question]
Reference: [reference answer]
Answer: [generated response]
Rating:
```

**Natural Questions**:

```
Rate the level of consistency between the answer to the question and the reference answer, from 0 to 100.
Question: who makes up the state council in russia
Reference: governors and presidents
Answer: governors and presidents
Rating: 100.

Question: when does real time with bill maher come back
Reference: November 9, 2018
Answer: September 8, 2000
Rating: 0.

Question: [question]
Reference: [reference answer]
Answer: [generated response]
Rating:
```

99.34% of the judgements by GPT can be parsed as an integer between 0 and 100 in the first attempt with a simple `str.split` in Python. We skipped the remaining samples for the experiments in

---

[12]https://huggingface.co/microsoft/deberta-large-mnli

this paper, but if necessary one could devise a better parsing algorithm as well as improved prompts to elicit judgement with higher coverage.

**Verifying the correctness of GPT evaluations** We sample 33 samples per dataset and model (OPT, LLaMA, GPT) and perform a human evaluation of the quality of the GPT judgement. That is, we compare the human judgement on whether a generated response is correct or not with the GPT's judgement, like in Kuhn et al. (2023). In Table 3, we show the breakdown of the accuracy by datasets.

Table 3: The accuracy of the GPT evaluation on the correctness of the responses, measured by the authors. For example, for `trivia`, the human evaluations and the GPT evaluations are aligned on 97 out of the 99 question/answer pairs, leading to an accuracy of 98.0.

|  | coqa | trivia | nq |
|---|---|---|---|
| Accuracy of GPT evaluation | 90.9 | 98.0 | 95.9 |

We also include a few typical examples where the GPT evaluation is more reliable than the rougeL evaluation used in Kuhn et al. (2023)

- "Q: Were they nice to the other critters? A: no". GPT correctly identifies that "They were mean to the other animals" is a correct answer, but the rougeL metric does not capture this logic.
- "Q: Who created the immunity plan? A: the Gulf Cooperation Council". GPT correctly identities that "GCC" is a correct answer (an abbreviation of the Gulf Coorperation Councol", but rougeL cannot identify this mechanically).
- "Q: when was the last bear killed in the uk A: c. 1000 AD". GPT classifies the response "The last bear in the UK was killed over a thousand years ago, in the 9th century, so there is no specific date available" as correct, but rougeL fails to do so.

The GPT evaluation also introduces its own problems, though. For example, when it is used to evaluate its own response to the question "when did it become law to stand for the national anthem" with reference answer "6/22/1942", it rates "There is no federal law that mandates standing for the national anthem in the United States" as 100. However, we believe it is using its own knowledge here and largely ignores the reference answer. Such problems could potentially be improved with better prompts.

### B.4 HYPER-PARAMETERS FOR UNCERTAINTY/CONFIDENCE MEASURES

For all $U$ and $C$ measures involving $a_{NLI,entail}$ and $a_{NLI,contra}$, we need to choose a temperature for the NLI model. The temperature is chosen from 0.1, 0.25, 0.5, 1, 3, 5, and 7. For $U_{\text{Ecc}}$ and $C_{\text{Ecc}}$, we also need to choose a cutoff for eigenvalues. For simplicity we use the same threshold for each experiment/dataset, and the threshold is chosen from 0.4, 0.5, 0.6, 0.7, 0.8, and 0.9.

### B.5 HYPERPARAMETERS

We perform a rough hyperparameter search for methods using the NLI model ($a_{NLI,entail}$ and $a_{NLI,contra}$). We search for temperature within $[0.1, 0.25, 0.5, 1, 3, 5, 7]$. For Ecc we need to set the cut-off for t he eigenvalue to choose the number of clusters, and the threshold is searched within $[0.4, 0.5, 0.6, 0.7, 0.8, 0.9]$.

### B.6 COMPUTATIONAL REQUIREMENTS

We perform all experiments on a machine with 2x AMD EPYC Milan 7513 CPU, 512 GB RAM, 8x A6000 GPUs. Generating 20 responses with OPT/LLaMA for `coqa`, `nq`, `trivia` takes from 2.5 to 15 hours, or from 2 to 11 seconds per question. CoQA takes the longest due to the story input. Running the NLI model for 20 responses (assuming the worst case that no two responses are exactly the same, which means 380 comparisons) takes about 0.8 seconds.

### B.7 FULL EXPERIMENT RESULTS

In the main text, we present only results for $a_{NLI,entail}$. We show results for $a_{Jaccard}$ and $a_{NLI,contra}$ in Tables 4 to 6. We observe that with the default generation configs, the choice of similarity measure ($a_{NLI,contra}, a_{NLI,entail}, a_{Jaccard}$) seems to play a bigger role than the construction of $U$ and $C$ measures. $a_{NLI,entail}$ consistently performs the best, especially on nq. This is likely due to the nature of the questions. For example, for questions starting with "why", for a collection of low-quality random answers, $a_{NLI,contra}$ does not consider them as high-uncertainty because they don't actually contradict each other, but $a_{NLI,entail}$ captures such high-uncertainty case, as illustrated by Fig. 6. On the other hand, for the more factual questions like in Fig. 5, both all similarity measures agree on high-uncertainty cases, as there are few other ways to state the correct answer. It will be interesting to see more future research on the choice of the appropriate similarity for different tasks (which could involve different types of questions). Fig. 7 also provides an example showing why semantic-based similarities are preferred over lexical-based ones.

Table 4: AUARC, $U(x)$ + Expected Accuracy, with $m = 20$ (similar to Table 1). The best black-box methods are in **bold** and the best overall is underscored.

| | Random | Baselines | | | | | | Ours | | | | | | White-box | |
| | | Oracle | NumSet | LexiSim | EigV (C) | Ecc (C) | Deg (C) | EigV (E) | Ecc (E) | Deg (E) | EigV (J) | Ecc (J) | Deg (J) | SE | P(true) |
| *m = 20* | | | | | | | | | | | | | | | |
| trivia(llama) | 61.18±0.07 | 87.03±0.05 | 78.78±0.17 | 80.32±0.06 | 84.97±0.06 | 84.48±0.07 | 85.04±0.06 | 85.01±0.08 | 84.66±0.06 | **85.27±0.06** | 83.94±0.07 | 83.74±0.07 | 84.29±0.06 | 79.15±0.08 | 64.98±0.10 |
| trivia(llama2) | 76.24±0.11 | 96.50±0.03 | 91.37±0.12 | 91.73±0.13 | 93.07±0.16 | 93.12±0.06 | **93.23±0.08** | 93.07±0.17 | 93.12±0.12 | 93.05±0.14 | 92.74±0.06 | 92.62±0.09 | 92.60±0.10 | 93.99±0.06 | 82.53±0.09 |
| trivia(opt) | 25.75±0.12 | 54.72±0.19 | 39.46±0.29 | 45.68±0.24 | 50.16±0.23 | 50.37±0.24 | 50.52±0.22 | 51.54±0.21 | 51.42±0.22 | **52.06±0.21** | 50.60±0.21 | 50.85±0.20 | 51.49±0.20 | 51.11±0.20 | 20.25±0.11 |
| trivia(gpt) | 87.42±0.08 | 99.09±0.01 | 93.18±0.11 | 94.69±0.15 | 94.82±0.23 | 94.82±0.19 | 94.92±0.13 | **95.16±0.21** | **95.21±0.22** | **95.00±0.23** | 94.83±0.15 | 94.70±0.17 | 94.62±0.21 | – | – |
| coqa(llama) | 62.46±0.11 | 86.29±0.06 | 67.58±0.17 | 78.17±0.15 | 80.85±0.11 | 78.81±0.12 | 80.86±0.12 | 80.55±0.14 | **81.50±0.15** | 79.38±0.13 | 79.60±0.13 | 80.39±0.15 | 78.83±0.16 | 64.04±0.18 |
| coqa(llama2) | 78.71±0.13 | 96.92±0.03 | 83.66±0.17 | 89.29±0.16 | 89.73±0.31 | 89.67±0.55 | 89.71±0.38 | **90.21±0.24** | **90.02±0.22** | **90.18±0.22** | 89.28±0.24 | 89.12±0.16 | 89.25±0.22 | 89.29±0.17 | 79.92±0.17 |
| coqa(opt) | 53.81±0.18 | 79.41±0.14 | 60.41±0.27 | 71.46±0.21 | 72.87±0.20 | 70.77±0.22 | 72.92±0.20 | 73.46±0.20 | 72.73±0.20 | **73.91±0.19** | 71.60±0.22 | 71.43±0.22 | 72.25±0.22 | 70.75±0.21 | 50.23±0.23 |
| coqa(gpt) | 79.76±0.14 | 97.45±0.03 | 80.69±0.22 | 86.60±0.13 | **88.72±0.13** | 88.11±0.25 | **88.70±0.17** | **88.80±0.10** | 88.67±0.13 | 88.63±0.23 | 87.66±0.29 | 88.13±0.19 | 88.08±0.19 | – | – |
| nq(llama) | 23.63±0.36 | 47.67±0.55 | 28.18±0.55 | 40.15±0.70 | 36.99±0.64 | 34.29±0.60 | 37.27±0.64 | **40.42±0.67** | 40.38±0.68 | **41.07±0.69** | 40.07±0.70 | 39.78±0.67 | 40.17±0.64 | 36.03±0.54 | 24.72±0.42 |
| nq(llama2) | 44.13±0.68 | 77.62±0.63 | 57.55±0.98 | 61.88±1.14 | **62.89±1.13** | **62.99±0.96** | **63.07±1.13** | **63.58±0.96** | **63.41±0.92** | **63.41±1.03** | 62.21±0.94 | 61.92±1.01 | 61.97±0.90 | 62.40±0.98 | 44.25±0.65 |
| nq(opt) | 8.60±0.18 | 23.28±0.43 | 10.36±0.27 | 15.92±0.55 | 15.05±0.46 | 14.21±0.43 | 15.08±0.47 | 18.20±0.44 | **18.82±0.46** | **18.43±0.48** | 17.98±0.50 | **18.38±0.49** | **18.56±0.48** | 18.40±0.44 | 7.63±0.22 |
| nq(gpt) | 62.72±0.39 | 90.65±0.21 | 68.92±0.74 | 73.40±0.74 | **75.49±0.66** | **75.04±0.70** | **75.21±0.76** | **75.17±0.80** | **75.40±0.49** | 74.35±0.78 | 73.75±0.89 | 73.11±0.77 | 72.95±0.88 | – | – |

Table 5: AUARC, $C(x, \mathbf{s})$ + Individual Accuracy, with $m = 20$ (similar to Table 2). The best black-box methods are in **bold** and the best overall is underscored.

| | Random | Baselines | | | | | | Ours | | | | | | White-box | |
| | | Oracle | NumSet | LexiSim | EigV (C) | Ecc (C) | Deg (C) | EigV (E) | Ecc (E) | Deg (E) | EigV (J) | Ecc (J) | Deg (J) | SE | P(true) |
| *m = 20* | | | | | | | | | | | | | | | |
| trivia(llama) | 61.18±0.07 | 91.24±0.03 | 78.78±0.17 | 80.32±0.06 | 84.97±0.06 | 87.51±0.06 | 88.51±0.06 | 85.01±0.08 | 88.17±0.11 | **88.86±0.05** | 83.94±0.07 | 86.88±0.09 | 87.65±0.06 | 79.15±0.08 | 67.27±0.11 |
| trivia(llama2) | 76.24±0.11 | 96.92±0.03 | 91.37±0.12 | 91.73±0.13 | 93.07±0.16 | 93.23±0.07 | **93.32±0.07** | 93.07±0.17 | 93.21±0.07 | 93.19±0.11 | 92.74±0.06 | 92.58±0.08 | 92.60±0.08 | 93.99±0.06 | 82.18±0.08 |
| trivia(opt) | 25.75±0.12 | 60.68±0.16 | 39.46±0.29 | 45.68±0.24 | 50.16±0.23 | 54.36±0.20 | 54.31±0.21 | 51.54±0.21 | 55.82±0.21 | **56.11±0.19** | 50.60±0.21 | 54.25±0.21 | 55.01±0.19 | 51.11±0.20 | 20.89±0.12 |
| trivia(gpt) | 87.42±0.08 | 99.17±0.01 | 93.18±0.11 | 94.69±0.15 | 94.82±0.23 | 94.79±0.20 | 94.82±0.19 | **95.16±0.21** | **95.11±0.18** | 94.94±0.16 | 94.83±0.15 | 94.58±0.13 | 94.52±0.16 | – | – |
| coqa(llama) | 62.46±0.11 | 91.85±0.05 | 67.58±0.17 | 78.17±0.15 | 80.85±0.11 | 80.07±0.11 | **84.56±0.13** | 81.27±0.12 | **84.62±0.13** | **84.60±0.15** | 79.38±0.13 | 84.03±0.13 | 78.83±0.16 | 66.23±0.19 |
| coqa(llama2) | 78.71±0.13 | 97.55±0.03 | 83.66±0.17 | 89.29±0.16 | 89.73±0.31 | 89.82±0.25 | 89.94±0.19 | **90.21±0.24** | **90.21±0.25** | 89.75±0.19 | 89.28±0.24 | 89.11±0.17 | 89.38±0.20 | 89.29±0.17 | 79.55±0.16 |
| coqa(opt) | 53.81±0.18 | 87.15±0.11 | 60.41±0.27 | 71.46±0.21 | 72.87±0.20 | 72.70±0.18 | 77.23±0.20 | 73.46±0.20 | **78.14±0.20** | 77.83±0.20 | 71.60±0.22 | 75.96±0.22 | 76.98±0.21 | 70.75±0.21 | 49.84±0.23 |
| coqa(gpt) | 79.76±0.14 | 97.80±0.03 | 80.69±0.22 | 86.60±0.13 | **88.72±0.13** | 87.83±0.12 | 88.37±0.14 | **88.80±0.10** | 88.42±0.19 | 88.02±0.31 | 87.66±0.29 | 87.54±0.13 | 87.88±0.14 | – | – |
| nq(llama) | 23.63±0.36 | 57.69±0.53 | 28.18±0.55 | 40.15±0.70 | 36.99±0.64 | 35.20±0.62 | 39.66±0.71 | 40.42±0.67 | 45.78±0.69 | **46.54±0.69** | 40.07±0.70 | 44.41±0.68 | 45.24±0.69 | 36.03±0.54 | 27.62±0.46 |
| nq(llama2) | 44.13±0.68 | 80.22±0.56 | 57.55±0.98 | 61.88±1.14 | **62.89±1.13** | **63.23±0.93** | **63.17±1.06** | **63.58±0.96** | **63.89±1.08** | **63.55±1.01** | 62.21±0.94 | 61.92±1.01 | 62.16±1.09 | 62.40±0.98 | 44.34±0.64 |
| nq(opt) | 8.60±0.18 | 29.65±0.45 | 10.36±0.27 | 15.92±0.55 | 15.05±0.46 | 15.15±0.47 | 16.55±0.47 | 18.20±0.44 | **21.00±0.49** | **21.30±0.50** | 17.98±0.50 | 20.15±0.54 | **20.99±0.51** | 18.40±0.44 | 8.07±0.21 |
| nq(gpt) | 62.72±0.39 | 91.97±0.18 | 68.92±0.74 | 73.40±0.74 | **75.49±0.66** | **75.10±0.68** | **74.99±0.60** | **75.17±0.80** | **75.43±0.67** | 74.67±0.64 | 73.75±0.89 | 72.53±0.82 | 72.58±0.75 | – | – |

Table 6: AUROC, $C(x, \mathbf{s})$ + Individual Accuracy, with $m = 20$. The best black-box methods are in **bold** and the best overall is underscored. The conclusions are similar to Table 8, with confidence measures based on $a_{NLI,entail}$ (E) performing the best.

| | Baselines | | | | | Ours | | | | | | White-box | |
| | NumSet | LexiSim | EigV (C) | Ecc (C) | Deg (C) | EigV (E) | Ecc (E) | Deg (E) | EigV (J) | Ecc (J) | Deg (J) | SE | P(true) |
| *m = 20* | | | | | | | | | | | | | |
| trivia(llama) | 78.79±0.13 | 75.92±0.04 | 86.29±0.07 | 92.61±0.06 | 93.69±0.06 | 86.47±0.14 | 93.51±0.35 | **94.60±0.05** | 84.58±0.08 | 90.14±0.22 | 91.45±0.07 | 76.61±0.10 | 59.21±0.09 |
| trivia(llama2) | 86.02±0.11 | 82.77±0.10 | 89.28±0.16 | 89.79±0.14 | **89.97±0.09** | 89.18±0.19 | 89.78±0.08 | 89.38±0.16 | 87.60±0.05 | 87.07±0.32 | 86.93±0.09 | 89.75±0.07 | 65.05±0.14 |
| trivia(opt) | 75.02±0.16 | 74.49±0.18 | 83.14±0.12 | 91.55±0.07 | 89.64±0.09 | 86.09±0.10 | 92.71±0.30 | **92.83±0.08** | 85.16±0.10 | 89.38±0.07 | 90.60±0.06 | 86.19±0.09 | 41.76±0.11 |
| trivia(gpt) | 74.69±0.24 | 78.18±0.17 | 80.90±0.50 | 80.92±0.54 | 80.88±0.49 | **81.78±0.25** | **81.80±0.22** | 80.50±0.22 | 79.37±0.18 | 77.58±0.24 | 77.42±0.19 | – | – |
| coqa(llama) | 59.34±0.14 | 71.09±0.17 | 76.78±0.11 | 76.24±0.17 | 84.47±0.11 | 77.65±0.09 | **85.07±0.12** | 84.91±0.15 | 74.43±0.12 | 81.34±0.12 | 83.13±0.11 | 73.32±0.16 | 55.39±0.17 |
| coqa(llama2) | 63.56±0.25 | 73.69±0.28 | 76.88±0.43 | 77.19±0.25 | 77.41±0.32 | **77.94±0.31** | **78.37±0.60** | 76.76±0.33 | 74.49±0.29 | 73.86±0.26 | 74.97±0.29 | 73.02±0.31 | 50.88±0.27 |
| coqa(opt) | 59.72±0.08 | 71.81±0.15 | 74.67±0.14 | 76.24±0.10 | 82.37±0.28 | 75.61±0.13 | **84.25±0.12** | 84.06±0.09 | 73.16±0.13 | 80.70±0.12 | 82.02±0.10 | 71.68±0.13 | 45.82±0.15 |
| coqa(gpt) | 52.44±0.08 | 63.39±0.19 | **72.26±0.27** | 69.29±0.17 | 70.64±0.25 | **72.17±0.17** | 70.40±0.38 | 69.36±0.87 | 68.18±0.21 | 67.46±0.19 | 68.57±0.19 | – | – |
| nq(llama) | 60.61±0.31 | 75.19±0.32 | 69.41±0.38 | 72.30±0.35 | 73.35±0.39 | 75.05±0.32 | 83.37±0.31 | **83.99±0.29** | 75.40±0.36 | 81.27±0.28 | 82.34±0.34 | 69.79±0.26 | 56.35±0.40 |
| nq(llama2) | 71.02±0.41 | 72.41±0.44 | 75.26±0.87 | 76.07±0.46 | 76.27±0.52 | 76.50±0.41 | **77.13±0.48** | 76.57±0.46 | 74.14±0.38 | 73.46±0.46 | 74.07±0.42 | 73.46±0.40 | 52.12±0.30 |
| nq(opt) | 60.45±0.47 | 70.66±0.69 | 68.01±0.58 | 74.78±0.43 | 71.77±0.47 | 77.03±0.32 | **83.61±0.29** | **83.51±0.30** | 77.55±0.45 | 79.23±0.75 | 82.25±0.41 | 79.16±0.26 | 46.74±0.51 |
| nq(gpt) | 62.88±0.51 | 65.32±0.62 | **69.99±0.66** | **69.71±0.56** | **69.34±0.46** | 69.05±0.63 | **69.44±0.69** | 67.81±0.60 | 66.40±0.69 | 63.38±0.97 | 63.68±0.71 | – | – |

```
Q: Who was the lead singer with Stylistics
A: Russell Thompkins

10 Generations:
David McKee
Darius Rucker
Bruce Watson
Billy Jer osteen
Paul Michael McMurray
Jermaine Rogers
Billy Davis
Steve Winwood
Stacy Lattisaw
David Bennett – yeah it was a 60's,
```

Figure 5: An example where contradiction-based $U_{\text{EigV}}$ indicates high uncertainty. In this case, the entailment-based and Jaccard-based similarity also indicates high uncertainty.

```
Q: why was the plague that struck athens so devastating
A: close quarters and poor hygiene

10 Generations:
There's no cure
a large flock of birds
because the plague was unknown at the time
The plague was devastating then because the diseases were unknown and a mystery to the \
    ancient world and the there where numerous diseases spread
it was so short and caused so much quaratic
Athens was located within a part of Greece that was full of fleas
The plague of Athens was a pandemic that hit the Greek city of Athens and surrounding \
    areas in 429-428 BC
Soldiers returning from battle
because they were
because it happened just as farming season began
```

Figure 6: An example where contradiction-based $U_{\text{EigV}}$ indicates low uncertainty, yet the generated responses are mostly low-quality. Entailment-based similarity suggests high uncertainty.

```
Q: How was he traveling then?
A: he was walking

by foot
by foot
on foot
walking
walked
Walking
walking
on foot
with his legs
with his own feet
```

Figure 7: An example where purely lexical similarity performs worse than semantic-based ones. Although the generations are phrased differently, they all convey the exact same meaning.

## C ADDITIONAL RESULTS

### C.1 NUMBER OF GENERATIONS

In the main text, we include results with only $m = 20$. Here, we show the full results with $m = 3, 5, 10$, in Tables 7 to 9. The results are very similar to when $m = 20$: In general, $a_{NLI,entail}$ seems to provide the best results, and the choice of similarity seems more important than the choice of construction (Ecc, Deg, EigV). As $m$ increases, the performance typically increases as well. With three generations ($m = 3$) we already see noticeable performance boost if we perform selective generation (AUARC).

Table 7: AUARC, $U(x)$ + Expected Accuracy, with $m = 3, 5, 10$ (similar to Table 1). The best black-box methods are in **bold** and the best overall is underscored.

| | Baselines | | | | | | | Ours | | | | | | White-box | |
|---|---|---|---|---|---|---|---|---|---|---|---|---|---|---|---|
| | Random | Oracle | NumSet | LexiSim | EigV (C) | Ecc (C) | Deg (C) | EigV (E) | Ecc (E) | Deg (E) | EigV (J) | Ecc (J) | Deg (J) | SE | P(true) |
| **m = 3** | | | | | | | | | | | | | | | |
| trivia(llama) | 61.13±0.07 | 87.00±0.05 | 76.51±0.15 | 77.79±0.13 | 79.65±0.15 | 79.63±0.29 | 79.62±0.16 | **79.84±0.15** | 79.54±0.16 | **79.86±0.19** | 78.98±0.14 | 78.15±0.23 | 78.99±0.11 | 77.53±0.11 | 64.32±0.08 |
| trivia(llama2) | 76.24±0.11 | 96.50±0.03 | 87.49±0.22 | 88.38±0.20 | **88.89±0.23** | 88.87±0.15 | **88.89±0.30** | 88.69±0.20 | **88.81±0.25** | 88.57±0.29 | 88.29±0.33 | 88.59±0.27 | 93.34±0.07 | 82.25±0.09 |
| trivia(opt) | 25.45±0.11 | 54.20±0.18 | 39.64±0.29 | 41.08±0.17 | 44.06±0.19 | 44.95±0.21 | 44.23±0.20 | **45.48±0.21** | 44.97±0.17 | **45.49±0.21** | 44.02±0.19 | 43.55±0.20 | 44.11±0.18 | 48.87±0.18 | 19.92±0.10 |
| trivia(gpt) | 87.42±0.08 | 99.09±0.01 | 91.17±0.11 | **92.38±0.12** | 92.36±0.31 | 92.22±0.34 | **92.57±0.23** | **92.58±0.25** | **92.53±0.16** | 92.28±0.09 | 92.18±0.25 | 92.28±0.12 | – | – |
| coqa(llama) | 62.46±0.11 | 86.29±0.06 | 67.19±0.16 | 75.28±0.17 | 76.29±0.13 | 75.90±0.20 | 76.30±0.17 | **76.96±0.18** | 76.21±0.25 | **77.00±0.19** | 76.05±0.17 | 75.42±0.26 | 76.11±0.21 | 77.68±0.16 | 63.75±0.20 |
| coqa(llama2) | 78.71±0.13 | 96.92±0.03 | 82.29±0.16 | 86.51±0.21 | **86.82±0.30** | 86.57±0.27 | **86.99±0.23** | 86.97±0.29 | 86.71±0.26 | **86.98±0.31** | 86.48±0.23 | 86.40±0.35 | 86.44±0.30 | 88.82±0.17 | 79.82±0.16 |
| coqa(opt) | 53.78±0.17 | 79.39±0.13 | 59.44±0.19 | 66.15±0.17 | 66.94±0.28 | 66.94±0.18 | 66.95±0.22 | **67.82±0.17** | 67.07±0.25 | **67.78±0.27** | 66.56±0.22 | 65.72±0.36 | 66.52±0.24 | 68.96±0.20 | 50.06±0.23 |
| coqa(gpt) | 79.76±0.14 | 97.45±0.03 | 80.22±0.22 | 85.72±0.26 | **86.46±0.25** | 85.44±0.39 | **86.41±0.29** | **86.32±0.35** | 86.08±0.31 | **86.43±0.28** | 85.87±0.23 | **86.32±0.23** | 85.98±0.20 | – | – |
| nq(llama) | 23.54±0.34 | 47.57±0.52 | 27.77±0.35 | 33.97±0.56 | 32.79±0.56 | 32.98±0.42 | 32.84±0.54 | **35.30±0.49** | 33.98±0.62 | **35.32±0.56** | 34.68±0.61 | 33.35±0.66 | 34.74±0.58 | 33.49±0.48 | 24.66±0.38 |
| nq(llama2) | 44.13±0.68 | 77.62±0.63 | 53.46±0.74 | 57.74±1.15 | **58.05±1.00** | 57.64±0.86 | **58.25±1.10** | **58.61±0.77** | 57.94±0.87 | **58.40±0.79** | 57.74±1.09 | 57.20±1.11 | 57.71±1.05 | 60.65±0.95 | 44.02±0.66 |
| nq(opt) | 8.64±0.18 | 23.32±0.42 | 10.34±0.24 | 12.32±0.37 | 12.94±0.38 | 13.80±0.45 | 13.03±0.40 | **14.90±0.43** | 14.40±0.43 | **14.96±0.46** | 13.99±0.39 | 13.41±0.47 | 14.05±0.38 | 17.34±0.38 | 7.68±0.19 |
| nq(gpt) | 62.72±0.39 | 90.65±0.21 | 66.85±0.97 | 70.02±0.85 | **70.97±0.85** | 70.83±0.82 | **71.06±0.87** | **70.93±0.91** | 70.14±1.14 | **70.86±0.93** | 70.36±0.75 | 69.37±0.90 | 70.13±0.80 | – | – |
| **m = 5** | | | | | | | | | | | | | | | |
| trivia(llama) | 61.15±0.07 | 87.01±0.05 | 78.76±0.14 | 78.86±0.09 | 82.20±0.14 | 82.25±0.10 | 82.32±0.10 | **82.45±0.09** | 82.21±0.12 | **82.47±0.12** | 81.59±0.08 | 81.07±0.09 | 81.62±0.07 | 78.44±0.08 | 64.58±0.09 |
| trivia(llama2) | 76.24±0.11 | 96.50±0.03 | 89.53±0.29 | 90.15±0.14 | **91.11±0.17** | 90.96±0.13 | **91.04±0.23** | 90.93±0.09 | **90.98±0.19** | 90.91±0.16 | 90.71±0.15 | 90.46±0.13 | 90.57±0.19 | 93.60±0.07 | 82.35±0.09 |
| trivia(opt) | 25.45±0.11 | 54.20±0.18 | 41.52±0.31 | 41.96±0.27 | 46.59±0.20 | 47.64±0.26 | 46.87±0.20 | 48.07±0.24 | 48.11±0.16 | **48.33±0.19** | 47.37±0.18 | 46.95±0.20 | 47.59±0.19 | 49.92±0.18 | 19.95±0.11 |
| trivia(gpt) | 87.42±0.08 | 99.09±0.01 | 92.04±0.11 | 93.47±0.17 | 93.62±0.25 | 93.49±0.34 | 93.51±0.34 | **93.72±0.13** | 93.48±0.18 | **93.75±0.12** | 93.57±0.19 | 93.49±0.11 | 93.53±0.12 | – | – |
| coqa(llama) | 62.46±0.11 | 86.29±0.06 | 68.31±0.16 | 76.25±0.17 | 77.98±0.15 | 77.59±0.11 | 78.06±0.07 | **78.79±0.12** | 78.18±0.19 | **78.85±0.17** | 77.74±0.12 | 77.36±0.24 | 77.93±0.14 | 78.25±0.16 | 63.94±0.19 |
| coqa(llama2) | 78.71±0.13 | 96.92±0.03 | 83.10±0.14 | 88.13±0.23 | **88.47±0.11** | **88.44±0.36** | **88.51±0.27** | **88.52±0.30** | **88.41±0.27** | **88.57±0.18** | 88.04±0.20 | 87.63±0.18 | 87.99±0.21 | 89.08±0.16 | 79.88±0.16 |
| coqa(opt) | 53.79±0.17 | 79.39±0.13 | 61.15±0.31 | 67.85±0.22 | 69.18±0.19 | 68.98±0.25 | 69.34±0.24 | **70.29±0.22** | 69.74±0.26 | **70.35±0.23** | 68.97±0.21 | 68.32±0.25 | 69.16±0.18 | 70.06±0.21 | 50.08±0.22 |
| coqa(gpt) | 79.76±0.14 | 97.45±0.03 | 80.39±0.23 | 86.58±0.21 | **87.77±0.16** | 86.88±0.23 | **87.63±0.28** | 87.43±0.16 | 87.34±0.22 | 87.49±0.31 | 86.98±0.20 | 87.40±0.24 | 87.16±0.15 | – | – |
| nq(llama) | 23.53±0.35 | 47.56±0.53 | 28.23±0.43 | 36.52±0.55 | 34.39±0.55 | 34.52±0.53 | 34.69±0.55 | **37.87±0.60** | 37.47±0.59 | **38.14±0.60** | 37.29±0.57 | 36.13±0.61 | 37.21±0.61 | 34.56±0.46 | 24.74±0.42 |
| nq(llama2) | 44.13±0.68 | 77.62±0.63 | 55.51±0.77 | 59.85±1.27 | 60.05±1.23 | **60.15±1.17** | 60.08±0.95 | **60.93±0.85** | 60.68±1.01 | **61.01±0.93** | **60.39±1.04** | 59.45±1.00 | 60.11±1.04 | 61.34±0.97 | 44.19±0.66 |
| nq(opt) | 8.64±0.18 | 23.33±0.42 | 10.67±0.31 | 13.67±0.51 | 13.66±0.38 | 14.08±0.39 | 13.87±0.39 | **16.39±0.44** | 16.28±0.49 | **16.48±0.45** | 16.00±0.40 | 16.29±0.42 | 16.29±0.42 | 18.08±0.42 | 7.64±0.20 |
| nq(gpt) | 62.72±0.39 | 90.65±0.21 | 67.94±1.12 | 71.18±0.83 | **72.59±0.78** | **72.30±1.07** | **72.48±0.81** | **72.74±0.94** | **72.20±0.82** | **72.38±0.70** | 71.37±0.87 | 70.75±0.84 | 71.29±0.67 | – | – |
| **m = 10** | | | | | | | | | | | | | | | |
| trivia(llama) | 61.16±0.07 | 87.02±0.05 | 79.26±0.20 | 79.46±0.05 | 83.98±0.08 | 83.84±0.06 | 84.04±0.08 | 84.10±0.05 | 83.80±0.09 | **84.27±0.06** | 83.32±0.05 | 82.90±0.07 | 83.44±0.06 | 78.99±0.08 | 64.79±0.10 |
| trivia(llama2) | 76.24±0.11 | 96.50±0.03 | 90.79±0.23 | 91.24±0.15 | **92.53±0.10** | **92.43±0.11** | **92.53±0.13** | **92.47±0.14** | 92.36±0.17 | 92.40±0.17 | 92.13±0.10 | 91.92±0.20 | 91.98±0.09 | 93.86±0.06 | 82.35±0.09 |
| trivia(opt) | 25.56±0.11 | 54.39±0.18 | 41.50±0.32 | 43.28±0.27 | 48.79±0.21 | 49.63±0.22 | 49.17±0.21 | 50.23±0.22 | 50.39±0.20 | **50.70±0.19** | 49.58±0.20 | 49.48±0.18 | 50.08±0.18 | 50.80±0.20 | 20.08±0.11 |
| trivia(gpt) | 87.42±0.08 | 99.09±0.01 | 92.81±0.10 | 94.44±0.17 | 94.38±0.25 | 94.44±0.29 | 94.28±0.29 | **94.65±0.16** | 94.66±0.15 | **94.52±0.17** | 94.40±0.15 | 94.29±0.17 | 94.26±0.16 | – | – |
| coqa(llama) | 62.46±0.11 | 86.29±0.06 | 68.33±0.33 | 77.47±0.12 | 79.72±0.12 | 78.59±0.13 | 79.81±0.12 | **80.42±0.15** | 79.79±0.17 | **80.49±0.14** | 78.99±0.11 | 78.89±0.17 | 79.50±0.13 | 78.72±0.16 | 63.99±0.18 |
| coqa(llama2) | 78.71±0.13 | 96.92±0.03 | 83.58±0.12 | 88.85±0.12 | 89.29±0.26 | 89.25±0.21 | 89.20±0.52 | **89.56±0.19** | **89.53±0.25** | **89.54±0.25** | 88.86±0.20 | 88.68±0.23 | 88.86±0.16 | 89.17±0.17 | 79.89±0.17 |
| coqa(opt) | 53.80±0.18 | 79.40±0.13 | 61.40±0.22 | 70.13±0.19 | 71.73±0.21 | 70.61±0.18 | 71.86±0.18 | **72.58±0.22** | 71.98±0.17 | **72.85±0.19** | 71.06±0.21 | 70.67±0.18 | 71.33±0.19 | 70.83±0.21 | 50.16±0.22 |
| coqa(gpt) | 79.76±0.14 | 97.45±0.03 | 80.65±0.22 | 86.59±0.18 | **88.41±0.24** | 87.23±0.50 | **88.42±0.13** | 88.28±0.19 | 87.91±0.32 | **88.14±0.27** | 87.45±0.32 | 87.84±0.30 | 87.74±0.20 | – | – |
| nq(llama) | 23.58±0.35 | 47.61±0.53 | 28.29±0.51 | 38.74±0.65 | 35.63±0.62 | 34.96±0.58 | 36.03±0.62 | **39.35±0.64** | 39.30±0.65 | **39.82±0.65** | 39.08±0.64 | 38.33±0.62 | 39.10±0.63 | 35.56±0.56 | 24.66±0.40 |
| nq(llama2) | 44.13±0.68 | 77.62±0.63 | 56.71±0.95 | 61.12±0.93 | **62.12±1.12** | **61.82±0.93** | **62.09±1.09** | **62.62±1.03** | 62.53±0.99 | **62.60±1.07** | 61.47±1.11 | 60.82±0.96 | 61.26±1.00 | 61.98±0.97 | 44.19±0.63 |
| nq(opt) | 8.62±0.18 | 23.29±0.42 | 10.85±0.20 | 15.16±0.49 | 14.47±0.38 | 14.81±0.42 | 14.77±0.41 | 17.67±0.44 | **18.12±0.44** | **18.12±0.44** | 17.45±0.47 | 17.60±0.46 | **17.79±0.46** | 18.49±0.44 | 7.59±0.22 |
| nq(gpt) | 62.72±0.39 | 90.65±0.21 | 68.85±1.28 | 72.53±0.82 | **74.26±0.64** | **74.24±0.59** | **74.07±0.91** | **74.12±0.78** | **73.98±0.70** | **73.75±0.81** | 72.90±0.79 | 72.48±0.81 | 72.43±0.82 | – | – |

Table 8: AUARC, $C(x, \mathbf{s})$ + Individual Accuracy, with $m = 3, 5, 10$ (similar to Table 2). The best black-box methods are in **bold** and the best overall is underscored.

| | Baselines | | | | | | | Ours | | | | | | White-box | |
|---|---|---|---|---|---|---|---|---|---|---|---|---|---|---|---|
| | Random | Oracle | NumSet | LexiSim | EigV (C) | Ecc (C) | Deg (C) | EigV (E) | Ecc (E) | Deg (E) | EigV (J) | Ecc (J) | Deg (J) | SE | P(true) |
| **m = 3** | | | | | | | | | | | | | | | |
| trivia(llama) | 61.21±0.07 | 91.25±0.03 | 81.00±0.17 | 82.84±0.11 | 84.98±0.14 | 85.59±0.09 | **85.93±0.08** | 85.19±0.17 | 85.83±0.12 | 85.76±0.34 | 83.98±0.10 | 84.02±0.10 | 84.78±0.14 | 80.15±0.10 | 67.51±0.09 |
| trivia(llama2) | 76.26±0.13 | 96.93±0.04 | 88.30±0.23 | 89.28±0.21 | **89.77±0.30** | **89.79±0.17** | **89.78±0.30** | 89.76±0.20 | **89.65±0.21** | 89.68±0.25 | 89.40±0.27 | 89.22±0.18 | 89.33±0.17 | 93.58±0.08 | 82.19±0.12 |
| trivia(opt) | 25.73±0.13 | 60.65±0.17 | 43.64±0.36 | 45.80±0.21 | 49.01±0.18 | 49.94±0.15 | 50.40±0.19 | 50.55±0.25 | 50.79±0.22 | 51.10±0.20 | 48.94±0.20 | 48.53±0.18 | 49.45±0.16 | 52.77±0.17 | 20.46±0.11 |
| trivia(gpt) | 87.45±0.08 | 99.18±0.01 | 91.46±0.12 | **92.61±0.13** | 92.59±0.27 | **92.69±0.17** | **92.61±0.23** | **92.78±0.26** | **92.62±0.15** | 92.50±0.10 | 92.39±0.12 | 92.49±0.07 | – | – | |
| coqa(llama) | 62.62±0.14 | 91.93±0.07 | 69.05±0.22 | 79.80±0.22 | 80.70±0.16 | 80.51±0.16 | **81.57±0.15** | **81.54±0.22** | **81.77±0.26** | **81.57±0.20** | 80.32±0.16 | 80.37±0.26 | 81.34±0.17 | 80.24±0.21 | 66.24±0.20 |
| coqa(llama2) | 78.92±0.16 | 97.60±0.04 | 82.74±0.19 | 87.24±0.19 | **87.60±0.21** | 87.34±0.39 | **87.70±0.16** | **87.67±0.18** | **87.63±0.23** | 87.48±0.23 | 87.18±0.23 | 87.08±0.22 | 87.25±0.18 | 89.17±0.17 | 79.76±0.16 |
| coqa(opt) | 53.70±0.22 | 87.09±0.14 | 61.60±0.23 | 71.18±0.22 | 71.96±0.35 | 72.21±0.28 | 73.17±0.25 | 73.20±0.28 | **73.66±0.36** | **73.31±0.21** | 71.69±0.26 | 71.63±0.27 | 72.68±0.24 | 72.15±0.23 | 49.55±0.26 |
| coqa(gpt) | 79.76±0.17 | 97.80±0.04 | 80.21±0.24 | 85.92±0.31 | **86.69±0.33** | 85.67±0.34 | **86.52±0.25** | **86.58±0.36** | 86.07±0.39 | **86.60±0.30** | 86.09±0.26 | 86.02±0.37 | 86.35±0.22 | – | – |
| nq(llama) | 23.40±0.31 | 57.37±0.46 | 29.00±0.46 | 38.30±0.68 | 36.28±0.73 | 36.30±0.64 | 37.06±0.67 | **39.84±0.67** | **39.93±0.76** | **40.45±0.63** | 39.03±0.66 | 38.67±0.76 | **40.46±0.73** | 36.45±0.52 | 27.15±0.44 |
| nq(llama2) | 44.11±0.75 | 80.20±0.61 | 54.43±0.86 | **59.01±1.31** | **59.22±1.09** | 59.12±1.07 | 59.48±1.05 | 59.78±0.84 | 59.24±1.12 | 59.32±0.95 | 58.91±1.14 | 58.53±1.08 | 58.53±1.08 | 61.22±0.96 | 44.59±0.71 |
| nq(opt) | 8.94±0.24 | 30.49±0.59 | 11.35±0.28 | 14.44±0.43 | 15.00±0.43 | 14.60±0.38 | 15.85±0.40 | **17.76±0.34** | 17.13±0.55 | **18.11±0.44** | 16.81±0.37 | 16.81±0.47 | 17.35±0.43 | 20.39±0.40 | 8.46±0.33 |
| nq(gpt) | 62.42±0.46 | 91.83±0.22 | 67.54±0.93 | 69.96±0.93 | **70.84±0.88** | **70.89±0.87** | **70.83±0.75** | **70.93±0.99** | **71.11±0.91** | **70.89±0.84** | 70.20±0.82 | 69.59±0.90 | 69.98±0.84 | – | – |
| **m = 5** | | | | | | | | | | | | | | | |
| trivia(llama) | 61.22±0.08 | 91.26±0.04 | 81.60±0.17 | 82.01±0.11 | 85.60±0.11 | 86.57±0.18 | **87.42±0.09** | 85.83±0.10 | 87.33±0.10 | **87.57±0.17** | 84.79±0.09 | 85.79±0.16 | 86.48±0.10 | 80.01±0.09 | 67.31±0.10 |
| trivia(llama2) | 76.28±0.12 | 96.93±0.04 | 90.07±0.28 | 90.61±0.16 | **91.62±0.15** | **91.55±0.13** | **91.55±0.24** | 91.44±0.09 | **91.59±0.18** | **91.56±0.14** | 91.17±0.15 | 90.94±0.09 | 91.00±0.16 | 93.75±0.07 | 82.07±0.11 |
| trivia(opt) | 25.54±0.11 | 60.40±0.14 | 44.00±0.36 | 44.67±0.27 | 49.71±0.18 | 50.43±0.23 | 52.13±0.17 | 52.59±0.22 | 53.19±0.18 | **53.48±0.17** | 50.44±0.16 | 51.34±0.26 | 52.09±0.16 | 52.16±0.17 | 20.46±0.10 |
| trivia(gpt) | 87.44±0.08 | 99.18±0.01 | 92.19±0.11 | 93.61±0.19 | **93.74±0.30** | 93.55±0.29 | **93.74±0.23** | 93.87±0.14 | **93.76±0.22** | 93.65±0.18 | 93.71±0.18 | 93.62±0.11 | 93.53±0.10 | – | – |
| coqa(llama) | 62.46±0.12 | 91.86±0.06 | 69.40±0.17 | 79.16±0.17 | 80.70±0.14 | 80.77±0.11 | **83.00±0.14** | 81.73±0.17 | **83.02±0.15** | **83.06±0.18** | 80.49±0.16 | 81.62±0.23 | 82.68±0.14 | 79.69±0.18 | 66.11±0.17 |
| coqa(llama2) | 78.85±0.15 | 97.58±0.04 | 83.45±0.15 | 88.59±0.25 | **88.98±0.42** | 88.30±0.27 | **89.05±0.16** | **89.03±0.28** | 88.80±0.19 | **89.08±0.16** | 88.51±0.20 | 88.26±0.19 | 88.40±0.17 | 89.30±0.18 | 79.78±0.16 |
| coqa(opt) | 53.63±0.22 | 87.04±0.14 | 62.58±0.43 | 70.92±0.23 | 72.27±0.22 | 72.60±0.26 | 75.02±0.20 | 73.60±0.21 | **75.63±0.21** | **75.43±0.21** | 72.21±0.21 | 73.74±0.21 | 74.81±0.22 | 71.82±0.22 | 49.76±0.26 |
| coqa(gpt) | 79.78±0.14 | 97.80±0.03 | 80.73±0.21 | 86.64±0.18 | **87.92±0.15** | 86.75±0.31 | 87.57±0.22 | 87.59±0.16 | 87.22±0.25 | 87.37±0.20 | 87.11±0.19 | 87.14±0.24 | 87.33±0.17 | – | – |
| nq(llama) | 23.41±0.66 | 57.84±0.45 | 29.05±0.58 | 39.82±0.56 | 37.13±0.60 | 35.42±0.58 | 38.63±0.62 | 41.37±0.58 | 43.17±0.55 | **44.00±0.57** | 40.69±0.58 | 41.36±0.58 | 43.13±0.59 | 36.43±0.41 | 27.99±0.53 |
| nq(llama2) | 44.16±0.66 | 80.24±0.54 | 55.92±0.82 | 60.59±1.31 | 60.61±1.23 | **60.92±1.13** | **60.91±1.10** | **61.54±0.92** | **61.52±0.92** | **61.22±0.93** | 61.01±1.08 | 60.51±1.02 | **60.70±0.98** | 61.59±0.97 | 44.48±0.63 |
| nq(opt) | 8.79±0.20 | 30.13±0.49 | 11.26±0.29 | 15.15±0.56 | 15.00±0.38 | 14.63±0.34 | 16.46±0.44 | 18.49±0.45 | 18.60±0.51 | **19.78±0.46** | 18.23±0.47 | 18.97±0.45 | **19.50±0.46** | 19.90±0.41 | 8.46±0.27 |
| nq(gpt) | 62.42±0.40 | 91.83±0.19 | 68.05±1.08 | 71.19±0.86 | **72.65±0.79** | **72.46±0.77** | **72.44±0.92** | **72.76±0.99** | **72.56±0.81** | **72.34±0.80** | 71.31±0.69 | 70.78±0.87 | 70.86±0.87 | – | – |
| **m = 10** | | | | | | | | | | | | | | | |
| trivia(llama) | 61.03±0.07 | 91.16±0.04 | 80.34±0.24 | 80.55±0.07 | 85.26±0.08 | 87.18±0.06 | 88.13±0.07 | 85.39±0.07 | 87.87±0.06 | **88.44±0.06** | 84.48±0.07 | 86.52±0.07 | 87.26±0.07 | 79.60±0.08 | 67.04±0.10 |
| trivia(llama2) | 76.25±0.12 | 96.92±0.03 | 90.88±0.23 | 91.37±0.17 | **92.66±0.11** | **92.76±0.09** | 92.58±0.05 | 92.60±0.14 | **92.69±0.11** | **92.69±0.11** | 92.27±0.11 | 92.03±0.14 | 91.97±0.09 | 93.90±0.06 | 82.03±0.10 |
| trivia(opt) | 25.65±0.11 | 60.54±0.15 | 42.34±0.33 | 44.23±0.27 | 49.90±0.21 | 52.66±0.19 | 53.51±0.19 | 51.38±0.21 | 54.79±0.19 | **55.20±0.17** | 50.72±0.18 | 53.22±0.17 | 54.00±0.17 | 51.64±0.19 | 20.69±0.11 |
| trivia(gpt) | 87.45±0.08 | 99.18±0.01 | 92.90±0.11 | 94.51±0.17 | 94.46±0.25 | 94.47±0.20 | 94.37±0.30 | **94.73±0.16** | 94.75±0.17 | **94.60±0.19** | 94.46±0.15 | 94.23±0.15 | 94.27±0.14 | – | – |
| coqa(llama) | 62.44±0.10 | 91.85±0.05 | 68.64±0.34 | 78.50±0.14 | 80.77±0.13 | 80.35±0.17 | 83.96±0.14 | 81.55±0.16 | **84.09±0.11** | **84.06±0.12** | 80.00±0.13 | 82.59±0.22 | 83.52±0.13 | 79.25±0.16 | 65.98±0.17 |
| coqa(llama2) | 78.70±0.13 | 97.55±0.03 | 83.62±0.12 | 88.88±0.12 | 89.45±0.27 | 89.28±0.38 | **89.59±0.18** | 89.72±0.19 | **89.77±0.24** | 89.33±0.23 | 89.01±0.21 | 88.81±0.21 | 88.97±0.16 | 89.22±0.17 | 79.55±0.17 |
| coqa(opt) | 53.78±0.18 | 87.14±0.11 | 61.93±0.24 | 71.45±0.20 | 72.97±0.20 | 72.89±0.17 | 76.73±0.20 | 73.93±0.24 | **77.39±0.27** | **77.30±0.19** | 72.41±0.21 | 75.59±0.24 | 76.57±0.20 | 71.57±0.22 | 49.96±0.24 |
| coqa(gpt) | 79.81±0.13 | 97.81±0.03 | 80.73±0.21 | 86.64±0.18 | **88.51±0.23** | 87.30±0.15 | **88.36±0.20** | 87.72±0.23 | 87.88±0.13 | 87.52±0.31 | 87.33±0.17 | 87.52±0.31 | 87.48±0.17 | – | – |
| nq(llama) | 23.67±0.36 | 57.75±0.52 | 28.51±0.54 | 39.91±0.64 | 36.62±0.69 | 34.97±0.68 | 39.13±0.72 | 40.57±0.69 | 44.78±0.74 | **45.63±0.71** | 40.35±0.66 | 43.21±0.70 | 44.74±0.67 | 36.16±0.55 | 27.52±0.45 |
| nq(llama2) | 44.07±0.70 | 80.16±0.58 | 56.70±0.94 | 61.28±0.94 | **62.24±1.13** | **62.48±1.08** | **62.32±1.15** | **62.67±1.10** | **63.25±1.13** | **62.58±1.00** | 61.59±1.13 | 61.23±1.08 | 61.57±1.07 | 62.00±0.98 | 44.22±0.67 |
| nq(opt) | 8.66±0.21 | 29.80±0.51 | 11.01±0.24 | 15.71±0.50 | 14.99±0.43 | 14.75±0.45 | 16.71±0.51 | 18.51±0.48 | 20.08±0.49 | **20.87±0.53** | 18.34±0.47 | 19.90±0.47 | **20.53±0.48** | 19.18±0.45 | 8.21±0.25 |
| nq(gpt) | 62.71±0.39 | 91.97±0.18 | 69.06±1.20 | 72.59±0.81 | **74.37±0.64** | **74.18±0.65** | **74.25±0.71** | **74.29±0.62** | **74.40±0.78** | **73.80±0.76** | 72.93±0.79 | 71.98±0.81 | 71.99±0.84 | – | – |

Table 9: AUROC, $C(x, \mathbf{s})$ + Individual Accuracy, with $m = 3, 5, 10$ (similar to Table 9). The best black-box methods are in **bold** and the best overall is underscored.

| | Baselines | | EigV (C) | Ecc (C) | Deg (C) | Ours | | | | | | White-box | |
|---|---|---|---|---|---|---|---|---|---|---|---|---|---|
| | NumSet | LexiSim | | | | EigV (E) | Ecc (E) | Deg (E) | EigV (J) | Ecc (J) | Deg (J) | SE | P(true) |
| **m = 3** | | | | | | | | | | | | | |
| trivia(llama) | 82.58±0.13 | 80.78±0.12 | 86.44±0.12 | 88.32±0.13 | **88.91±0.10** | 87.36±0.17 | **89.01±0.11** | 88.77±0.48 | 84.40±0.12 | 84.46±0.39 | 86.13±0.10 | 77.96±0.14 | 59.42±0.08 |
| trivia(llama2) | 79.25±0.15 | 80.41±0.21 | **82.52±0.43** | 82.64±0.22 | 82.60±0.47 | **82.45±0.17** | 82.34±0.24 | 82.14±0.41 | 80.84±0.21 | 80.46±0.24 | 80.52±0.22 | 88.43±0.08 | 65.00±0.19 |
| trivia(opt) | 78.24±0.09 | 73.52±0.14 | 80.50±0.07 | 82.63±0.33 | 82.63±0.07 | 83.40±0.10 | **83.69±0.46** | **83.75±0.08** | 80.70±0.06 | 79.73±0.19 | 80.97±0.07 | 87.51±0.07 | 41.12±0.10 |
| trivia(gpt) | 66.67±0.21 | 71.93±0.30 | 71.76±1.33 | 72.24±0.35 | 72.34±0.98 | **73.04±0.32** | 72.60±0.49 | 72.11±0.33 | 71.63±0.29 | 70.50±0.45 | 71.22±0.28 | – | – |
| coqa(llama) | 62.67±0.12 | 73.85±0.23 | 76.61±0.40 | 76.94±0.22 | 79.10±0.26 | **79.17±0.23** | **80.02±0.92** | **79.26±0.19** | 75.84±0.15 | 75.90±0.32 | 77.97±0.16 | 75.23±0.24 | 55.11±0.16 |
| coqa(llama2) | 59.63±0.22 | 70.89±0.26 | **72.47±0.24** | 71.70±0.73 | **72.51±0.28** | **72.76±0.40** | **72.53±0.52** | 71.88±0.40 | 70.68±0.22 | 70.21±0.29 | 70.67±0.21 | 72.10±0.30 | 50.97±0.31 |
| coqa(opt) | 63.48±0.18 | 71.84±0.17 | 73.30±0.41 | 74.84±0.31 | 76.31±0.25 | 76.84±0.26 | **78.25±0.38** | 77.14±0.13 | 73.73±0.13 | 73.77±0.49 | 75.56±0.15 | 73.62±0.15 | 45.57±0.16 |
| coqa(gpt) | 51.46±0.10 | 64.37±0.16 | **67.41±0.26** | 64.29±0.16 | 66.56±0.22 | **67.42±0.53** | 65.29±0.64 | 66.18±0.42 | 64.86±0.20 | 64.40±1.06 | 65.33±0.21 | – | – |
| nq(llama) | 61.30±0.33 | 70.48±0.40 | 66.22±0.60 | 68.96±0.47 | 68.42±0.45 | 73.16±0.66 | 71.98±0.76 | **73.94±0.42** | 72.44±0.40 | 70.57±0.86 | **73.78±0.39** | 68.84±0.44 | 56.55±0.60 |
| nq(llama2) | 67.70±0.43 | 70.69±0.55 | 72.14±0.56 | 72.25±0.53 | 72.41±0.43 | **72.92±0.29** | 72.20±0.39 | 72.26±0.35 | 70.86±0.48 | 69.08±1.01 | 70.34±0.45 | 71.15±0.36 | 52.08±0.41 |
| nq(opt) | 61.18±0.52 | 61.07±0.63 | 64.25±0.73 | 65.52±0.87 | 67.02±0.66 | **72.07±0.42** | 69.77±1.65 | 71.16±0.56 | 68.40±0.75 | 66.83±0.63 | 68.55±0.83 | 81.01±0.44 | 47.40±0.80 |
| nq(gpt) | 60.05±0.52 | 62.72±0.75 | **65.27±0.89** | **64.92±0.73** | **65.11±0.57** | **64.82±0.70** | 64.34±0.80 | 64.30±0.75 | 62.66±0.73 | 61.44±0.47 | 62.19±0.73 | – | – |
| **m = 5** | | | | | | | | | | | | | |
| trivia(llama) | 82.94±0.09 | 77.82±0.13 | 87.03±0.10 | 90.09±0.21 | 91.51±0.08 | 87.77±0.12 | 91.56±0.22 | **92.01±0.26** | 85.66±0.08 | 87.70±0.32 | 89.18±0.07 | 77.64±0.10 | 59.16±0.07 |
| trivia(llama2) | 83.12±0.13 | 82.53±0.16 | **86.39±0.19** | **86.56±0.17** | **86.57±0.33** | 86.23±0.17 | **86.44±0.19** | 86.14±0.16 | 84.52±0.17 | 84.20±0.47 | 84.01±0.18 | 89.00±0.08 | 64.82±0.17 |
| trivia(opt) | 79.46±0.11 | 71.23±0.22 | 82.05±0.10 | 85.45±0.10 | 86.12±0.09 | 85.38±0.20 | 88.11±0.37 | **88.32±0.09** | 84.11±0.07 | 84.97±0.44 | 85.64±0.10 | 87.27±0.05 | 41.37±0.10 |
| trivia(gpt) | 69.99±0.22 | 74.81±0.19 | 75.97±0.66 | 75.60±0.76 | 75.92±0.73 | **76.34±0.24** | 75.84±1.15 | 75.49±0.36 | 75.04±0.21 | 74.05±0.26 | 74.16±0.21 | – | – |
| coqa(llama) | 62.60±0.09 | 72.42±0.15 | 76.55±0.27 | 77.01±0.09 | 81.83±0.21 | 79.08±0.17 | **82.22±0.33** | **82.32±0.14** | 76.31±0.11 | 78.75±0.14 | 80.75±0.12 | 74.61±0.19 | 55.15±0.17 |
| coqa(llama2) | 62.01±0.26 | 73.23±0.29 | 75.33±0.29 | 75.10±0.53 | **75.60±0.26** | **75.85±0.53** | **76.03±0.44** | 74.65±0.32 | 73.37±0.25 | 72.63±0.48 | 73.06±0.25 | 72.71±0.33 | 51.04±0.28 |
| coqa(opt) | 63.87±0.11 | 70.68±0.16 | 73.79±0.23 | 75.17±0.14 | 79.24±0.09 | 76.89±0.20 | **81.21±0.11** | 80.73±0.13 | 74.56±0.10 | 77.39±0.26 | 79.09±0.08 | 73.23±0.15 | 45.90±0.12 |
| coqa(gpt) | 51.81±0.10 | 64.88±0.15 | **70.23±0.24** | 66.20±0.21 | 68.70±0.51 | 69.69±0.64 | 67.96±0.60 | 67.96±0.60 | 66.89±0.20 | 67.01±0.53 | 67.43±0.17 | – | – |
| nq(llama) | 62.01±0.37 | 72.21±0.37 | 67.69±0.42 | 67.61±0.56 | 70.47±0.36 | 74.88±0.40 | 77.45±0.39 | **78.40±0.27** | 74.49±0.47 | 74.45±0.68 | 77.51±0.29 | 69.09±0.28 | 57.23±0.55 |
| nq(llama2) | 69.12±0.38 | 71.50±0.52 | 73.21±0.70 | 73.77±0.58 | 73.92±0.66 | **74.72±0.49** | **74.71±0.39** | **74.32±0.43** | 72.84±0.46 | 72.26±0.44 | 72.45±0.47 | 71.95±0.44 | 52.16±0.32 |
| nq(opt) | 62.68±0.48 | 64.64±0.93 | 65.70±0.60 | 68.68±0.68 | 69.79±0.56 | 75.66±0.46 | 75.24±2.49 | **77.34±0.51** | 74.84±0.47 | 74.01±0.62 | 75.29±0.56 | 81.25±0.33 | 47.77±0.63 |
| nq(gpt) | 61.36±0.39 | 63.80±0.82 | **67.56±0.60** | **67.25±0.64** | **67.18±0.71** | 66.55±0.72 | 66.58±0.78 | 65.86±0.66 | 64.19±0.88 | 62.78±0.90 | 62.83±0.84 | – | – |
| **m = 10** | | | | | | | | | | | | | |
| trivia(llama) | 81.26±0.15 | 75.14±0.07 | 86.66±0.13 | 91.69±0.08 | 93.02±0.08 | 87.07±0.08 | 92.88±0.16 | **93.85±0.05** | 85.40±0.07 | 89.29±0.12 | 90.76±0.06 | 77.37±0.10 | 59.02±0.07 |
| trivia(llama2) | 85.16±0.11 | 82.79±0.11 | 88.43±0.14 | 88.67±0.15 | **88.94±0.12** | 88.31±0.22 | 88.68±0.32 | 88.31±0.15 | 86.63±0.09 | 85.99±0.53 | 85.96±0.11 | 89.46±0.08 | 64.88±0.16 |
| trivia(opt) | 77.90±0.14 | 70.51±0.26 | 82.63±0.11 | 89.18±0.09 | 88.50±0.08 | 85.87±0.13 | 91.25±0.32 | **91.42±0.08** | 85.16±0.08 | 87.86±0.26 | 88.94±0.07 | 86.72±0.07 | 41.54±0.11 |
| trivia(gpt) | 72.69±0.25 | 77.40±0.18 | 79.16±0.62 | 78.84±0.57 | 78.81±0.86 | **79.81±0.42** | **79.77±0.46** | 78.76±0.64 | 77.95±0.19 | 76.27±0.67 | 76.44±0.20 | – | – |
| coqa(llama) | 61.20±0.12 | 71.19±0.17 | 76.63±0.11 | 76.27±0.12 | 83.50±0.11 | 78.25±0.12 | **84.03±0.20** | **84.07±0.12** | 75.46±0.12 | 80.37±0.60 | 82.43±0.10 | 73.97±0.16 | 55.10±0.14 |
| coqa(llama2) | 62.97±0.26 | 73.48±0.28 | 76.44±0.31 | 76.33±0.33 | 76.78±0.27 | **77.33±0.27** | **77.51±0.44** | 75.98±0.29 | 74.19±0.26 | 73.53±0.29 | 74.15±0.26 | 72.85±0.31 | 50.87±0.28 |
| coqa(opt) | 62.65±0.10 | 71.33±0.10 | 74.55±0.11 | 75.70±0.14 | 81.55±0.22 | 76.39±0.17 | **83.36±0.35** | **83.15±0.11** | 74.25±0.09 | 79.87±0.36 | 81.33±0.11 | 72.70±0.13 | 46.04±0.14 |
| coqa(gpt) | 52.24±0.12 | 63.89±0.22 | **71.46±0.54** | 67.77±0.20 | 69.97±0.40 | **71.22±0.32** | 69.39±0.38 | 69.19±0.19 | 67.61±0.24 | 67.41±0.57 | 68.17±0.20 | – | – |
| nq(llama) | 61.47±0.31 | 74.24±0.38 | 68.22±0.40 | 70.51±0.35 | 72.32±0.36 | 74.76±0.40 | 80.96±0.61 | **82.13±0.34** | 75.27±0.38 | 78.74±0.59 | 81.23±0.29 | 69.43±0.28 | 56.28±0.49 |
| nq(llama2) | 70.64±0.37 | 72.44±0.35 | 75.09±0.71 | 75.45±0.51 | 75.70±0.58 | 75.90±0.49 | **76.44±0.49** | 75.85±0.51 | 73.55±0.35 | 72.89±0.43 | 73.49±0.41 | 72.91±0.39 | 51.98±0.27 |
| nq(opt) | 62.39±0.56 | 68.27±0.71 | 67.13±0.66 | 72.83±0.51 | 71.33±0.60 | 76.90±0.42 | 80.73±0.35 | **81.34±0.43** | 77.47±0.35 | 77.80±0.90 | 79.43±0.35 | 80.46±0.21 | 46.96±0.56 |
| nq(gpt) | 62.24±0.52 | 64.77±0.71 | **68.97±0.66** | **68.65±0.45** | **68.64±0.61** | 68.11±0.72 | 68.19±0.76 | 67.16±0.63 | 65.46±0.76 | 63.21±0.92 | 63.48±0.75 | – | – |

## C.2 UNCERTAINTY + INDIVIDUAL ACCURACY

We present the result of using uncertainty to predict individual accuracy in Table 10. This is similar to the practice of Kuhn et al. (2023); Malinin & Gales (2021) Formally, this setting is $U(x)$ + Individual Accuracy: We use the uncertainty estimated on $m$ samples for each of the $m$ samples to predict each answer's accuracy. For AUROC, this means

$$AUROC_{\text{U+IA}} = \sum_{j=1}^{m} AUROC([-U(x_1), \dots, -U(x_N)], [acc_{1,j}, \dots, acc_{N,j}]). \qquad (12)$$

## C.3 EFFECTS OF SAMPLING TEMPERATURE OF LLM

In the main text, we use the default generation config from `hugginface` or OpenAI's API for the LLMs. In Tables 7 to 9, the temperature is 1 for all models except for LLaMA2 (which uses 0.6) and `top_p` is 1 for all models except for LLaMA2 (which uses 0.9). Being sampling-based uncertainty quantification methods, our proposed measures obviously are affected by the temperature of the base LLM, and as we noted in the main paper, when very low temperature we do not expect such sampling-based black-box methods to work at all. Thus, in Tables 11 to 13, we lower the temperature to 0.5 and observe the effects.

Lower temperature leads to a less divergent posterior, which makes it more difficult for sampling based methods to get an estimate of uncertainty or confidence. As a result, compared with results at higher temperature, we observe that in general, the white-box method (SE) performs better than our black-box methods when $m$ is small. However, as $m$ increases ($\geq 10$), our methods' performance picks up.

## C.4 GENERATION WITHOUT REJECTION

Previous results indicate the practical performance of using the proposed $U$ and $C$ for selective generation, where some hard questions are ignored. In Table 14 we simply pick the most confident response. Even without rejection, most confidence measures can pick better responses and significantly improve the accuracy.

Table 10: AUROC, $U(x)$ + Individual Accuracy. Compared with Table 9, the AUROC for confidence measures (Ecc and Deg) are typically lower, which is consistent to the belief that confidence measures captures the quality of each responses.

| | NumSet | LexiSim | EigV(C) | Ecc(C) | Deg(C) | EigV(E) | Ecc(E) | Deg(E) | EigV(J) | Ecc(J) | Deg(J) | SE | P(true) |
|---|---|---|---|---|---|---|---|---|---|---|---|---|---|
| **m = 3** | | | | | | | | | | | | | |
| trivia(llama) | 82.58±0.13 | 80.78±0.12 | 86.44±0.12 | 86.86±0.11 | 86.38±0.25 | **87.36±0.17** | 86.36±0.17 | **87.34±0.17** | 84.40±0.12 | 82.58±0.22 | 84.30±0.11 | 77.96±0.14 | 56.23±0.09 |
| trivia(llama2) | 79.25±0.15 | 80.41±0.21 | **82.52±0.43** | 82.50±0.23 | 82.55±0.39 | 82.45±0.17 | 82.08±0.18 | **82.35±0.22** | 80.84±0.21 | 80.18±0.50 | 80.84±0.21 | 88.43±0.08 | 64.83±0.19 |
| trivia(opt) | 78.24±0.09 | 73.52±0.14 | 80.50±0.07 | 82.83±0.16 | 80.89±0.07 | 83.40±0.10 | 82.87±0.22 | **83.52±0.10** | 80.70±0.06 | 79.57±0.32 | 80.79±0.06 | **87.51±0.07** | 40.21±0.09 |
| trivia(gpt) | 66.67±0.21 | 71.93±0.30 | 71.76±1.33 | 71.67±1.36 | 71.60±1.57 | **73.04±0.32** | 72.26±0.57 | **72.95±0.33** | 71.63±0.29 | 70.59±0.34 | 71.58±0.29 | – | – |
| coqa(llama) | 62.67±0.12 | 73.85±0.23 | 76.61±0.40 | 76.46±0.39 | 76.84±0.17 | **79.17±0.23** | 76.93±0.72 | **79.14±0.21** | 75.84±0.15 | 74.33±0.35 | 75.93±0.14 | 75.23±0.24 | 52.78±0.16 |
| coqa(llama2) | 59.63±0.22 | 70.89±0.26 | **72.47±0.24** | 71.94±0.34 | **72.51±0.29** | **72.76±0.40** | 71.74±0.96 | **72.75±0.38** | 70.68±0.22 | 69.92±0.24 | 70.66±0.22 | 72.10±0.30 | 51.17±0.31 |
| coqa(opt) | 63.48±0.18 | 71.84±0.17 | 73.30±0.41 | 74.35±0.30 | 73.48±0.43 | **76.84±0.26** | 75.52±0.27 | **76.91±0.18** | 73.73±0.13 | 72.15±0.52 | 73.68±0.13 | 73.62±0.15 | 45.60±0.18 |
| coqa(gpt) | 51.46±0.10 | 64.37±0.16 | **67.41±0.26** | 64.43±0.24 | **67.38±0.46** | **67.42±0.53** | 66.22±0.56 | **67.40±0.52** | 64.86±0.20 | 66.33±0.45 | 65.19±0.20 | – | – |
| nq(llama) | 61.30±0.33 | 70.48±0.40 | 66.22±0.60 | 69.69±0.71 | 66.50±0.59 | **73.16±0.66** | 70.47±0.53 | **73.28±0.63** | 72.44±0.40 | 69.43±0.94 | 72.45±0.41 | 68.84±0.44 | 53.46±0.53 |
| nq(llama2) | 67.70±0.43 | 70.69±0.55 | 72.14±0.56 | 72.08±0.50 | 72.27±0.63 | **72.92±0.29** | 71.80±0.32 | **72.74±0.46** | 70.86±0.48 | 69.15±0.45 | 70.76±0.44 | 71.15±0.36 | 51.68±0.37 |
| nq(opt) | 61.18±0.52 | 61.07±0.63 | 64.25±0.73 | 69.53±0.66 | 64.58±0.81 | **72.07±0.42** | 70.54±0.79 | **72.14±0.43** | 68.40±0.75 | 66.83±0.55 | 68.44±0.75 | **81.01±0.44** | 46.56±0.74 |
| nq(gpt) | 60.05±0.52 | 62.72±0.75 | **65.27±0.89** | 65.23±0.82 | **65.36±0.77** | 64.82±0.70 | 63.86±0.82 | **64.67±0.76** | 62.66±0.73 | 60.95±0.89 | 62.44±0.72 | – | – |
| **m = 5** | | | | | | | | | | | | | |
| trivia(llama) | 82.94±0.09 | 77.82±0.13 | 87.03±0.10 | 87.38±0.13 | 87.19±0.11 | 87.77±0.12 | 87.32±0.12 | **87.98±0.12** | 85.66±0.08 | 84.51±0.08 | 85.56±0.08 | 77.64±0.10 | 55.14±0.06 |
| trivia(llama2) | 83.12±0.13 | 82.53±0.16 | **86.39±0.19** | 85.98±0.26 | **86.33±0.20** | **86.23±0.17** | 85.88±0.33 | 86.06±0.22 | 84.52±0.17 | 84.13±0.15 | 84.42±0.17 | **89.00±0.08** | 64.74±0.18 |
| trivia(opt) | 79.46±0.11 | 71.23±0.22 | 82.05±0.10 | 85.11±0.10 | 82.56±0.10 | 85.38±0.20 | 85.69±0.17 | **85.99±0.24** | 84.11±0.07 | 83.67±0.14 | 84.35±0.07 | **87.27±0.05** | 40.33±0.10 |
| trivia(gpt) | 69.99±0.22 | 74.81±0.19 | 75.97±0.66 | 75.32±0.80 | 75.92±0.75 | **76.34±0.24** | 75.49±1.23 | **76.19±0.29** | 75.04±0.21 | 74.32±0.33 | 74.64±0.21 | – | – |
| coqa(llama) | 62.60±0.09 | 72.42±0.15 | 76.55±0.27 | 76.03±0.10 | 76.80±0.23 | 79.08±0.17 | 77.91±0.22 | **79.28±0.11** | 76.31±0.11 | 75.54±0.24 | 76.63±0.12 | 74.61±0.19 | 52.29±0.16 |
| coqa(llama2) | 62.01±0.26 | 73.23±0.29 | 75.33±0.29 | 75.36±0.25 | 74.98±1.32 | **75.85±0.53** | 75.31±0.81 | **75.87±0.26** | 73.37±0.25 | 72.55±0.40 | 73.24±0.26 | 72.71±0.33 | 51.27±0.29 |
| coqa(opt) | 63.87±0.11 | 70.68±0.16 | 73.79±0.23 | 74.05±0.17 | 74.20±0.22 | 76.89±0.20 | 75.97±0.12 | **77.08±0.14** | 74.56±0.10 | 73.71±0.17 | 74.65±0.11 | 73.23±0.15 | 45.95±0.12 |
| coqa(gpt) | 51.81±0.10 | 64.88±0.15 | **70.23±0.24** | 67.19±0.65 | 69.93±0.63 | 69.69±0.64 | 68.61±0.38 | 69.73±0.37 | 66.89±0.20 | 68.56±0.71 | 67.64±0.20 | – | – |
| nq(llama) | 62.01±0.37 | 72.21±0.37 | 67.69±0.42 | 69.83±0.37 | 68.25±0.42 | **74.88±0.40** | 73.61±0.68 | **75.20±0.43** | 74.49±0.47 | 72.45±0.45 | 74.33±0.44 | 69.09±0.28 | 53.06±0.57 |
| nq(llama2) | 69.12±0.38 | 71.50±0.52 | 73.21±0.70 | 73.27±0.57 | 73.26±0.64 | **74.72±0.49** | 73.85±0.43 | **74.51±0.48** | 72.84±0.46 | 71.71±0.49 | 72.36±0.50 | 71.95±0.44 | 51.86±0.32 |
| nq(opt) | 62.68±0.48 | 64.64±0.93 | 65.70±0.60 | 70.08±0.59 | 66.28±0.61 | **75.66±0.46** | 75.73±0.64 | **75.98±0.47** | 74.84±0.47 | 74.08±0.57 | 75.08±0.49 | **81.25±0.33** | 46.74±0.57 |
| nq(gpt) | 61.36±0.39 | 63.80±0.82 | **67.56±0.60** | 67.33±0.80 | 67.52±0.76 | 66.55±0.72 | 65.79±0.88 | 66.04±0.74 | 64.19±0.88 | 62.67±0.69 | 63.52±0.84 | – | – |
| **m = 10** | | | | | | | | | | | | | |
| trivia(llama) | 81.26±0.15 | 75.14±0.07 | 86.66±0.13 | 86.75±0.06 | 86.80±0.13 | 87.07±0.08 | 86.71±0.08 | **87.59±0.12** | 85.40±0.07 | 84.61±0.07 | 85.52±0.07 | 77.37±0.10 | 54.54±0.05 |
| trivia(llama2) | 85.16±0.11 | 82.79±0.11 | **88.43±0.14** | 88.26±0.14 | **88.53±0.22** | 88.31±0.22 | 88.13±0.15 | 88.10±0.19 | 86.63±0.09 | 86.06±0.59 | 86.15±0.10 | 89.46±0.08 | 64.85±0.17 |
| trivia(opt) | 77.90±0.14 | 70.51±0.26 | 82.63±0.11 | 85.58±0.09 | 83.26±0.11 | 85.87±0.13 | 86.66±0.13 | **86.90±0.08** | 85.16±0.08 | 84.94±0.19 | 85.76±0.07 | 86.72±0.07 | 40.20±0.10 |
| trivia(gpt) | 72.69±0.25 | 77.40±0.18 | 79.16±0.62 | 78.93±0.61 | 78.92±0.67 | **79.81±0.42** | **79.60±0.56** | 78.96±0.79 | 77.95±0.19 | 77.07±0.35 | 77.17±0.20 | – | – |
| coqa(llama) | 61.20±0.12 | 71.19±0.17 | 76.63±0.11 | 74.56±0.16 | 76.84±0.26 | 78.25±0.12 | 77.18±0.13 | **78.54±0.13** | 75.46±0.12 | 75.40±0.30 | 76.34±0.12 | 73.97±0.16 | 51.73±0.15 |
| coqa(llama2) | 62.97±0.26 | 73.48±0.28 | 76.44±0.31 | 76.31±0.44 | 76.22±0.85 | **77.33±0.27** | 76.81±0.33 | 77.02±0.35 | 74.19±0.26 | 73.44±0.26 | 74.04±0.26 | 72.85±0.31 | 51.08±0.29 |
| coqa(opt) | 62.65±0.10 | 71.33±0.10 | 74.55±0.11 | 72.94±0.12 | 74.86±0.24 | 76.39±0.17 | 75.59±0.14 | **76.94±0.12** | 74.25±0.09 | 73.92±0.08 | 74.65±0.11 | 72.70±0.13 | 46.02±0.14 |
| coqa(gpt) | 52.24±0.12 | 63.89±0.22 | **71.46±0.54** | 68.22±0.21 | **71.54±0.52** | **71.22±0.32** | 70.11±0.41 | 70.71±0.41 | 67.61±0.24 | 69.46±1.14 | 68.83±0.23 | – | – |
| nq(llama) | 61.47±0.31 | 74.24±0.38 | 68.22±0.40 | 69.86±0.38 | 68.68±0.43 | 74.76±0.40 | 74.55±0.45 | **75.39±0.40** | **75.27±0.38** | 73.63±0.40 | **75.09±0.34** | 69.43±0.28 | 51.37±0.50 |
| nq(llama2) | 70.64±0.37 | 72.44±0.35 | 75.09±0.71 | 74.94±0.36 | 75.05±0.72 | **75.90±0.49** | **75.51±0.41** | **75.61±0.44** | 73.55±0.35 | 72.55±0.31 | 73.22±0.38 | 72.91±0.39 | 51.70±0.31 |
| nq(opt) | 62.39±0.56 | 68.27±0.71 | 67.13±0.66 | 71.74±0.56 | 67.33±0.68 | 76.90±0.42 | **78.71±0.39** | 78.71±0.36 | 77.47±0.35 | 77.16±0.50 | 78.00±0.32 | 80.46±0.21 | 45.65±0.45 |
| nq(gpt) | 62.24±0.52 | 64.77±0.71 | **68.97±0.66** | **68.81±0.56** | **68.93±0.64** | 68.11±0.72 | 67.89±0.70 | 67.11±0.70 | 65.46±0.76 | 64.40±0.81 | 64.27±0.77 | – | – |
| **m = 20** | | | | | | | | | | | | | |
| trivia(llama) | 78.79±0.13 | 75.92±0.04 | 86.29±0.07 | 85.59±0.08 | 86.43±0.07 | 86.47±0.14 | 85.86±0.08 | **87.09±0.07** | 84.58±0.08 | 84.15±0.10 | 85.19±0.07 | 76.61±0.10 | 54.43±0.06 |
| trivia(llama2) | 86.02±0.11 | 82.77±0.10 | **89.28±0.16** | 89.18±0.15 | **89.47±0.19** | 89.18±0.19 | 89.11±0.20 | 89.05±0.11 | 87.60±0.05 | 87.23±0.33 | 86.94±0.07 | 89.75±0.07 | 65.02±0.15 |
| trivia(opt) | 75.02±0.16 | 74.49±0.18 | 83.14±0.12 | 85.22±0.12 | 83.71±0.12 | 86.09±0.10 | 86.55±0.12 | **87.20±0.10** | 85.16±0.10 | 85.50±0.11 | 86.31±0.08 | 86.19±0.09 | 40.23±0.11 |
| trivia(gpt) | 74.69±0.24 | 78.18±0.17 | 80.90±0.50 | 80.74±0.73 | 81.24±0.24 | **81.78±0.25** | **81.75±0.30** | 80.73±0.46 | 79.37±0.18 | 78.64±0.22 | 78.15±0.18 | – | – |
| coqa(llama) | 59.34±0.14 | 71.09±0.17 | 76.78±0.11 | 72.97±0.13 | 76.89±0.20 | 77.65±0.09 | 76.48±0.21 | **78.17±0.15** | 74.43±0.12 | 75.04±0.11 | 76.19±0.12 | 73.32±0.16 | 51.69±0.19 |
| coqa(llama2) | 63.56±0.25 | 73.69±0.28 | 76.88±0.43 | 76.61±0.88 | 76.93±0.71 | **77.94±0.31** | 77.42±0.30 | **77.76±0.27** | 74.49±0.29 | 73.56±0.32 | 74.66±0.28 | 73.02±0.31 | 51.09±0.28 |
| coqa(opt) | 59.72±0.08 | 71.81±0.15 | 74.67±0.14 | 71.16±0.16 | 74.85±0.26 | 75.61±0.13 | 74.74±0.24 | **76.49±0.12** | 73.16±0.13 | 73.28±0.16 | 74.32±0.14 | 71.68±0.13 | 45.77±0.14 |
| coqa(gpt) | 52.44±0.08 | 63.39±0.19 | **72.26±0.27** | 70.26±0.30 | **71.99±0.55** | **72.17±0.17** | 71.39±0.36 | 71.53±0.32 | 68.18±0.21 | 69.93±0.83 | 69.46±0.22 | – | – |
| nq(llama) | 60.61±0.31 | 75.19±0.32 | 69.41±0.38 | 69.08±0.33 | 69.54±0.39 | 75.05±0.32 | 74.94±0.38 | **75.94±0.35** | 75.40±0.36 | 74.41±0.28 | 75.32±0.31 | 69.79±0.26 | 51.26±0.44 |
| nq(llama2) | 71.02±0.41 | 72.41±0.44 | 75.26±0.87 | 75.19±0.78 | 75.66±0.65 | **76.50±0.41** | **76.18±0.43** | **76.25±0.43** | 74.14±0.38 | 73.40±0.37 | 73.85±0.43 | 73.46±0.40 | 51.92±0.35 |
| nq(opt) | 60.45±0.47 | 70.66±0.69 | 68.01±0.58 | 70.49±0.52 | 67.43±0.57 | 77.03±0.32 | **79.65±0.34** | 77.85±0.29 | 77.55±0.45 | 77.59±0.51 | 78.50±0.38 | 79.16±0.26 | 45.22±0.50 |
| nq(gpt) | 62.88±0.51 | 65.32±0.62 | **69.99±0.66** | **69.75±0.51** | **69.86±0.62** | 69.05±0.63 | **69.57±0.66** | 67.43±0.68 | 66.40±0.69 | 65.01±0.53 | 64.40±0.72 | – | – |

## C.5 CONFIDENCE VS CALIBRATION

As suggested in Section 2, model calibration is an orthogonal research direction. Once the confidence measures are given, a calibration method could then be applied to calibrate the confidence scores into something close to the probability of correct answer. We applied the classical histogram binning method Zadrozny & Elkan (2001) on all methods, and compute the adaptive calibration error (ACE) Nixon et al. (2019). The number of bins is set to 15 following the standard practice Nixon et al. (2019), and the confidence measures are calibrated on the 1st generation of 1000 calibration samples and evaluated on the rest. Confidence measures are estimated using 20 generations. The results are in Table 16. After calibration, the confidence scores can faithfully reflect the accuracy. For example, for `Ecc (E)` on `trivia(llama)`, the gap between calibrated probability and the actual accuracy is only 0.026.

Table 11: AUARC, $U(x)$ + Expected Accuracy, with $m = 3, 5, 10, 20$ (similar to Table 1) and the temperature of the LLM set to 0.5. The best black-box methods are in **bold** and the best overall is underscored.

| | Random | Oracle | NumSet | LexiSim | EigV(C) | Ecc(C) | Deg(C) | EigV(E) | Ecc(E) | Deg(E) | EigV(J) | Ecc(J) | Deg(J) | SE | P(true) |
|---|---|---|---|---|---|---|---|---|---|---|---|---|---|---|---|
| | | Baselines | | | | | | | | | Ours | | | White-box | |
| **m = 3** | | | | | | | | | | | | | | | |
| trivia(llama) | 74.43±0.10 | 95.85±0.03 | 85.88±0.16 | 86.87±0.15 | **87.29±0.24** | **87.32±0.14** | **87.29±0.25** | 87.25±0.15 | 87.16±0.16 | 87.27±0.16 | 86.95±0.12 | 86.67±0.14 | 86.96±0.12 | 92.20±0.07 | 75.92±0.14 |
| trivia(llama2) | 75.92±0.11 | 96.35±0.03 | 86.96±0.16 | 87.56±0.27 | 88.21±0.28 | 88.17±0.32 | **88.29±0.20** | 88.17±0.18 | 87.98±0.36 | 88.14±0.24 | 87.77±0.24 | 87.55±0.24 | 87.82±0.21 | 93.02±0.05 | 82.07±0.10 |
| trivia(opt) | 40.53±0.13 | 75.08±0.13 | 58.77±0.23 | 59.40±0.45 | 60.98±0.30 | 61.06±0.48 | 60.98±0.29 | **61.26±0.35** | 61.02±0.47 | **61.33±0.41** | 60.25±0.32 | 60.04±0.43 | 60.35±0.37 | 67.52±0.18 | 35.56±0.20 |
| trivia(gpt) | 87.79±0.10 | 99.20±0.01 | 89.98±0.13 | **91.64±0.18** | 91.49±0.29 | 91.54±0.26 | 91.52±0.25 | 91.62±0.16 | 91.55±0.19 | 91.63±0.17 | **91.60±0.16** | 91.55±0.16 | **91.60±0.16** | – | – |
| coqa(llama) | 76.27±0.08 | 95.97±0.02 | 80.00±0.16 | 84.72±0.30 | **85.35±0.17** | 85.20±0.43 | **85.42±0.14** | 85.28±0.52 | 85.21±0.57 | 85.33±0.45 | 84.72±0.29 | 84.39±0.28 | 84.73±0.27 | 87.84±0.17 | 74.93±0.17 |
| coqa(llama2) | 78.36±0.12 | 96.77±0.03 | 81.86±0.13 | **85.78±0.18** | 85.68±0.77 | 85.97±0.34 | 85.70±0.69 | **86.09±0.27** | 85.90±0.31 | **86.10±0.32** | 85.67±0.28 | 85.31±0.31 | 85.76±0.18 | 88.45±0.16 | 79.65±0.18 |
| coqa(opt) | 70.40±0.15 | 93.70±0.07 | 75.31±0.13 | 78.94±0.22 | 79.11±0.77 | **79.41±0.63** | 79.23±0.70 | **79.58±0.25** | 79.34±0.36 | **79.52±0.32** | 78.92±0.34 | 78.68±0.35 | 79.02±0.28 | 82.35±0.20 | 68.42±0.26 |
| coqa(gpt) | 80.02±0.13 | 97.74±0.03 | 80.27±0.25 | 85.08±0.27 | **85.40±0.29** | 84.91±0.59 | **85.37±0.33** | 85.33±0.54 | 85.16±0.49 | 85.37±0.55 | 85.30±0.26 | 85.22±0.26 | **85.27±0.33** | – | – |
| nq(llama) | 40.72±0.56 | 73.91±0.51 | 50.88±0.66 | 53.82±0.66 | 54.67±0.80 | 54.74±0.89 | 54.84±0.45 | 55.00±0.81 | 54.30±0.65 | **54.89±0.79** | 53.71±0.71 | 52.61±0.91 | 53.42±0.65 | 57.22±0.68 | 46.68±0.67 |
| nq(llama2) | 44.01±0.64 | 77.45±0.60 | 55.33±0.51 | 56.62±0.80 | 57.28±0.88 | 57.05±0.80 | 57.08±0.85 | **57.39±0.92** | 56.80±0.85 | 57.26±1.09 | 56.65±0.85 | 56.11±0.98 | **56.95±0.95** | 60.31±0.88 | 44.58±0.54 |
| nq(opt) | 18.06±0.36 | 45.06±0.65 | 24.54±0.66 | 26.67±0.99 | 28.24±1.00 | 28.50±0.90 | 28.22±0.93 | **28.88±0.80** | 28.51±0.72 | **28.96±0.81** | 28.05±0.89 | 27.72±0.91 | 28.12±0.74 | 33.34±0.83 | 15.94±0.40 |
| nq(gpt) | 62.90±0.45 | 91.42±0.22 | 65.74±0.80 | 67.77±0.87 | 67.93±1.29 | 67.81±1.29 | 68.07±1.29 | **68.22±0.94** | 68.15±0.87 | 68.20±0.91 | 67.74±0.91 | 67.67±0.65 | 67.75±0.91 | – | – |
| **m = 5** | | | | | | | | | | | | | | | |
| trivia(llama) | 74.43±0.10 | 95.85±0.03 | 87.66±0.12 | 89.09±0.08 | **89.87±0.25** | **89.89±0.13** | **89.88±0.28** | 89.70±0.17 | 89.72±0.21 | 89.80±0.18 | 89.53±0.10 | 89.03±0.17 | 89.49±0.10 | 92.50±0.07 | 76.16±0.13 |
| trivia(llama2) | 75.92±0.11 | 96.35±0.03 | 89.38±0.24 | 90.15±0.26 | 90.08±0.22 | **90.21±0.16** | 90.14±0.18 | 90.08±0.19 | **90.18±0.15** | 89.91±0.22 | 89.47±0.25 | 89.85±0.31 | 93.28±0.06 | 82.20±0.10 |
| trivia(opt) | 40.55±0.13 | 75.10±0.13 | 60.52±0.24 | 60.31±0.30 | **63.51±0.55** | **63.41±0.37** | 63.13±0.47 | **63.58±0.35** | 63.56±0.26 | 63.49±0.28 | 62.45±0.23 | 62.26±0.14 | 62.87±0.20 | 67.92±0.18 | 35.62±0.20 |
| trivia(gpt) | 87.79±0.10 | 99.20±0.01 | 90.49±0.14 | **92.07±0.17** | 92.15±0.19 | **92.21±0.18** | 92.11±0.29 | 92.13±0.20 | **92.13±0.18** | 92.11±0.21 | 92.07±0.22 | – | – | | |
| coqa(llama) | 76.27±0.08 | 95.97±0.02 | 80.51±0.15 | 85.93±0.24 | **86.80±0.23** | 86.64±0.27 | **86.94±0.10** | 86.96±0.38 | 86.75±0.39 | **86.99±0.13** | 86.10±0.17 | 85.83±0.18 | 86.28±0.23 | 88.00±0.16 | 74.93±0.18 |
| coqa(llama2) | 78.36±0.12 | 96.77±0.03 | 82.54±0.16 | 86.79±0.25 | **87.56±0.19** | 87.40±0.27 | 87.60±0.23 | **87.60±0.17** | 87.58±0.21 | 87.59±0.12 | 87.11±0.22 | 86.74±0.31 | 87.07±0.17 | 88.67±0.16 | 79.69±0.18 |
| coqa(opt) | 70.40±0.15 | 93.70±0.07 | 76.43±0.22 | 80.25±0.26 | 80.86±0.79 | 81.00±0.50 | 80.74±0.41 | 81.40±0.33 | **81.19±0.46** | 81.40±0.33 | 80.55±0.20 | 80.19±0.24 | 80.59±0.26 | 82.70±0.20 | 68.41±0.26 |
| coqa(gpt) | 80.02±0.13 | 97.74±0.03 | 80.27±0.25 | 85.94±0.16 | **86.19±0.49** | 85.90±0.48 | **86.19±0.28** | 86.28±0.28 | **86.21±0.51** | 86.14±0.37 | **86.03±0.23** | 85.94±0.21 | **86.11±0.23** | – | – |
| nq(llama) | 40.71±0.56 | 73.91±0.51 | 52.88±0.80 | 55.59±0.71 | **56.73±0.74** | 56.88±0.53 | 56.60±0.65 | **57.21±0.65** | 57.09±0.82 | **57.25±0.70** | 55.93±0.65 | 54.95±0.64 | 55.63±0.74 | 57.91±0.66 | 46.86±0.60 |
| nq(llama2) | 44.01±0.64 | 77.45±0.60 | 55.36±0.62 | 58.71±0.91 | 59.43±0.92 | 59.46±0.80 | 59.56±0.76 | 59.72±0.66 | 59.42±0.92 | 59.68±1.18 | 58.93±0.83 | 58.25±0.87 | 58.75±0.98 | 60.94±0.88 | 44.62±0.52 |
| nq(opt) | 18.08±0.36 | 45.11±0.65 | 26.13±0.78 | 26.94±0.96 | 30.56±0.93 | 30.55±0.84 | 31.21±0.79 | **31.03±1.00** | 31.00±0.81 | 30.29±0.95 | 30.14±0.75 | 33.89±0.83 | 16.11±0.38 | | |
| nq(gpt) | 62.90±0.45 | 91.42±0.22 | 66.49±0.86 | **68.62±0.93** | 69.49±1.00 | 69.34±0.85 | 69.46±0.85 | **69.47±0.58** | 68.94±0.98 | 69.26±0.86 | 68.78±0.86 | 68.27±0.78 | **68.71±1.12** | – | – |
| **m = 10** | | | | | | | | | | | | | | | |
| trivia(llama) | 74.43±0.10 | 95.85±0.03 | 89.54±0.08 | 89.84±0.13 | **91.31±0.19** | 91.23±0.16 | **91.42±0.12** | **91.39±0.06** | 91.27±0.13 | **91.35±0.09** | 90.86±0.11 | 90.67±0.26 | 90.76±0.09 | 92.76±0.06 | 76.18±0.14 |
| trivia(llama2) | 75.92±0.11 | 96.35±0.03 | 90.76±0.06 | 90.71±0.08 | 92.27±0.19 | 92.19±0.21 | **92.21±0.18** | 92.17±0.21 | 92.13±0.14 | 92.12±0.10 | 91.80±0.09 | 91.70±0.11 | 91.65±0.11 | 93.60±0.06 | 82.30±0.10 |
| trivia(opt) | 40.59±0.13 | 75.14±0.13 | 62.12±0.32 | 60.95±0.27 | 65.75±0.40 | 66.01±0.25 | 65.80±0.35 | 65.87±0.26 | **66.14±0.26** | **66.21±0.27** | 64.90±0.20 | 64.97±0.29 | 65.23±0.26 | 68.27±0.18 | 35.65±0.19 |
| trivia(gpt) | 87.79±0.10 | 99.20±0.01 | 91.17±0.13 | **92.75±0.24** | 92.73±0.19 | 92.69±0.28 | **92.73±0.33** | 92.81±0.20 | 92.83±0.18 | **92.78±0.22** | **92.72±0.27** | 92.67±0.26 | **92.67±0.20** | – | – |
| coqa(llama) | 76.27±0.08 | 95.97±0.02 | 80.98±0.16 | 87.32±0.20 | 88.16±0.25 | **88.37±0.30** | 88.12±0.65 | **88.54±0.27** | **88.30±0.28** | **88.51±0.24** | 87.76±0.17 | 87.48±0.19 | 87.92±0.18 | 88.27±0.16 | 75.00±0.18 |
| coqa(llama2) | 78.36±0.12 | 96.77±0.03 | 83.20±0.14 | 87.85±0.21 | **88.70±0.15** | 88.45±0.40 | 88.58±0.19 | **88.78±0.22** | 88.57±0.20 | **88.79±0.19** | 88.08±0.21 | 87.91±0.21 | 88.12±0.24 | 88.86±0.16 | 79.70±0.18 |
| coqa(opt) | 70.40±0.15 | 93.70±0.07 | 78.18±0.14 | 81.97±0.30 | 82.59±0.58 | **82.94±0.38** | 82.58±0.44 | **83.36±0.45** | 83.09±0.40 | **83.35±0.29** | 82.30±0.19 | 81.77±0.30 | 82.17±0.28 | 83.23±0.19 | 68.43±0.27 |
| coqa(gpt) | 80.02±0.13 | 97.74±0.03 | 80.44±0.24 | 86.34±0.24 | **87.29±0.39** | 87.00±0.42 | **87.47±0.21** | 87.36±0.26 | 87.23±0.34 | 87.25±0.24 | 86.82±0.41 | 87.16±0.36 | 86.90±0.32 | – | – |
| nq(llama) | 40.72±0.56 | 73.92±0.51 | 53.73±0.66 | 57.24±0.73 | 58.49±0.95 | **58.96±0.96** | 58.64±0.74 | **59.35±0.84** | 59.31±0.88 | **58.88±0.75** | 57.70±0.75 | 56.94±0.84 | 57.26±0.75 | 58.71±0.70 | 46.81±0.64 |
| nq(llama2) | 44.01±0.64 | 77.45±0.60 | 56.35±0.83 | 60.30±0.89 | **61.55±1.16** | 61.42±1.18 | 61.44±1.04 | **61.88±0.97** | 61.77±1.14 | **61.54±1.06** | 60.96±1.15 | 60.33±1.03 | 60.56±1.04 | 61.55±0.94 | 44.71±0.55 |
| nq(opt) | 18.14±0.36 | 45.21±0.66 | 26.53±0.67 | 27.49±0.57 | 32.09±0.83 | 32.32±0.87 | 32.30±0.73 | 33.12±0.94 | 33.07±0.88 | **32.97±0.73** | 31.74±0.90 | **32.40±0.89** | **32.25±0.78** | 34.13±0.79 | 16.22±0.39 |
| nq(gpt) | 62.90±0.45 | 91.42±0.22 | 66.44±0.94 | 69.00±0.95 | **70.34±1.01** | 69.98±1.11 | 70.30±1.05 | 70.20±0.98 | **69.97±0.60** | 70.11±0.64 | 69.40±0.89 | 69.20±0.83 | 69.04±0.91 | – | – |
| **m = 20** | | | | | | | | | | | | | | | |
| trivia(llama) | 74.43±0.10 | 95.85±0.03 | 90.83±0.11 | 90.27±0.08 | 92.46±0.09 | 92.45±0.08 | **92.60±0.12** | 92.41±0.13 | 92.37±0.11 | 92.42±0.11 | 91.85±0.09 | 91.76±0.08 | 91.71±0.10 | 92.98±0.06 | 76.13±0.15 |
| trivia(llama2) | 75.92±0.11 | 96.35±0.03 | 91.77±0.15 | 91.21±0.09 | **93.23±0.18** | 93.14±0.21 | **93.28±0.15** | **93.23±0.15** | 93.21±0.10 | **93.21±0.12** | 92.79±0.11 | 92.67±0.11 | 92.59±0.08 | 93.80±0.06 | 82.31±0.09 |
| trivia(opt) | 40.65±0.13 | 75.21±0.13 | 62.91±0.26 | 61.07±0.29 | 67.61±0.25 | 67.58±0.19 | 67.49±0.28 | 67.66±0.30 | 67.64±0.22 | **67.88±0.20** | 66.72±0.26 | 66.62±0.21 | 66.73±0.25 | 68.58±0.18 | 35.75±0.20 |
| trivia(gpt) | 87.79±0.10 | 99.20±0.01 | 91.61±0.15 | 93.02±0.24 | 92.87±0.24 | 93.04±0.32 | 93.02±0.31 | **93.23±0.25** | **93.31±0.13** | **93.25±0.22** | 93.08±0.27 | 92.95±0.19 | 93.02±0.27 | – | – |
| coqa(llama) | 76.27±0.08 | 95.97±0.02 | 81.17±0.17 | 87.57±0.19 | 88.68±0.20 | 88.82±0.27 | 88.71±0.44 | **89.18±0.24** | 88.94±0.22 | **89.25±0.10** | 88.19±0.22 | 88.02±0.36 | 88.34±0.21 | 88.44±0.16 | 75.03±0.18 |
| coqa(llama2) | 78.36±0.12 | 96.77±0.03 | 83.34±0.12 | 88.40±0.12 | 89.42±0.21 | 89.28±0.51 | 89.44±0.29 | **89.83±0.27** | **89.62±0.30** | **89.80±0.30** | 88.81±0.29 | 88.65±0.23 | 88.87±0.21 | 89.07±0.16 | 79.72±0.19 |
| coqa(opt) | 70.40±0.15 | 93.70±0.07 | 78.67±0.15 | 82.57±0.23 | 83.75±0.37 | 83.67±0.13 | 83.94±0.28 | **84.23±0.42** | **84.26±0.25** | **84.44±0.40** | 83.10±0.29 | 82.61±0.24 | 82.97±0.26 | 83.53±0.18 | 68.44±0.27 |
| coqa(gpt) | 80.02±0.13 | 97.74±0.03 | 80.38±0.22 | 86.74±0.27 | **87.91±0.17** | 87.47±0.57 | **88.04±0.29** | 87.83±0.31 | **87.85±0.25** | 87.66±0.28 | 87.36±0.32 | 87.39±0.35 | 87.47±0.25 | – | – |
| nq(llama) | 40.72±0.56 | 73.92±0.51 | 53.73±0.73 | 58.29±0.64 | **60.11±0.97** | 60.04±0.91 | 60.11±0.91 | 60.84±0.80 | 60.51±0.93 | **60.26±0.76** | 58.97±0.68 | 58.21±0.72 | 58.23±0.76 | 59.23±0.77 | 46.84±0.64 |
| nq(llama2) | 44.01±0.64 | 77.45±0.60 | 56.41±0.84 | 61.19±0.80 | **62.80±0.98** | 62.35±0.83 | **62.59±1.08** | **63.24±0.84** | 62.76±0.95 | **62.91±0.88** | 62.12±0.95 | 61.53±0.87 | 61.49±1.02 | 61.96±0.95 | 44.69±0.55 |
| nq(opt) | 18.18±0.37 | 45.27±0.67 | 26.30±0.70 | 27.06±0.53 | 33.24±0.80 | 32.87±0.79 | 33.07±0.78 | **34.06±0.83** | 34.07±0.79 | **33.78±0.86** | 32.40±0.77 | 33.24±0.82 | 33.06±0.81 | 34.29±0.79 | 16.71±0.43 |
| nq(gpt) | 62.90±0.45 | 91.42±0.22 | 66.61±0.74 | 69.89±0.89 | **71.11±0.86** | 71.45±0.81 | 71.33±0.87 | 71.39±0.77 | **71.43±0.78** | 70.67±0.74 | 70.70±0.83 | 70.21±0.87 | 69.97±0.68 | – | – |

Table 12: AUARC, $C(x, \mathbf{s})$ + Individual Accuracy, with $m = 3, 5, 10, 20$ (similar to Table 2) and the temperature of the LLM set to 0.5. The best black-box methods are in **bold** and the best overall is underscored.

| | Random | Oracle | NumSet | LexiSim | EigV(C) | Ecc(C) | Deg(C) | EigV(E) | Ecc(E) | Deg(E) | EigV(J) | Ecc(J) | Deg(J) | SE | P(true) |
|---|---|---|---|---|---|---|---|---|---|---|---|---|---|---|---|
| | | Baselines | | | | | | | | | Ours | | | White-box | |
| **m = 3** | | | | | | | | | | | | | | | |
| trivia(llama) | 74.25±0.09 | 96.36±0.03 | 86.97±0.15 | 87.96±0.16 | **88.39±0.22** | **88.50±0.12** | **88.51±0.15** | 88.37±0.14 | 88.36±0.14 | 88.34±0.22 | 88.01±0.13 | 87.76±0.13 | 88.01±0.12 | 92.52±0.07 | 75.79±0.15 |
| trivia(llama2) | 75.95±0.10 | 96.84±0.03 | 88.05±0.16 | 88.81±0.30 | **89.51±0.32** | **89.41±0.33** | **89.34±0.36** | 89.44±0.26 | 89.41±0.23 | 88.14±0.24 | 89.00±0.28 | 88.75±0.30 | 89.04±0.19 | 93.37±0.05 | 82.07±0.10 |
| trivia(opt) | 40.49±0.14 | 77.10±0.12 | 60.61±0.23 | 61.19±0.48 | 62.89±0.35 | 63.10±0.43 | 62.97±0.29 | **63.31±0.23** | 63.15±0.43 | 62.92±0.40 | 62.02±0.35 | 61.77±0.40 | 62.20±0.39 | 68.35±0.18 | 35.51±0.20 |
| trivia(gpt) | 87.78±0.10 | 99.22±0.01 | 90.07±0.13 | **91.59±0.16** | 91.56±0.26 | 91.60±0.18 | 91.52±0.34 | **91.70±0.15** | 91.66±0.17 | 91.65±0.14 | **91.64±0.17** | 91.58±0.30 | **91.64±0.17** | – | – |
| coqa(llama) | 76.27±0.09 | 96.49±0.03 | 80.24±0.18 | 85.89±0.35 | **86.52±0.15** | 86.47±0.27 | 86.62±0.22 | **86.53±0.48** | 86.39±0.56 | 86.29±0.62 | 85.83±0.31 | 85.46±0.30 | 85.85±0.27 | 88.32±0.17 | 74.80±0.18 |
| coqa(llama2) | 78.31±0.16 | 97.46±0.04 | 82.55±0.13 | 86.86±0.20 | 86.71±0.78 | **86.93±0.27** | 87.09±0.18 | **87.17±0.26** | 87.07±0.23 | **87.03±0.18** | 86.67±0.33 | 86.56±0.26 | 86.73±0.18 | 88.86±0.16 | 79.15±0.25 |
| coqa(opt) | 70.31±0.21 | 95.08±0.07 | 76.03±0.18 | 80.12±0.25 | 80.25±0.82 | **80.60±0.56** | 80.52±0.29 | **80.85±0.30** | 80.79±0.26 | **80.37±0.32** | 80.08±0.37 | 79.63±0.30 | 80.25±0.26 | 82.81±0.22 | 68.11±0.29 |
| coqa(gpt) | 80.00±0.13 | 97.85±0.03 | 80.18±0.24 | 85.15±0.27 | **85.47±0.30** | 85.14±0.28 | **85.48±0.23** | 85.41±0.57 | 85.18±0.46 | 85.16±0.36 | **85.36±0.27** | 85.15±0.25 | **85.28±0.23** | – | – |
| nq(llama) | 41.01±0.63 | 77.54±0.56 | 52.59±0.78 | 55.50±0.79 | **56.57±0.71** | 56.66±0.76 | 56.79±0.83 | **56.97±0.88** | 56.43±0.76 | **56.63±0.65** | 55.37±0.76 | 54.04±0.80 | 55.15±0.69 | 58.19±0.71 | 47.48±0.62 |
| nq(llama2) | 43.84±0.58 | 79.98±0.48 | 53.89±0.47 | 57.71±0.84 | 58.27±0.92 | 58.24±0.76 | 58.55±0.84 | 58.44±0.95 | 58.16±0.92 | 58.41±0.94 | 57.69±0.92 | 57.16±0.84 | 57.75±1.01 | 60.56±0.86 | 44.59±0.46 |
| nq(opt) | 17.83±0.44 | 48.55±0.75 | 25.31±0.78 | 28.05±1.19 | 29.63±1.20 | 29.64±1.01 | 29.83±1.12 | 30.30±0.94 | 30.11±0.96 | 30.50±1.07 | 29.42±1.17 | 29.06±1.06 | 29.42±0.96 | 33.64±0.96 | 16.32±0.55 |
| nq(gpt) | 62.95±0.45 | 92.08±0.21 | 65.64±0.76 | 68.04±0.86 | 68.15±1.35 | 68.10±1.31 | **67.93±1.63** | 68.52±0.90 | 68.42±0.82 | 68.24±1.02 | 68.04±0.84 | 67.99±0.76 | 68.03±0.74 | – | – |
| **m = 5** | | | | | | | | | | | | | | | |
| trivia(llama) | 74.35±0.09 | 96.38±0.03 | 88.48±0.13 | 89.86±0.09 | **90.67±0.22** | **90.74±0.16** | **90.71±0.36** | 90.57±0.14 | 90.61±0.19 | 90.59±0.19 | 90.29±0.10 | 90.10±0.17 | 90.21±0.10 | 92.75±0.07 | 75.89±0.16 |
| trivia(llama2) | 75.97±0.09 | 96.85±0.03 | 89.55±0.15 | 90.22±0.26 | **91.00±0.25** | **91.14±0.15** | 91.07±0.32 | 90.95±0.17 | **91.02±0.14** | 91.03±0.20 | 90.77±0.25 | 90.59±0.19 | 90.65±0.20 | 93.51±0.05 | 81.92±0.11 |
| trivia(opt) | 40.56±0.15 | 77.16±0.13 | 61.58±0.25 | 61.31±0.32 | 64.65±0.56 | 64.80±0.26 | **64.92±0.21** | 64.80±0.35 | **65.09±0.24** | 65.03±0.23 | 63.49±0.21 | 63.63±0.25 | 64.06±0.18 | 68.41±0.19 | 35.66±0.20 |
| trivia(gpt) | 87.74±0.11 | 99.22±0.01 | 90.50±0.10 | **92.09±0.17** | 92.14±0.14 | 92.09±0.34 | 92.15±0.20 | 92.14±0.18 | **92.14±0.13** | 92.14±0.15 | 92.06±0.18 | 92.00±0.18 | **92.05±0.19** | – | – |
| coqa(llama) | 76.32±0.09 | 96.94±0.02 | 80.82±0.16 | 86.74±0.23 | 87.55±0.27 | 87.49±0.30 | **87.79±0.15** | **87.76±0.38** | 87.56±0.41 | 87.56±0.45 | 86.85±0.18 | 86.75±0.19 | 87.11±0.18 | 88.33±0.16 | 74.91±0.17 |
| coqa(llama2) | 78.26±0.15 | 97.44±0.04 | 82.92±0.17 | 87.46±0.29 | **88.27±0.21** | 88.17±0.17 | 88.28±0.19 | **88.30±0.16** | **88.28±0.30** | 88.00±0.28 | 87.76±0.27 | 87.43±0.25 | 87.73±0.22 | 88.91±0.16 | 79.06±0.22 |
| coqa(opt) | 70.31±0.17 | 95.08±0.06 | 76.95±0.19 | 81.01±0.27 | 81.70±0.78 | 81.76±0.58 | 82.02±0.26 | **82.32±0.42** | 82.54±0.36 | 81.80±0.26 | 81.39±0.18 | 81.34±0.23 | 81.50±0.22 | 83.00±0.19 | 68.16±0.26 |
| coqa(gpt) | 80.05±0.13 | 97.86±0.03 | 80.24±0.25 | 85.95±0.15 | **86.25±0.47** | 85.84±0.53 | **86.20±0.27** | 86.34±0.29 | 85.99±0.45 | **86.13±0.22** | 86.06±0.23 | 85.88±0.17 | **86.10±0.25** | – | – |
| nq(llama) | 40.83±0.60 | 77.39±0.54 | 53.76±0.90 | 56.77±0.71 | 57.91±0.77 | 57.86±0.75 | **58.50±0.69** | 58.51±0.90 | 58.55±0.73 | 58.90±0.65 | 57.11±0.71 | 56.61±0.65 | 56.94±0.64 | 58.56±0.64 | 47.27±0.59 |
| nq(llama2) | 43.95±0.61 | 80.07±0.50 | 56.00±0.63 | 59.60±0.98 | **60.33±0.93** | 60.46±0.83 | 60.76±0.71 | 60.62±0.74 | 60.93±0.85 | 60.71±0.78 | 59.80±0.88 | 59.35±0.93 | 59.60±0.95 | 61.14±0.84 | 44.83±0.50 |
| nq(opt) | 17.91±0.45 | 48.69±0.78 | 26.70±0.82 | 27.84±1.13 | **31.35±1.06** | 30.66±0.92 | 31.69±0.98 | **31.89±0.98** | 31.23±0.92 | **31.77±1.01** | 30.77±1.15 | 30.70±0.98 | 29.42±0.96 | 33.98±0.93 | 16.53±0.50 |
| nq(gpt) | 62.95±0.49 | 92.08±0.23 | 66.52±0.94 | **68.75±0.95** | 69.61±1.04 | 69.32±1.18 | 69.33±1.06 | 69.56±0.58 | 69.34±0.81 | 69.25±0.84 | 68.90±0.87 | 68.52±0.82 | **68.72±0.77** | – | – |
| **m = 10** | | | | | | | | | | | | | | | |
| trivia(llama) | 74.41±0.10 | 96.40±0.03 | 89.92±0.09 | 90.20±0.14 | 91.66±0.17 | **91.92±0.13** | 91.88±0.17 | 91.75±0.06 | **91.89±0.09** | 91.83±0.10 | 91.20±0.12 | 91.09±0.19 | 91.12±0.09 | 92.88±0.07 | 75.87±0.14 |
| trivia(llama2) | 75.93±0.10 | 96.84±0.03 | 90.98±0.08 | 90.93±0.08 | **92.52±0.21** | 92.66±0.24 | **92.69±0.27** | 92.46±0.21 | 92.67±0.13 | **92.64±0.09** | 92.08±0.09 | 92.02±0.09 | 91.99±0.18 | 93.70±0.05 | 81.96±0.10 |
| trivia(opt) | 40.64±0.13 | 77.23±0.12 | 62.51±0.30 | 61.33±0.26 | 66.06±0.44 | 67.02±0.22 | 66.88±0.39 | 66.29±0.20 | 67.09±0.25 | 67.04±0.22 | 65.61±0.20 | 66.00±0.21 | 66.33±0.26 | 68.44±0.18 | 35.75±0.17 |
| trivia(gpt) | 87.75±0.10 | 99.22±0.01 | 91.12±0.13 | **92.76±0.24** | 92.73±0.19 | **92.67±0.26** | 92.68±0.30 | 92.82±0.20 | **92.98±0.24** | 92.74±0.20 | **92.72±0.27** | 92.61±0.16 | 92.59±0.18 | – | – |
| coqa(llama) | 76.39±0.09 | 96.96±0.03 | 81.31±0.16 | 87.70±0.20 | 88.57±0.24 | 88.45±0.27 | **88.90±0.20** | **88.92±0.27** | 88.98±0.24 | **88.77±0.29** | 88.12±0.18 | 88.03±0.21 | 88.43±0.18 | 88.46±0.17 | 75.13±0.18 |
| coqa(llama2) | 78.30±0.13 | 97.45±0.03 | 83.32±0.14 | 88.10±0.22 | **88.97±0.15** | 88.77±0.21 | **88.97±0.18** | 89.05±0.23 | 89.03±0.17 | **88.69±0.22** | 88.34±0.22 | 88.14±0.19 | 88.45±0.22 | 88.93±0.17 | 79.18±0.19 |
| coqa(opt) | 70.33±0.15 | 95.08±0.05 | 78.46±0.16 | 82.30±0.29 | 82.96±0.57 | 83.32±0.47 | 83.46±0.31 | **83.76±0.46** | 83.95±0.41 | 83.36±0.31 | 82.66±0.19 | 82.51±0.30 | 82.91±0.17 | 83.37±0.19 | 68.16±0.26 |
| coqa(gpt) | 80.00±0.14 | 97.85±0.03 | 80.39±0.25 | 86.36±0.24 | **87.30±0.40** | 86.97±0.34 | **87.25±0.16** | 87.39±0.26 | 87.19±0.20 | 87.09±0.19 | 86.85±0.41 | 86.69±0.26 | 87.01±0.15 | – | – |
| nq(llama) | 40.85±0.57 | 77.40±0.51 | 54.32±0.78 | 57.84±0.77 | 59.04±0.99 | 59.11±0.86 | 59.65±0.91 | 60.01±0.85 | 60.51±0.72 | 60.06±0.63 | 58.54±0.70 | 58.72±0.71 | 59.06±0.69 | 61.68±0.95 | 47.21±0.64 |
| nq(llama2) | 43.90±0.62 | 80.19±0.51 | 56.66±0.80 | 60.71±0.92 | **61.85±1.05** | 62.03±0.98 | 62.39±1.07 | **62.27±0.98** | 62.68±0.98 | **62.22±1.10** | 61.33±1.18 | 60.94±0.98 | 61.09±1.05 | 61.84±0.95 | 45.12±0.49 |
| nq(opt) | 18.01±0.37 | 48.86±0.64 | 26.60±0.68 | 27.88±0.59 | 32.38±0.83 | 32.34±0.77 | **33.21±0.81** | **33.40±0.96** | 33.39±0.85 | **33.88±0.85** | 32.06±0.91 | 32.70±0.86 | 32.91±0.80 | 34.10±0.80 | 16.61±0.42 |
| nq(gpt) | 63.06±0.46 | 92.13±0.21 | 66.66±0.98 | 69.11±0.95 | **70.47±1.01** | 70.22±0.72 | 70.39±0.76 | 70.31±0.97 | 70.04±0.73 | 70.14±0.88 | **69.51±0.87** | 68.97±0.92 | 69.00±0.82 | – | – |
| **m = 20** | | | | | | | | | | | | | | | |
| trivia(llama) | 74.43±0.10 | 96.41±0.03 | 90.83±0.11 | 90.27±0.08 | 92.46±0.09 | **92.73±0.16** | **92.73±0.09** | 92.41±0.13 | **92.75±0.08** | 92.66±0.11 | 91.85±0.09 | 91.89±0.08 | 91.81±0.08 | 92.98±0.06 | 75.84±0.15 |
| trivia(llama2) | 75.92±0.11 | 96.83±0.03 | 91.77±0.15 | 91.21±0.09 | 93.23±0.18 | 93.35±0.27 | **93.53±0.08** | 93.43±0.13 | **93.51±0.10** | 93.40±0.09 | 92.79±0.11 | 92.64±0.08 | 92.55±0.08 | 93.80±0.06 | 81.96±0.10 |
| trivia(opt) | 40.65±0.13 | 77.24±0.12 | 62.91±0.26 | 61.07±0.29 | 67.61±0.25 | **68.42±0.18** | 68.29±0.17 | 67.66±0.30 | 68.45±0.19 | **68.51±0.20** | 66.23±0.20 | 66.27±0.26 | 66.69±0.21 | 68.58±0.18 | 35.80±0.18 |
| trivia(gpt) | 87.79±0.10 | 99.22±0.01 | 91.61±0.15 | 93.02±0.24 | 92.87±0.24 | 92.93±0.16 | **93.09±0.33** | **93.23±0.25** | **93.27±0.20** | **93.17±0.15** | 93.08±0.27 | 92.94±0.25 | 92.92±0.20 | – | – |
| coqa(llama) | 76.27±0.08 | 96.93±0.02 | 81.17±0.17 | 87.57±0.19 | 88.68±0.20 | 88.86±0.15 | **89.15±0.21** | **89.18±0.24** | 88.94±0.22 | **89.25±0.23** | 88.19±0.22 | 88.24±0.21 | 88.61±0.19 | 88.44±0.16 | 75.11±0.17 |
| coqa(llama2) | 78.36±0.12 | 97.47±0.03 | 83.34±0.12 | 88.40±0.12 | 89.42±0.21 | 89.51±0.23 | 89.65±0.16 | **89.83±0.27** | **89.97±0.30** | 89.41±0.20 | 88.81±0.29 | 88.63±0.17 | 88.99±0.17 | 89.07±0.16 | 79.37±0.18 |
| coqa(opt) | 70.40±0.15 | 95.11±0.05 | 78.67±0.15 | 82.57±0.23 | 83.75±0.37 | 83.98±0.19 | 84.22±0.23 | **84.23±0.42** | **84.97±0.26** | 84.23±0.21 | 83.05±0.23 | 83.05±0.24 | 83.23±0.20 | 83.53±0.18 | 68.19±0.27 |
| coqa(gpt) | 80.02±0.13 | 97.86±0.03 | 80.38±0.22 | 86.74±0.27 | **87.91±0.17** | 87.59±0.36 | **87.80±0.21** | 87.83±0.31 | **87.84±0.27** | 87.43±0.21 | 87.36±0.32 | 87.05±0.15 | 87.37±0.19 | – | – |
| nq(llama) | 40.72±0.56 | 77.29±0.50 | 53.73±0.73 | 58.29±0.64 | **60.11±0.97** | 60.41±0.86 | **60.84±0.80** | 60.84±0.80 | 61.40±0.83 | **60.87±0.75** | 58.97±0.68 | 58.29±0.70 | 58.72±0.71 | 59.23±0.77 | 46.98±0.64 |
| nq(llama2) | 44.01±0.64 | 80.11±0.53 | 56.41±0.84 | 61.19±0.80 | **62.80±0.98** | 62.87±0.86 | **63.33±0.92** | **63.24±0.84** | 63.49±0.87 | **63.26±0.96** | 62.12±0.95 | 61.54±0.90 | 61.81±0.93 | 61.96±0.95 | 45.02±0.53 |
| nq(opt) | 18.18±0.37 | 49.14±0.63 | 26.30±0.70 | 27.06±0.53 | 33.24±0.80 | 33.04±0.68 | 33.91±0.76 | **34.06±0.83** | **34.39±0.79** | **34.73±0.76** | 32.40±0.77 | 33.40±0.78 | 33.69±0.75 | 34.29±0.79 | 16.71±0.43 |
| nq(gpt) | 62.90±0.45 | 92.06±0.21 | 66.61±0.74 | 69.89±0.89 | **71.11±0.86** | 71.38±0.73 | 71.48±0.84 | 71.39±0.77 | **71.48±0.78** | 71.02±0.81 | 70.70±0.83 | 69.99±0.73 | 69.90±0.76 | – | – |

Table 13: AUROC, $C(x, \mathbf{s})$ + Individual Accuracy, with $m = 3, 5, 10, 20$ (similar to Table 9) and the temperature of the LLM set to 0.5. The best black-box methods are in **bold** and the best overall is underscored.

| | Baselines | | | | | Ours | | | | | | White-box | |
|---|---|---|---|---|---|---|---|---|---|---|---|---|---|
| | NumSet | LexiSim | EigV(C) | Ecc(C) | Deg(C) | EigV(E) | Ecc(E) | Deg(E) | EigV(J) | Ecc(J) | Deg(J) | SE | P(true) |
| **m = 3** | | | | | | | | | | | | | |
| trivia(llama) | 78.71±0.16 | 79.49±0.20 | 81.75±0.26 | **82.08±0.17** | 82.12±0.18 | 81.72±0.23 | 81.79±0.45 | 81.58±0.33 | 80.04±0.21 | 79.42±0.40 | 79.98±0.19 | 87.51±0.13 | 56.45±0.18 |
| trivia(llama2) | 78.62±0.17 | 79.59±0.20 | 81.84±0.38 | 81.77±0.40 | 81.82±0.53 | 81.94±0.21 | 81.83±0.20 | 81.47±0.55 | 80.29±0.20 | 79.38±0.87 | 80.16±0.19 | 88.15±0.10 | 64.42±0.17 |
| trivia(opt) | 78.24±0.13 | 76.29±0.13 | 80.62±0.19 | 80.94±0.12 | 80.93±0.36 | 81.05±0.13 | 80.93±0.32 | 80.81±0.35 | 78.79±0.08 | 78.38±0.34 | 79.04±0.08 | 85.22±0.12 | 45.94±0.17 |
| trivia(gpt) | 59.98±0.13 | 66.28±0.27 | 66.06±1.46 | 66.15±0.44 | 65.82±1.73 | 66.41±0.41 | 66.20±0.43 | 66.17±0.70 | 66.36±0.26 | 65.95±0.26 | 66.16±0.26 | – | – |
| coqa(llama) | 60.68±0.19 | 71.78±0.23 | 74.05±0.28 | 73.93±0.37 | 74.08±0.25 | 74.60±0.33 | 74.34±0.50 | 73.47±0.50 | 71.72±0.23 | 70.64±0.61 | 71.93±0.23 | 73.90±0.28 | 48.14±0.20 |
| coqa(llama2) | 59.83±0.19 | 70.20±0.19 | 71.38±1.53 | 71.38±0.70 | 72.06±0.27 | 72.13±0.38 | 71.60±1.00 | 71.46±0.45 | 69.96±0.23 | 69.43±0.18 | 70.26±0.22 | 72.50±0.27 | 51.12±0.26 |
| coqa(opt) | 62.34±0.15 | 68.75±0.23 | 69.99±1.10 | 70.71±0.54 | 70.79±0.25 | 71.40±0.42 | 71.24±0.71 | 70.29±0.46 | 69.22±0.25 | 68.75±0.53 | 69.51±0.23 | 70.34±0.19 | 46.62±0.33 |
| coqa(gpt) | 50.95±0.08 | 63.01±0.29 | 64.37±0.44 | 62.80±0.87 | 64.15±0.36 | 64.35±0.75 | 63.65±0.77 | 63.37±1.35 | 63.01±0.30 | 62.91±0.41 | 63.09±0.29 | – | – |
| nq(llama) | 68.36±0.44 | 69.74±0.30 | 72.63±0.46 | 72.75±0.16 | 72.98±0.38 | 73.42±0.43 | 72.67±0.29 | 72.11±0.43 | 70.26±0.32 | 67.99±0.65 | 69.87±0.34 | 71.05±0.35 | 58.86±0.61 |
| nq(llama2) | 67.25±0.23 | 69.53±0.42 | 71.63±0.34 | 71.50±0.31 | 72.03±0.34 | 72.25±0.37 | 71.55±0.48 | 71.67±0.45 | 70.16±0.49 | 68.59±0.69 | 69.97±0.51 | 70.50±0.43 | 52.71±0.32 |
| nq(opt) | 68.42±0.43 | 66.75±0.52 | 71.90±0.65 | 72.30±0.69 | 73.01±0.85 | 74.50±0.61 | 73.44±0.80 | 74.58±0.57 | 71.37±0.58 | 69.97±0.70 | 71.23±0.66 | 77.38±0.55 | 46.37±0.84 |
| nq(gpt) | 56.18±0.32 | 58.59±0.43 | 60.48±0.99 | 60.08±0.80 | 60.06±1.22 | 60.45±0.45 | 59.88±0.55 | 59.89±0.65 | 58.97±0.47 | 58.11±0.52 | 58.67±0.47 | – | – |
| **m = 5** | | | | | | | | | | | | | |
| trivia(llama) | 82.31±0.17 | 81.59±0.17 | 85.44±0.23 | 85.77±0.52 | 85.79±0.43 | 85.38±0.20 | 85.67±0.68 | 85.51±0.18 | 83.53±0.18 | 83.33±0.40 | 83.30±0.16 | 88.05±0.12 | 56.41±0.19 |
| trivia(llama2) | 82.06±0.13 | 81.28±0.18 | 85.31±0.56 | 85.80±0.30 | 85.73±0.76 | 85.44±0.22 | 85.63±0.21 | 85.55±0.32 | 83.66±0.19 | 83.25±0.84 | 83.39±0.18 | 88.57±0.08 | 64.23±0.16 |
| trivia(opt) | 80.45±0.12 | 75.03±0.16 | 83.08±0.39 | 83.78±0.24 | 83.88±0.19 | 83.48±0.11 | 84.08±0.29 | 84.06±0.14 | 81.22±0.07 | 81.20±0.27 | 81.95±0.07 | 85.28±0.11 | 46.17±0.17 |
| trivia(gpt) | 62.19±0.17 | 68.21±0.33 | 69.00±0.47 | 68.81±0.26 | 68.36±1.50 | 68.94±0.44 | 68.64±0.60 | 68.46±0.63 | 68.42±0.32 | 67.90±0.30 | 67.88±0.32 | – | – |
| coqa(llama) | 61.93±0.15 | 72.29±0.20 | 75.30±0.31 | 75.50±0.24 | 75.89±0.26 | 76.18±0.37 | 76.31±0.33 | 75.76±0.40 | 73.04±0.18 | 72.74±0.28 | 73.98±0.18 | 73.83±0.27 | 48.18±0.19 |
| coqa(llama2) | 61.48±0.19 | 71.68±0.21 | 74.55±0.25 | 74.56±0.18 | 74.81±0.28 | 75.04±0.22 | 74.91±0.97 | 73.99±0.63 | 72.29±0.25 | 71.54±0.26 | 72.48±0.25 | 72.73±0.27 | 51.12±0.25 |
| coqa(opt) | 64.53±0.17 | 69.76±0.20 | 72.22±0.71 | 72.76±0.78 | 73.21±0.21 | 73.67±0.26 | 74.30±0.28 | 72.91±0.31 | 70.87±0.22 | 70.94±0.30 | 71.44±0.27 | 70.89±0.17 | 46.75±0.28 |
| coqa(gpt) | 51.15±0.09 | 63.90±0.30 | 66.17±0.94 | 64.31±1.13 | 65.66±0.48 | 66.30±0.47 | 65.05±0.83 | 65.20±0.35 | 64.64±0.31 | 64.22±0.47 | 64.75±0.31 | – | – |
| nq(llama) | 69.54±0.35 | 70.63±0.26 | 73.64±0.60 | 74.05±0.53 | 74.53±0.24 | 74.69±0.31 | 75.33±0.36 | 74.54±0.33 | 71.80±0.21 | 71.26±0.24 | 71.80±0.21 | 71.88±0.28 | 59.02±0.58 |
| nq(llama2) | 69.18±0.38 | 70.99±0.56 | 73.49±0.44 | 73.62±0.34 | 74.15±0.31 | 74.08±0.57 | 74.63±0.46 | 74.02±0.30 | 72.18±0.50 | 71.78±0.41 | 71.96±0.50 | 71.42±0.34 | 52.81±0.31 |
| nq(opt) | 70.52±0.35 | 63.81±0.46 | 74.60±0.67 | 74.25±0.55 | 75.54±0.52 | 76.65±0.45 | 75.94±0.60 | 76.63±0.54 | 73.92±0.42 | 73.54±0.46 | 74.10±0.45 | 77.97±0.59 | 46.36±0.64 |
| nq(gpt) | 57.87±0.39 | 59.48±0.49 | 62.54±0.62 | 62.02±0.87 | 62.02±0.71 | 61.54±0.42 | 61.52±0.46 | 61.02±0.47 | 60.08±0.59 | 59.10±0.54 | 59.11±0.58 | – | – |
| **m = 10** | | | | | | | | | | | | | |
| trivia(llama) | 84.62±0.14 | 81.17±0.17 | 87.45±0.26 | 88.32±0.15 | 88.38±0.20 | 87.53±0.21 | 88.26±0.15 | 87.84±0.17 | 85.40±0.14 | 85.01±0.72 | 85.16±0.11 | 88.40±0.10 | 56.26±0.18 |
| trivia(llama2) | 85.18±0.12 | 81.61±0.09 | 88.12±0.42 | 88.80±0.28 | 88.92±0.36 | 88.06±0.38 | 88.81±0.28 | 88.31±0.37 | 86.28±0.11 | 86.21±0.12 | 85.88±0.12 | 89.20±0.08 | 64.40±0.14 |
| trivia(opt) | 81.18±0.11 | 72.45±0.18 | 83.93±0.68 | 85.76±0.19 | 85.45±0.36 | 84.60±0.18 | 85.86±0.13 | 85.77±0.15 | 82.37±0.09 | 82.88±0.09 | 83.63±0.09 | 85.24±0.11 | 46.22±0.17 |
| trivia(gpt) | 64.47±0.15 | 70.65±0.33 | 71.44±0.87 | 71.12±0.88 | 71.04±1.29 | 71.82±0.38 | 71.30±0.57 | 71.01±0.37 | 70.82±0.31 | 70.02±0.43 | 69.91±0.30 | – | – |
| coqa(llama) | 62.88±0.20 | 72.52±0.28 | 76.53±0.26 | 77.10±0.31 | 77.63±0.33 | 77.73±0.27 | 78.22±0.45 | 77.60±0.28 | 74.58±0.26 | 74.45±0.33 | 75.78±0.24 | 74.01±0.28 | 48.40±0.19 |
| coqa(llama2) | 62.90±0.20 | 71.76±0.20 | 75.71±0.29 | 75.94±0.28 | 76.38±0.20 | 76.41±0.28 | 76.61±0.40 | 75.38±0.35 | 73.27±0.24 | 72.73±0.29 | 73.76±0.24 | 72.73±0.29 | 51.29±0.24 |
| coqa(opt) | 66.42±0.20 | 70.49±0.25 | 73.94±0.63 | 75.08±0.25 | 75.09±0.76 | 75.28±0.23 | 76.24±0.49 | 75.00±0.25 | 72.31±0.22 | 72.48±0.36 | 72.89±0.20 | 71.63±0.21 | 46.79±0.28 |
| coqa(gpt) | 51.57±0.12 | 64.60±0.24 | 68.45±0.75 | 67.22±0.80 | 67.77±0.48 | 68.68±0.37 | 67.45±0.83 | 66.94±0.32 | 66.39±0.26 | 65.34±0.54 | 66.38±0.26 | – | – |
| nq(llama) | 70.61±0.47 | 71.31±0.42 | 74.50±1.32 | 75.64±0.38 | 75.88±0.68 | 75.96±0.42 | 77.01±0.33 | 76.20±0.35 | 73.10±0.32 | 72.46±0.41 | 73.03±0.29 | 72.77±0.34 | 58.89±0.64 |
| nq(llama2) | 69.63±0.49 | 71.31±0.46 | 74.47±0.80 | 74.85±0.51 | 75.55±0.47 | 75.39±0.44 | 76.06±0.38 | 75.33±0.52 | 73.30±0.46 | 72.80±0.45 | 73.13±0.48 | 72.28±0.44 | 52.99±0.36 |
| nq(opt) | 70.22±0.31 | 61.49±0.60 | 75.79±0.58 | 76.37±0.26 | 77.36±0.33 | 78.20±0.38 | 78.23±0.38 | 78.83±0.37 | 75.36±0.41 | 75.86±0.41 | 76.50±0.39 | 78.09±0.48 | 46.62±0.65 |
| nq(gpt) | 58.24±0.35 | 59.80±0.58 | 63.76±0.62 | 63.36±0.46 | 63.61±0.59 | 62.80±0.59 | 62.34±0.48 | 62.09±0.52 | 60.74±0.66 | 59.39±0.72 | 59.58±0.64 | – | – |
| **m = 20** | | | | | | | | | | | | | |
| trivia(llama) | 85.55±0.12 | 79.81±0.13 | 88.62±0.31 | 89.67±0.22 | 89.75±0.14 | 88.50±0.20 | 89.65±0.29 | 89.24±0.12 | 86.34±0.09 | 86.44±0.22 | 86.30±0.10 | 88.64±0.08 | 56.25±0.18 |
| trivia(llama2) | 86.29±0.12 | 80.39±0.11 | 89.18±0.36 | 89.95±0.31 | 90.22±0.28 | 89.04±0.40 | 90.20±0.21 | 89.75±0.28 | 87.20±0.08 | 87.36±0.21 | 86.92±0.09 | 89.45±0.09 | 64.40±0.16 |
| trivia(opt) | 81.53±0.12 | 70.38±0.18 | 85.10±0.13 | 86.99±0.10 | 86.65±0.09 | 85.41±0.27 | 86.93±0.15 | 86.96±0.12 | 82.97±0.09 | 83.68±0.14 | 84.63±0.08 | 85.45±0.11 | 46.31±0.17 |
| trivia(gpt) | 66.44±0.23 | 72.10±0.32 | 72.83±0.85 | 72.74±0.84 | 72.88±1.19 | 73.67±0.31 | 73.53±0.64 | 72.66±0.36 | 72.68±0.29 | 71.62±0.54 | 71.44±0.28 | – | – |
| coqa(llama) | 62.96±0.24 | 71.79±0.34 | 76.89±0.77 | 77.58±0.26 | 78.22±0.66 | 77.99±0.33 | 78.60±0.31 | 78.19±0.21 | 74.37±0.30 | 74.98±0.31 | 76.47±0.25 | 74.10±0.27 | 48.56±0.18 |
| coqa(llama2) | 63.36±0.24 | 71.76±0.22 | 76.50±0.66 | 77.06±0.44 | 77.54±0.41 | 77.55±0.26 | 78.38±0.55 | 76.75±0.32 | 73.72±0.25 | 73.33±0.22 | 73.96±0.16 | 72.98±0.28 | 51.59±0.24 |
| coqa(opt) | 66.40±0.14 | 70.18±0.20 | 74.60±0.70 | 75.66±0.40 | 76.34±0.50 | 75.70±0.38 | 77.39±0.57 | 76.12±0.22 | 72.46±0.23 | 72.75±0.36 | 73.58±0.19 | 71.73±0.19 | 46.82±0.26 |
| coqa(gpt) | 51.70±0.10 | 64.35±0.24 | 69.68±0.34 | 68.18±1.18 | 68.87±0.56 | 69.80±0.28 | 69.04±0.89 | 67.50±0.31 | 66.95±0.26 | 65.67±0.33 | 67.11±0.25 | – | – |
| nq(llama) | 70.08±0.46 | 71.34±0.40 | 75.49±1.09 | 76.13±0.35 | 76.63±0.31 | 76.51±0.43 | 77.80±0.38 | 76.89±0.39 | 73.47±0.32 | 72.36±0.28 | 73.29±0.28 | 73.30±0.41 | 58.78±0.60 |
| nq(llama2) | 69.52±0.46 | 71.09±0.43 | 75.44±0.70 | 75.70±0.39 | 76.39±0.32 | 76.07±0.41 | 76.65±0.35 | 76.22±0.44 | 74.11±0.45 | 73.22±0.52 | 73.66±0.42 | 72.97±0.47 | 53.07±0.30 |
| nq(opt) | 69.62±0.37 | 59.19±0.61 | 76.51±0.34 | 77.25±0.26 | 78.02±0.25 | 78.58±0.34 | 79.20±0.35 | 79.80±0.36 | 75.45±0.36 | 76.84±0.30 | 77.38±0.27 | 78.17±0.45 | 46.50±0.63 |
| nq(gpt) | 59.23±0.45 | 61.05±0.57 | 65.38±0.71 | 65.25±0.62 | 65.02±0.58 | 64.28±0.51 | 64.32±0.45 | 63.08±0.59 | 62.06±0.72 | 60.79±0.62 | 60.39±0.69 | – | – |

Table 14: Accuracy achieved by picking the most confident answer among $m = 20$ generations.

| | Base Accuracy | Ecc (C) | Deg (C) | Ecc (E) | Deg (E) | Ecc (J) | Deg (J) | P(true) |
|---|---|---|---|---|---|---|---|---|
| trivia(llama) | 61.18±0.07 | 68.27±0.12 | 71.78±0.11 | 74.04±0.32 | 75.75±0.15 | 74.25±0.06 | **76.19±0.10** | 55.19±0.15 |
| trivia(llama2) | 76.24±0.11 | 78.22±0.42 | **78.85±0.14** | 78.68±0.20 | **78.96±0.23** | 78.41±0.32 | **78.81±0.16** | 73.36±0.13 |
| trivia(opt) | 25.75±0.12 | 31.86±0.38 | 31.91±0.15 | 39.10±0.28 | 40.25±0.38 | 39.98±0.22 | **41.63±0.17** | 17.89±0.13 |
| trivia(gpt) | 87.42±0.08 | 87.74±0.16 | 87.91±0.20 | 88.08±0.15 | **88.36±0.11** | 87.96±0.12 | 88.15±0.11 | – |
| coqa(llama) | 62.46±0.11 | 63.02±0.25 | 73.48±0.14 | 73.96±0.68 | 76.53±0.33 | 74.73±0.17 | **77.41±0.12** | 56.97±0.21 |
| coqa(llama2) | 78.71±0.13 | 79.64±0.40 | **80.83±0.15** | 80.30±0.20 | **80.84±0.18** | 79.91±0.57 | **80.86±0.16** | 75.36±0.15 |
| coqa(opt) | 53.81±0.18 | 52.01±0.65 | 63.86±0.30 | 67.29±0.33 | 69.57±0.30 | 69.01±0.18 | **72.64±0.18** | 36.87±0.22 |
| coqa(gpt) | 79.76±0.14 | 79.59±0.30 | **80.22±0.47** | 80.13±0.17 | 79.70±0.09 | 79.58±0.17 | **80.27±0.14** | – |
| nq(llama) | 23.63±0.36 | 26.40±0.62 | 20.93±0.55 | 36.32±0.46 | 38.59±1.07 | 35.84±0.58 | **40.66±0.50** | 25.16±0.40 |
| nq(llama2) | 44.13±0.68 | 45.18±0.45 | 46.69±0.83 | **47.47±0.79** | **47.90±0.81** | 46.62±0.48 | **47.78±0.84** | 41.49±0.74 |
| nq(opt) | 8.60±0.18 | 8.35±0.79 | 8.03±0.29 | 14.26±0.26 | 17.25±0.33 | 16.26±0.61 | **18.33±0.40** | 5.62±0.29 |
| nq(gpt) | 62.72±0.39 | 63.10±0.69 | 63.43±0.55 | 64.07±0.69 | **66.05±0.52** | 63.70±0.56 | 63.90±0.56 | – |

Table 15: Accuracy achieved by picking the most confident answer among $m = 20$ generations, with temperature of the LLM set to 0.5.

| | Base Accuracy | Ecc (C) | Deg (C) | Ecc (E) | Deg (E) | Ecc (J) | Deg (J) | P(true) |
|---|---|---|---|---|---|---|---|---|
| trivia(llama) | 74.43±0.10 | 76.54±0.43 | **77.44±0.26** | **77.44±0.30** | **77.50±0.26** | 76.96±0.29 | **77.48±0.11** | 70.13±0.18 |
| trivia(llama2) | 75.92±0.11 | 77.90±0.25 | 78.58±0.27 | 78.60±0.15 | **78.91±0.28** | 78.44±0.20 | **78.80±0.15** | 70.94±0.13 |
| trivia(opt) | 40.65±0.13 | 44.17±0.51 | 45.30±0.15 | **45.58±0.26** | 45.10±0.17 | 44.68±0.18 | 45.12±0.18 | 34.69±0.16 |
| trivia(gpt) | 87.79±0.10 | **87.99±0.08** | 87.94±0.10 | 87.82±0.10 | **88.12±0.14** | 87.96±0.11 | **88.00±0.10** | – |
| coqa(llama) | 76.27±0.08 | 77.23±0.61 | 79.08±0.27 | 78.88±0.19 | **79.85±0.14** | 79.26±0.09 | 79.35±0.10 | 72.01±0.15 |
| coqa(llama2) | 78.36±0.12 | 79.04±0.33 | 80.63±0.23 | 79.71±0.43 | **81.00±0.22** | 80.19±0.47 | 80.56±0.13 | 73.20±0.17 |
| coqa(opt) | 70.40±0.15 | 72.31±0.16 | 73.78±0.21 | 72.97±0.43 | **74.08±0.19** | 72.93±0.58 | **74.19±0.16** | 62.99±0.21 |
| coqa(gpt) | 80.02±0.13 | 80.02±0.17 | **80.67±0.18** | 80.02±0.19 | 79.69±0.20 | 80.05±0.20 | 80.03±0.14 | – |
| nq(llama) | 40.72±0.56 | 42.24±0.65 | 44.34±0.69 | 45.20±0.63 | **46.73±0.86** | 44.21±1.24 | 45.34±0.71 | 38.46±0.74 |
| nq(llama2) | 44.01±0.64 | 46.05±0.75 | 46.76±0.68 | 46.89±0.63 | **47.93±0.63** | 46.64±0.61 | **47.76±0.75** | 41.48±0.58 |
| nq(opt) | 18.18±0.37 | 19.10±0.42 | 20.66±0.45 | **21.93±0.42** | 22.25±0.73 | 22.06±0.45 | **22.28±0.39** | 15.16±0.31 |
| nq(gpt) | 62.90±0.45 | 63.74±0.72 | 63.00±0.50 | 63.35±0.35 | **65.26±0.44** | 63.77±0.50 | 62.93±0.49 | – |

Table 16: Adaptive Calibration Error after applying Histogram Binning Zadrozny & Elkan (2001). Lower is better.

| ACE (in $10^{-2}$) | Baselines | | Ours | | | | | | | | | White-box | |
| --- | --- | --- | --- | --- | --- | --- | --- | --- | --- | --- | --- | --- | --- |
| | NumSet | LexiSim | EigV (C) | Ecc (C) | Deg (C) | EigV (E) | Ecc (E) | Deg (E) | EigV (J) | Ecc (J) | Deg (J) | SE | P(true) |
| trivia(llama) | 3.68±0.54 | 3.68±0.88 | 3.24±0.67 | 2.54±0.63 | 2.49±0.53 | 3.00±0.52 | 2.60±0.51 | 2.37±0.60 | 3.49±0.73 | 2.93±0.56 | 2.72±0.83 | 4.56±1.10 | 4.30±1.01 |
| trivia(llama2) | 3.06±0.49 | 3.05±0.71 | 2.65±0.53 | 2.82±0.69 | 2.75±0.63 | 2.76±0.39 | 2.73±0.64 | 2.67±0.65 | 2.93±0.60 | 3.11±0.68 | 3.01±0.49 | 2.58±0.59 | 4.43±0.71 |
| trivia(opt) | 3.20±0.70 | 3.34±0.65 | 3.16±0.62 | 2.38±0.58 | 2.84±0.86 | 3.11±0.51 | 2.15±0.50 | 2.64±0.51 | 2.83±0.44 | 2.62±0.60 | 2.59±0.53 | 3.16±0.62 | 4.19±0.79 |
| trivia(gpt) | 3.02±0.54 | 2.93±0.54 | 2.78±0.74 | 2.68±0.51 | 2.83±0.62 | 2.54±0.54 | 2.89±0.72 | 2.82±0.59 | 2.59±0.52 | 2.47±0.71 | 2.75±0.76 | – | – |
| coqa(llama) | 4.78±1.01 | 4.30±0.66 | 3.78±0.86 | 3.92±0.54 | 3.54±0.63 | 4.08±1.17 | 3.05±0.51 | 3.33±0.67 | 3.70±0.59 | 3.32±0.77 | 3.27±0.78 | 3.85±0.69 | 4.91±1.37 |
| coqa(llama2) | 3.68±0.40 | 3.67±0.38 | 3.87±0.63 | 3.32±0.44 | 3.62±0.80 | 3.58±0.60 | 3.07±0.85 | 3.07±0.64 | 3.38±0.36 | 3.58±0.78 | 3.27±0.72 | 3.62±0.54 | 4.16±0.73 |
| coqa(opt) | 4.79±0.86 | 4.10±0.76 | 4.04±0.78 | 4.73±0.89 | 3.61±0.53 | 4.04±0.58 | 3.69±0.97 | 3.39±0.60 | 4.59±0.73 | 3.74±0.90 | 3.79±0.68 | 4.06±0.67 | 4.48±0.51 |
| coqa(gpt) | 4.32±0.91 | 3.46±0.62 | 3.80±0.43 | 3.73±0.96 | 3.28±0.75 | 3.59±0.74 | 3.62±0.64 | 3.80±0.69 | 3.90±0.54 | 3.71±0.85 | 3.81±0.62 | – | – |
| nq(llama) | 4.71±0.96 | 3.54±0.72 | 4.07±0.92 | 4.51±1.05 | 4.23±0.78 | 4.20±0.66 | 3.82±0.94 | 3.86±0.57 | 4.12±0.73 | 3.80±0.57 | 3.57±0.86 | 4.45±1.03 | 4.66±1.08 |
| nq(llama2) | 5.02±0.72 | 4.68±0.69 | 5.01±0.94 | 5.52±1.33 | 5.23±1.03 | 4.99±1.24 | 4.69±0.89 | 4.87±1.10 | 5.17±0.92 | 5.22±1.11 | 5.25±1.40 | 5.38±0.96 | 5.89±1.16 |
| nq(opt) | 3.20±0.77 | 3.32±0.34 | 3.01±0.31 | 2.87±0.46 | 3.12±0.71 | 2.67±0.67 | 2.64±0.62 | 2.40±0.73 | 2.87±0.66 | 2.57±0.52 | 2.54±0.56 | 2.96±0.82 | 4.09±0.54 |
| nq(gpt) | 5.09±0.54 | 5.06±0.62 | 5.64±0.99 | 5.00±0.94 | 4.71±0.58 | 5.09±0.63 | 5.72±0.73 | 5.67±1.06 | 4.79±1.10 | 5.05±1.12 | 4.71±1.22 | – | – |

