# OpenReview forum: "Generating with Confidence: Uncertainty Quantification for Black-box Large Language Models"
_ICLR.cc/2024/Conference — Submitted to ICLR 2024_

### Official Review · Reviewer_sn9f · 2023-10-24

**Soundness:** 3 good
**Presentation:** 2 fair
**Contribution:** 3 good
**Rating:** 6
**Confidence:** 3

**Summary:**

The paper explores how to detect whether text generated from an LLM is accurate or not, without having access to the models probabilities (so called "black box" inference). It does so based on decoding multiple outputs (token sequences) for a single input via sampling and then exploring various measures for how much these multiple outputs agree with each other.
It reports experimental results across a range of different models on the task of question answering (QA). QA is chosen to make it easier to evaluate the accuracy of model outputs.
Relevant literature from other ML fields which have made greater progress than LLMs/NLP in quantifying model uncertainty are referenced.

**Strengths:**

On the whole the paper is well written. The topic of uncertainty quantification of LLMs is well motivated. The paper builds of some very recent work in this field, and reports good empirical results across several models which indicate the proposed methods for measuring black-box uncertainty are promising.
Repeating the experiments and reporting mean + stdDev is helpful for gaining confidence in the reported numbers.
It's interesting to see another example of how poorly white-box probabilities reflect accuracy, per tables 1,2,3.

**Weaknesses:**

The paper mentions several times that uncertainty is not the same as confidence. Confidence scores are defined based on the token probabilities, which are not available in black-box inference settings. The paper however isn't very clear in how uncertainty and confidence are not the same thing, with multiple sentences referring to "uncertainty/confidence" as though they are the same, and section 4.2. not written in a way that helps clarify the working definitions of how these differ either. This is frustrating for the reader.

The main limitation is only using question answering to test the models here. There are a spectrum of NLG tasks, and I agree with the authors that caring about uncertainty is less likely for completely open generation. However there are tasks like data-to-text (ie knowledge grounded) NLG, where the outputs need to be trusted, and it remains an open issue whether these methods reported here are transferable to tasks other than QA. The QA task is also interesting in that it is extracting answers from the LLMs weights, rather than using the LLM as a general tool for e.g. the data-to-text task, where the info is given as structural inputs.

**Questions:**

* In the introduction it's presented that LLMs are gaining attention -> it's crucial to quantify their uncertainty.  After reading the paper, I still have to ask why? It seems _crucial_ to improve their accuracy. Is the only rational for improving their uncertainty so that their accuracy can be improved, e.g. by blocking uncertain outputs? Or is there other reasons as well?
* How is the issue of model calibration different?
* Do you think these results transfer to other tasks such as data-to-text? Or do you think there may be different results beyond QA tasks?

---

> ### Author Response · Authors · 2023-11-18
> **Responses (1/2)**
>
> We thank the reviewer for the constructive review, and appreciating for our experiments and general motivation. We are also thankful for the useful feedback.
> We have made revisions to our manuscript as well as performed additional experiments following the reviewer's suggestions.
>
> ## Uncertainty vs Confidence
>
> Apologize for this confusion.
> To clarify, in short, $U$ is a property of the model's perceived posterior and depends only on $x$ (see L3-5 in Section 3.1, and Eq 1).
> $C$ depends on both $x$ and the answer $\mathbf{s}$ (L2-L3 of Section 3.2 and Eq 2).
> We have updated Section 3.2 to further clarify this.
>
> > Confidence scores are defined based on the token probabilities, which are not available in black-box inference settings
>
> Just want to clarify that Eq(1) and Eq(2) are only one possible uncertainty/confidence (that are however mainstream).
> Exact definition is just anything taking the form of $U(x)$ and $C(x,\mathbf{s})$.
> We hope the updated section 3.2 and the example helps clarify this.
>
> ## Non-QA NLP applications
>
> (We assume "data-to-text" means tasks like captioning. Please let us know if this is a misunderstanding.)
> Other NLP applications like data-to-text are definitely something we plan to explore next.
> This paper mostly just follows existing UQ literature and uses QA as it is the easiest to perform evaluation on it.
> Evaluation in general is still quite hard.
> In fact, we have seen very recent explorations on using model-predicted likelihoood as confidence scores on X-ray report generation task (but they also judge the quality of the report with ROUGE).
>
> We believe that a lot of the ideas are transferrable and we would like note that CoQA is to some extent "data-to-text" except that the data is in textual form as well.
> Our methods do not assume that the input has to be text, as we are only comparing the generations' similarities.
> If the input is data, we think there are two possible solutions:
> We skip the data (image or audio) and directly use Jaccard or NLI model on the question and answers, OR;
> We fine-tune a domain-specific NLI model by training a "bridging" module that "translates" the other modalities to embeddings that the NLI model understands.

---

> ### Author Response · Authors · 2023-11-18
> **Responses (2/2)**
>
> ## Why UQ for NLG is important
> > In the introduction it's presented that LLMs are gaining attention -> it's crucial to quantify their uncertainty. After reading the paper, I still have to ask why? It seems crucial to improve their accuracy. Is the only rational for improving their uncertainty so that their accuracy can be improved, e.g. by blocking uncertain outputs? Or is there other reasons as well?
>
> To begin with, improving the accuracy is important, but selective generation (i.e. blocking uncertain outputs) is not just about improving accuracy.
> For example, an influential work [1] in selective classification (an area with a similar motivation, but with much more literature) aims at providing risk guarantees after the selection.
>
> Secondly, if improving accuracy *on the un-rejected samples* is important, then finding a good UQ measure for such selective generation task is also important as a better UQ measure could lead to higher accuracy at the same rejection rate.
> A bad UQ, such as the "Random" baseline in Table 1, leads to no improvement.
>
> Finally, selective generation/classification is one of the most important applications of UQ, but in general UQ is about conveying this information about uncertainty to the decision maker (human or algorithm) [2].
> In UQ literature, people tend to focus on AUROC to check whether the UQ measure is reliable, and we also use AUARC (a metric related to selective generation) as we think it is more directly related to real-world applications.
> They are both machinery to probe the reliability of UQ measure, whose true importance is due to that decision-making processes intrinsically need to have a reliablity measure on the prediction.
>
> [1] Geifman, Yonatan, and Ran El-Yaniv. "Selective classification for deep neural networks." Advances in neural information processing systems 30 (2017).
>
> [2] Seoni, Silvia, Vicnesh Jahmunah, Massimo Salvi, Prabal Datta Barua, Filippo Molinari, and U. Rajendra Acharya. "Application of uncertainty quantification to artificial intelligence in healthcare: A review of last decade (2013–2023)." Computers in Biology and Medicine (2023): 107441.
>
> ## UQ vs Calibration
> > How is the issue of model calibration different?
>
> (See end of Section 2 for a discussion, and a newly added Appendix C.5 for calibration experiments.)
> The UQ we focus on tries to find a measure of uncertainty or confidence that is *discriminative* of what's uncertain and what's not.
> In other words, we care about whether the *ranking* of our UQ measure could effectively predict the reliability of the answers.
> This is also why most literature uses AUROC as an evaluation metric.
> Calibration cares about the discrepancy between the predicted probability and the frequency.
> (Such "discriminative-ness" is often referred to as "sharpness" and is considered orthogonal to calibration [3].)
> In the extreme case - if we know the model's accuracy is 0.6 on average, and we keep predicting a answer's confidence is 0.6 regardless of the answer, we get something that is prefectly calibrated, yet this constant confidence is not useful because it doesn't tell us which answer is more reliable than others (i.e. not "sharp").
>
> [3] Gneiting, T., Balabdaoui, F., and Raftery, A. E. Probabilistic forecasts, calibration and sharpness. Journal of the Royal Statistical Society: Series B (Statistical Methodology), 69 (2):243–268, 2007.

---

### Official Review · Reviewer_YpC9 · 2023-11-01

**Soundness:** 3 good
**Presentation:** 3 good
**Contribution:** 3 good
**Rating:** 6
**Confidence:** 3

**Summary:**

This paper presents an innovative approach for generating confidence and uncertainty scores within the context of natural language generation (NLG), while working under the realistic constraint of no white-box access to the underlying model. In this scenario, where only the generated sequences are observable and the language model (LLM) can be queried repeatedly for confidence and uncertainty assessments, the proposed method leverages pair-wise similarity or entailment probabilities between generated samples to construct a weight graph. Spectral clustering is employed on that graph to compute confidence and uncertainty scores. Intriguingly, the paper's results suggest that this method can outperform traditional white-box techniques in certain instances, highlighting its potential for enhancing NLG performance.

**Strengths:**

- Explores an important problem: with the advent of LLMs it is necessary to address how to compute confidence and uncertainty estimates which was easier to obtain in traditional deep learning setting.
- Paper is well written and easy to follow
- The paper proposes an interesting idea to use spectral clustering to produce uncertainty and confidence measures.
- Evaluates performance on different QA datasets and different LLMs as well.

**Weaknesses:**

- It might be helpful to measure expected or adaptive calibration error to show this method outputs confidence scores which are better in comparison to other methods.
- Relevant work: https://arxiv.org/pdf/2306.13063.pdf, might be helpful to include comparison between their method and your proposed method.

**Questions:**

- As pointed out in the weakness, for confidence scores measuring ECE or ACE might be helpful in showcasing the effectiveness of the approach.
- In cases where we have access to model embeddings, I think same method of clustering could be applied where we use similarity between embeddings instead of entailment prob/Jaccard similarity. Would this method give better confidence or uncertainty estimates in those settings as well?

---

> ### Author Response · Authors · 2023-11-18
> **Responses**
>
> We thank the reviewer for the constructive review and acknowledging our contributions vis-a-vis experiments and presentation, as well as for providing useful feedback.
> In pursuance to the suggestions made by the reviewer, we have made revisions to our manuscript, and performed some additional experiments.
>
> ## ECE or ACE (calibration measures)
>
> Thank you for the suggestion.
> We tried to distinguish the focus of our paper at the end of Section 2.
> Just like in our baseline papers, we focus on the uncertainty estimation, and we care more about the *ranking* of such measures - that is, whether a high confidence means highly reliable prediction or accuracy.
> All uncertainty or confidence estimates could be conformalized or calibrated to bear concrete meaning in the frequentist sense, by using a calibration dataset.
> There is a large collection of calibration literature that we can leverage here.
> In repsonse to your suggestion, we have also run additional experiments using histogram binning to calibrate all methods, and report the ACE in the Appendidx C.3.
>
> ## Additional Reference
> Thank you for the suggestion and we have included this reference to our revision for completeness.
> After reading the paper, we found that this method is more suitable for questions with exact answers (e.g. multiple choice or arithmetics) as the consistency-based component requires comparing exact matches of predictions, which is quite hard to achieve for examples like Figure 6 in the Appendix.
> The "verbalized confidence" is similar to the P(true) baseline we tried.
>
> ## Use of model embeddings
>
> > Would this method (using model embeddings) give better confidence or uncertainty estimates in those settings as well?
>
> Thank you for the question.
> While we did not perform the experiments ourselves (and is not a black-box method), we noticed that our baseline paper's (Kuhn et al. 2023) official implementation tried this idea.
> As it was not used nor reported in their paper, it is probably not performing well.

---

### Official Review · Reviewer_LJF2 · 2023-11-01

**Soundness:** 3 good
**Presentation:** 3 good
**Contribution:** 3 good
**Rating:** 6
**Confidence:** 3

**Summary:**

The paper proposes a new method for uncertainty quantification for NLG when the model used is a black-box LLMs (no access to logits). The framework used consists of three steps (1) response samples generation, (2) pairwise similarity calculation between responses, and (3) uncertainty and confidence score estimation. Using this framework, the authors compare different estimation methods using number of semantic sets, sum of eigenvalues, degree matrix, and eccentricity. For the pairwise similarity, they compare the standard Jaccard similarity as well as model-based method using predicted probabilities from a natural language inference (NLI) model. Experiment results show that the proposed uncertainty and confidence score estimation outperform the baselines, and sometimes better than white-box methods.

**Strengths:**

- A novel approach to estimate uncertainty and model confidence.
- Thorough experiments on multiple QA datasets and public LLMs.

**Weaknesses:**

- The paper addresses an interesting question that relevant to most of the current LLM works. However, I feel the work is not fully complete, with many results but lack of analysis and insights on what to be the main findings of the paper. There are also some missing details, e.g. what DeBERTa model used for NLI? Was it trained using NLI data?
- There is a discussion regarding uncertainty vs. model confidence and why it matters, but along the way the differentiation is not clear - Sec 3.2 does not even differentiate the two in details, where I expect that the discussion should be there because of the subsection title.
- There are many variations/ablations regarding the methods used, i.e. similarity metrics and also uncertainty/confidence measurement but little analysis on which perform best, why, and what is the recommendation for future work if they want to quantify uncertainty / model confidence.
- I am also a bit concerned regarding evaluation with GPT model. Although the authors perform human verification on a subset, but the subset is pretty small (33 samples per dataset). I think many of these datasets are factual, thus human evaluation might not be as tricky as say creative writing task, so having more human verification on the results would make the claim stronger. There isn't also explanation about inter-annotator agreement on the verification. Though, I appreciate that the authors acknowledge the limitation of using GPT models for evaluation.

**Questions:**

- What is the reasoning for a_{NLI, contra}? If you use a standard NLI model which has three classes, Eq 4 (right) would take into account the entailment and neutral class (since it is 1-p_{contra}, is that correct?
- Unclear why uncertainty is used for expected accuracy and model confidence for individual accuracy.

Suggestions for paper:
- Provide examples on when the uncertainty and confidence estimation aligns/doesn't align with the prediction
- More depth analysis on why a particular method (similarity metric/quantification method) outperform the other methods used in the experiment

---

> ### Author Response · Authors · 2023-11-18
> **Responses**
>
> We are grateful to the reviewer for the constructive comments.
> To begin, we would like to emphasize that this paper focuses on the uncertainty quantification of the LLM, and the experiment setups largely follow the white-box baseline (Kuhn et al. 2023).
> We have made revisions to our manuscript as well as performed additional experiments following the reviewer's suggestions.
> Below we present responses to each of the points raised:
>
> ## Clarification of Details.
>
> > what DeBERTa model used for NLI? Was it trained using NLI data?
>
> For a fair comparison, we followed (Kuhn et al. 2023) and used the same official pretrained DeBERTa-large on hugginface, which was fine-tuned on MNLI data.
> More training details of this model could be found on the model card of `deberta-large-mnli` on huggingface.
>
>
>
> ## Uncertainty vs Confidence
>
> We apologize for the confusion here.
> The structure was that 3.1 defines uncertainty and 3.2 defines confidence (which is different).
> $U$ is a property of the model's perceived posterior and depends only on $x$ (see L3-5 in Section 3.1, and Eq 1).
> $C$ depends on both $x$ and the answer $\mathbf{s}$ (L2-L3 of Section 3.2 and Eq 2).
> These are commonly used terms in two somewhat disconnected domains and we have made updates to Section 3.2 to emphasize this distinction.
>
> ## Recommended Variation / Why a particular method outperforms
>
> Thank you for the constructive question/suggestion.
> In section 5.3, under *Uncertainty Measures*, our conclusion was that the similarity measure (Section 4.1) seems more important than variants proposed in Section 4.2.
> We hypothesize that "entail" works better because generally the output space is huge, and for two generations, "not contradicting" is likely not a sufficient indicator of low-uncertainty (as they can both be bad random responses).
> We have updated section 5.3 and Appendix B.7 with examples to reflect this hypothesis.
>
> ## GPT evaluation
>
> We would like to clarify that the number of evaluations is *99* per dataset instead of 33 (33 per (model, dataset) pair  * 3 models)
> The number of total human evaluations is chosen basing on (Kuhn et al. 2023), which evaluated 200 answers for two datasets (we have 297 on three).
> We tried our best to verify the reliability of GPT evaluation, but the verification process is actually quite slow, as for CoQA we need to read the passage first before judging the answers, and even for TriviaQA or NQ, sometimes seemingly unrelated answers are just two names of the same person and require careful research.
> We agree that a larger-scale human evaluation with mulitple annotators is more desirable, but the main focus of this paper is UQ, and we think the topic of automatic evaluation is an important research area by itself.
> Existing UQ literature almost always use purely lexical measures like ROUGE-based automatic evaluation, and we believe the GPT evaluation is better (please refer to Appendix B.3 for a discussion).
>
>
> To address your concern, we increased the number of annotations to 200 per dataset (50 per (model, dataset) * 4 models) and the new accuracies are 89.4/96.5/93 for coqa/trivia/nq.
> Using the same annotations, the ROUGE-L based metric's accuracy is 81.5/93.5/85.
> If we construct a confidence interval we could reject that "GPT and ROUGE are equally good" (our estimate is that further shrinking the confidence interval by half requires about 20 hours of annotation time).
> We found that llama2 (newly added) tends to generate different expressions of the same answer, which is sometimes judged incorrect by GPT.
> Excluding llama2, the GPT accuracies are 90.6/98/94 for coqa/trivia/nq.
>
>
>
> ## Other Questions
>
>
> > What is the reasoning for a_{NLI, contra}? If you use a standard NLI model which has three classes, Eq 4 (right) would take into account the entailment and neutral class (since it is 1-p_{contra}, is that correct?
>
> $a_{NLI, contra}$ accounts for both neutral and entailment.
> We could consider the difference between $a_{NLI, contra}$ and $a_{NLI, entail}$ as whether we think "neutral" is closer to “similar” or “dissimilar”.
>
> > Unclear why uncertainty is used for expected accuracy and model confidence for individual accuracy.
>
> The setting of uncertainty + individual accuracy was in Table 11 (now 10), and we discuss in Appendix C.2 why we choose not to include it in the main text.
> Essentially, since $U$ does not depend on a particular generation (i.e. it is a property of the model's posterior, for a random generation), we think it should be used to predict the (estimated) expected accuracy (which also depends on the posterior/a random generation).
>
> > Provide examples on when the uncertainty and confidence estimation aligns/doesn't align with the prediction
>
> We've added more examples (e.g. Fig 5-7) in the Appendix in the updated pdf, but we are not entirely sure with what you mean by aligning with the prediction.
> Could you check if the new examples address your concerns, or clarify your suggestion? Thank you!

---

> > ### Comment · Reviewer_LJF2 · 2023-11-23
> >
> > Thanks for your response and clarification. The paper is clearer now and I've updated my score accordingly.

---

### Official Review · Reviewer_1J6D · 2023-11-03

**Soundness:** 3 good
**Presentation:** 2 fair
**Contribution:** 3 good
**Rating:** 6
**Confidence:** 3

**Summary:**

The paper proposed several metrics to estimate the model output uncertainty for a given question without access to the logic, i.e., close models (openAI models). The proposed metric uses three types of similarity methods to estimate multiple model generations for a question, then uses the pair-wise similarity score to estimate the model’s uncertainty on a given question with different metrics like the eigenvalue of the graph laplacian. They later evaluate the metrics compared with baselines and white-box uncertainty measures. The results showed improvement.

**Strengths:**

1. The paper proposed an interesting idea of estimating black-box generation output without access to the output logits.
2. The motivation is strong, and the paper is timely as the wide usage of close source models estimating model uncertainty without logits could help downstream tasks avoid using uncertain model output.
3. The proposed method is intuitive and reasonable to be applied to this situation.

**Weaknesses:**

1. Some more analysis should be conducted, e.g., the impact of similarity functions and intuitions of why one similarity function is more preferred than the other.
2. Some quantitative examples would be great, such as what kind of questions are evaluated as uncertain as the proposed metric vs the baselines, etc.
3. The presentation and writing of the paper can be more intuitive. Tables 1, 2, and 3 are very hard to read; an alternative is to include the important numbers and put the rest into the appendix.

**Questions:**

1. Is there any reason for the definition of the degree matrix?
2. Have you tried other similarity measures, such as cosine similarity, using existing word embeddings?

---

> ### Author Response · Authors · 2023-11-18
> **Responses**
>
> We appreciate the positive feedback on the soundness and overall contribution
> of our work---we are glad that the reviewer found it intuitive and interesting.
> However, we would also like to take this opportunity to respond to some concerns regarding the clarity/presentation of our work.
> We have made revisions to our manuscript as well as performed additional experiments following the reviewer's suggestions.
>
>
> ## Analysis on the impact of similarity functions
>
> Thank you for this constructive suggestion.
> We had some discussion in Appendix B.7.
> To clarify this further, we have updated section 5.3 and Appendix B.7 with some examples to reflect our hypothesis. Newly added text is reflected in blue.
>
>
>
> ## Discussion of advantages over baselines
>
> Thank you for the suggestion.
> The only two black-box baselines are NumSet nad LexiSim.
> NumSet is discrete, so our method (EigV in particular) has an advantage over it as it's more granular.
> LexiSim is purely lexical based, and we have some discussion of why meaning-based measures are more useful in Appendix right before B.4.
> We have also updated the appendix B.7 to include discussion and example of why the proposed measures are better than the baselines.
>
> ## Readability of Table 1-3
>
> Thank you for the feedback!
> If you notice, we had already moved most of the results to the Appendix, but due to the large number of experiments we ran it is quite hard to reduce the information.
> The general message is that the middle rows ("Ours") tend to perform better.
> We have followed your suggestion and also moved AUROC (Table 3) to the Appendix now.
>
>
> ## Questions
>
>
> > Have you tried other similarity measures, such as cosine similarity, using existing word embeddings?
>
>
> This is an interesting idea and seems related to the Jaccard similarity.
> We just finished running some experiments using the largest Glove embedding (glove.840B.300d) where an answer's embedding is computed as the average embedding of its words, and it performs significantly worse than Jaccard.
> While we haven't fully explored this idea due to limited time, we did notice that the baseline paper (Kuhn et al. 2023) seems to have tried the sentence embeddings in their code release but didn't use it in the paper, and we assumed that sentence embedding-based method didn't work well.
>
>
> > Is there any reason for the definition of the degree matrix?
>
> The degree matrix (Eq.7) was used because it seems like quite a straightforward measure of ``dispersion'', as $C_{deg}$ is just the average distance to other nodes.

---

### Author Response · Authors · 2023-11-22

Dear Reviewers,

Thanks again for your insightful and thorough feedback. We'd appreciate it if you could inform us if we have addressed your queries or if additional clarification is required.

Best regards,
Submission8527 Authors

---

### Meta-Review · Area_Chair_66rn · 2023-12-09

**Metareview:**

In this paper, the authors explore different ways to quantify the uncertainty associated with question answering in large language models.   The methods explored, in-general, involve different ways of measuring either similarity or dispersion of a set of predicted sequences.  The authors use these various methods on samples drawn from llama, opt, and GPT in a black-box fashion to perform selective prediction on QA tasks and to draw out AUROC and "AUARC" curves.

Some of the reviewers found the paper well written and easy to follow.  They seemed to find the work well motivated and interesting to the community.  The review scores were all borderline accept with very little dispersion (score of 6).  Weaknesses were that the presented methods were rather simple, the evaluation was limited to QA tasks and reviewers asked for more insightful analysis and discussion.  Multiple reviewers wanted to see how well the computed uncertainty lined up with accuracy, through e.g. calibration measurements.  Also, a few wanted to see some more analysis of why particular methods did better than others.  A couple of them complained that the usage of the words uncertainty and confidence seemed inconsistent or confusing.

The paper is quite borderline.  The reviewers all felt is was just above the acceptance threshold, but unfortunately, none of the reviewers seemed willing to champion the paper.  It seemed that the reviewers were left wanting more in terms of analysis.  As it stands, the paper seems to present a number of things that were tried and what works best, but the reviewers wanted to have more to take away about why.  This suggests the paper could be stronger than its current form, and hopefully that would also increase the impact of the work.  Although the work is quite close to being ready for acceptance, it seems that it could benefit from another round of improvements, additional analysis and careful rewriting.

This doesn't factor in the decision, but please note, the discussion about the Bayesian perspective isn't really correct.  I think the association that the authors are trying to make is with probabilistic machine learning and not necessarily Bayesian ML.  I.e. using the variance as a measure of uncertainty is a standard frequentist approach.  Bayesian statistics involves integrating over uncertainty, and in the deep learning setting this usually means integrating over uncertainty in the model parameters.  The authors should essentially replace every mention of "Bayesian" with "probabilistic".

**Justification For Why Not Higher Score:**

Honestly this paper is *very* borderline and I wouldn't complain if it was pushed up.  It just seems underwhelming in terms of the contribution.  The authors tried a bunch of stuff on black-box model api's and reported the results.  It seems like the results certainly could be useful to practitioners, but I'm not sure we're learning a whole lot from the paper.

**Justification For Why Not Lower Score:**

NA

---

### Decision · Program_Chairs · 2024-01-16

Reject